# AdaSCALE: Adaptive Scaling for OOD Detection

## Abstract

The ability of the deep learning model to recognize when a sample falls outside its learned distribution is critical for safe and reliable deployment. Recent state-of-the-art out-of-distribution (OOD) detection methods leverage activation shaping to improve the separation between in-distribution (ID) and OOD inputs. These approaches resort to sample-specific scaling but apply a static percentile threshold across all samples regardless of their nature, resulting in suboptimal ID-OOD separability. In this work, we propose **AdaSCALE**, an adaptive scaling procedure that dynamically adjusts the percentile threshold based on a sample's estimated OOD likelihood. This estimation leverages our key observation: OOD samples exhibit significantly more pronounced activation shifts at high-magnitude activations under minor perturbation compared to ID samples. AdaSCALE enables stronger scaling for likely ID samples and weaker scaling for likely OOD samples, yielding highly separable energy scores. Our approach achieves state-of-the-art OOD detection performance, outperforming the latest rival OptFS by **14.94%** in near-OOD and **21.67%** in far-OOD datasets in average FPR@95 metric on the ImageNet-1k benchmark across eight diverse architectures.

## 1 Introduction

The reliable deployment of deep learning models hinges on their ability to handle previously unseen inputs, a task commonly known as OOD detection. One critical application is in medical diagnosis, where a model trained on common diseases should be able to flag inputs representing unknown conditions as potential outliers, requiring further review by clinicians. OOD detection primarily involves identifying semantic shifts, with robustness to covariate shifts being a highly desirable characteristic [1, 2]. As modern deep learning models scale in both data and parameter counts, effective OOD detection within large-scale settings is critical. Given the difficulties of iterating on large models, *post-hoc* approaches that preserve ID accuracy are generally preferred.

A variety of post-hoc approaches have emerged, broadly categorized by where they operate. One class of methods focuses on computing OOD scores directly in the output space [6, 7, 8, 9, 3, 10], while another operates in the activation space [11, 12, 13, 14]. Finally, a more recent line of research also explores a hybrid approach [15, 16], combining information from both spaces. The efficacy of many high-performing methods relies on either accurate computation of ID statistics [17, 18, 19, 20, 21] or retention of training data statistics [12, 14]. However, as retaining full access to training data becomes increasingly impractical in large-scale settings, methods that operate effectively with minimal ID samples without performance degradation are particularly valuable for practical applications.

Alleviating the dependence on ID training data/statistics, recent state-of-the-art post-hoc approaches center around the concept of "fixed scaling." ASH [3] prunes and scales activations on a per-sample basis. SCALE [4], the direct successor of ASH, critiques pruning and focuses purely on scaling, which improves OOD detection without accuracy degradation. LTS [5] extends this concept by directly scaling logits instead, using post-ReLU activations. These methods leverage a key insight:

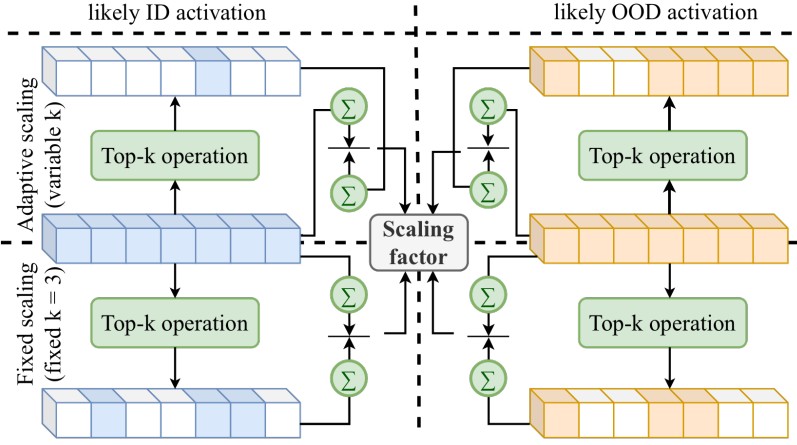

Figure 1: **Adaptive scaling (AdaSCALE) vs. fixed scaling (ASH [3], SCALE [4], LTS [5]).** While fixed scaling approaches uses a constant percentile threshold $p$ and hence constant $k$ (e.g., $k = 3$) across all samples, AdaSCALE adjusts $k$ based on estimated OOD likelihood. AdaSCALE assigns larger $k$ values (e.g., $k = 5$) to OOD-likely samples, producing smaller scaling factors, and smaller $k$ values (e.g., $k = 1$) to ID-likely samples, yielding larger scaling factors. This adaptive mechanism enhances ID-OOD separability. (See Figure 4 for complete working mechanism.)

scaling based on the relative strength of a sample's *top-k* activations (with respect to the entire activations) produces highly separable ID-OOD energy scores. However, although such approaches provide sample-specific scaling factors, the scaling mechanism remains uniform across all samples as the percentile threshold $p$ and thereby $k$ is fixed, as shown in Figure 1. This static approach is inherently limiting for optimal ID-OOD separation while also failing to leverage even minimal ID data, which could be reasonably practical in most deployment scenarios.

We hypothesize that designing an adaptive scaling procedure based on each sample's predetermined OOD likelihood offers greater control for enhancing ID-OOD separability. Specifically, this mechanism should assign smaller scaling factors for samples with high OOD likelihood to yield lower energy scores and larger scaling factors for probable ID samples to yield higher energy scores. To achieve this, we propose a heuristic for predetermining OOD likelihood based on a key observation in activation space: minor perturbations applied to OOD samples induce significantly more pronounced shifts in their top-k activations compared to ID samples. Consequently, samples exhibiting substantial activation shifts are assigned lower scaling factors, while those with minimal shifts receive higher scaling factors. This adaptive scaling mechanism can be applied in either logit or activation space. Our method, **AdaSCALE**, achieves state-of-the-art performance, delivering significant improvements in OOD detection while requiring only minimal ID samples.

We conduct an extensive evaluation across 8 architectures on ImageNet-1k and 2 architectures on CIFAR benchmarks, demonstrating the substantial effectiveness of AdaSCALE. For instance, AdaSCALE surpasses the average performance of the *best-generalizing* method, OptFS [10], by **14.94%/8.96%** for near-OOD detection and **21.48%/3.39%** for far-OOD detection in terms of FPR@95 / AUROC, on the ImageNet-1k benchmark across eight architectures. Furthermore, AdaS-CALE outperforms the *best-performing* method, SCALE [4], when evaluated on the ResNet-50 architecture, achieving performance gains of **12.95%/6.44%** for near-OOD and **16.79%/0.79%** for far-OOD detection. Additionally, AdaSCALE consistently demonstrates superiority in full-spectrum OOD (FSOOD) detection [1]. Our key contributions are summarized as follows:

- We reveal that OOD inputs exhibit more pronounced shifts in top-k activations under minor perturbations compared to ID inputs. Leveraging this, we propose a novel post-hoc OOD detection method using adaptive scaling that attains state-of-the-art OOD detection.

- We demonstrate state-of-the-art generalization of AdaSCALE via extensive evaluations across many setups. AdaSCALE requires tuning just one additional percentile hyperparameter compared to SCALE for a given setup, while the other introduced hyperparameters generalize well across all 10 architectures and 3 datasets.

## 2 Related Works

**Post-hoc methods.** Early research on OOD detection primarily focused on designing scoring functions based on logit information [6, 7, 8, 9, 22]. While these methods leveraged logit-based scores, alternative approaches have explored gradient-based information, such as GradNorm [23], GradOrth [24], GAIA [25], and Greg-OOD [26]. Given the limited dimensionality of the logit space, which may not encapsulate sufficient information for OOD detection, subsequent studies have investigated activation-space-based methods. These approaches exploit the high-dimensional activations, leading to both parametric techniques such as MDS [11], MDS Ensemble [11], and RMDS [13], as well as non-parametric methods such as KNN-based OOD detection [12, 27]. Recent advancements have proposed hybrid methodologies that integrate parametric and non-parametric techniques to improve robustness. For instance, ComboOOD [14] combines these paradigms to enhance near-OOD detection performance. Similarly, VIM [15] employs a combination of logit-based and distance-based metrics. However, reliance of such approaches on ID statistics [20, 28, 29] can become a constraint, hindering scalability and practical deployment in real-world applications. To mitigate computational challenges for real-world deployment, recent methods, such as FDBD [30] and NCI [31], have focused on enhancing efficiency. Recent advances, such as NECO [32] examines connections to neural collapse phenomena, while WeiPer [33], explore class-direction perturbations. Unlike WeiPer, our work deals with perturbation in the input image similar to ODIN [7].

**Activation-shaping post-hoc methods.** A seminal work in OOD detection, ReAct [17], identified abnormally high activation patterns in OOD samples and proposed clipping extreme activations. This approach has been further generalized by BFAct [18] and VRA [19], which extend activation clipping for enhanced effectiveness. Additionally, BATS [34] refines activation distributions by aligning them with their respective typical sets, while LAPS [35] enhances this strategy by incorporating channel-aware typical sets. Inspired by activation clipping, another line of research explores activation "scaling" as a means to improve OOD detection. ASH [3] introduces a method to compute a scaling factor as a function of the activation itself, pruning and rescaling activations to enhance the separation of energy scores between ID and OOD samples. However, this approach results in a slight degradation in ID classification accuracy. In response, SCALE [4] observes that pruning adversely affects performance and thus eliminates it, leading to improved OOD detection while preserving ID accuracy. SCALE currently represents the state-of-the-art method for ResNet-50-based OOD detection. Despite their efficacy, these activation-based methods exhibit limited generalization across diverse architectures. To address this issue, LTS [5] extends SCALE by computing scaling factors using post-ReLU activations and applying them directly to logits rather than activations. Our work builds on this line of work, introducing the adaptive scaling mechanism. ATS [21] argues that relying solely on final-layer activations may result in the loss of critical information beneficial for OOD detection and proposes to leverage intermediate-layer activations too. However, its efficacy is contingent upon the availability of a large number of training samples, whereas our approach attains state-of-the-art performance while utilizing a minimal number of ID samples. A newly proposed method OptFS [10] introduces a piecewise constant shaping function with the goal of generalization across diverse architectures in large-scale settings, while our work exhibits superior generalization extending to small-scale settings too.

**Training methods.** The training methods incorporate adjustments during training to enhance the ID-OOD differentiating characteristics. They either make architectural adjustments [36, 37, 38], apply enhanced data augmentations [39, 40, 41], or make simple training modifications [42, 43, 44]. More recent methods have adopted contrastive learning in the context of OOD detection [45, 46, 47, 48]. Moreover, some approaches also either utilize external real outliers [49, 50, 51, 52] or synthesize virtual outliers either in image space [53, 54, 55, 56, 57, 58, 59, 60] or in feature space [61, 62, 63, 64]. However, training methods can be costlier and less effective than post-hoc approaches in some large-scale setups [65].

## 3 Preliminaries

Let $\mathcal{X}$ denote the input space and $\mathcal{Y} = \{1, 2, ..., C\}$ denote the label space, where $C$ is the number of classes. We consider a multi-class classification setting where a classifier $h$ is trained on ID data drawn from an underlying joint distribution $\mathcal{P}_{\text{ID}}(x, y)$, where $x \in \mathcal{X}$ and $y \in \mathcal{Y}$. The ID

training dataset is denoted as $\mathcal{D}_{\text{ID}} = \{(x_i, y_i)\}_{i=1}^N$, where $N$ is the number of training samples and $(x_i, y_i) \sim \mathcal{P}_{\text{ID}}(x, y)$. The classifier $h$ is composed of a feature extractor $f_\theta : \mathcal{X} \to \mathcal{A} \in \mathbb{R}^D$, and a classifier $g_{\mathcal{W}} : \mathcal{A} \to \mathcal{Z} \in \mathbb{R}^C$. The feature extractor maps an input $x$ to a feature vector $\mathbf{a} \in \mathcal{A}$, where $\mathbf{a} = f_\theta(x)$ and the classifier then maps this feature vector to a logit vector $\mathbf{z} = g_{\mathcal{W}}(\mathbf{a}) \in \mathbb{R}^C$. We refer to individual dimensions of the feature vector $\mathbf{a}$ as activations, denoted by $a_j$ for the $j$-th dimension. The classifier $h$ is trained on $\mathcal{D}_{\text{ID}}$ to minimize the empirical risk: $\min_{\theta, \mathcal{W}} \frac{1}{N} \sum_{i=1}^N \mathcal{L}(g_W(f_\theta(x_i)), y_i)$ where $\mathcal{L}$ is a loss function, such as cross-entropy loss. During inference, the model may encounter data points drawn from a different distribution, denoted as $\mathcal{P}_{\text{OOD}}(x)$, which is referred to as OOD data. The OOD detection problem aims to identify whether a given input $x$ is drawn from marginal distribution $\mathcal{P}_{\text{ID}}(x)$ or from $\mathcal{P}_{\text{OOD}}(x)$. Hence, the goal is to design a scoring function $S(x) : \mathcal{X} \to \mathbb{R}$ that assigns a scalar score to each input $x$, reflecting its likelihood of being an OOD sample. A higher score typically indicates a higher probability of the input being OOD. A threshold $\tau$ is used to classify an input as either ID or OOD: $\text{OOD}(x) = \begin{cases} \text{True,} & \text{if } S(x) > \tau \\ \text{False,} & \text{if } S(x) \leq \tau \end{cases}$.

## 4 Method

In this section, we introduce AdaSCALE, a novel post-processing approach that dynamically adapts the scaling mechanism based on each sample's estimated OOD likelihood. We first present our key empirical observations regarding activation behavior under minor perturbations, building upon insights from ReAct [17]. Next, we revisit and analyze the core principle underlying recent state-of-the-art approaches. Finally, we detail our proposed adaptive scaling mechanism that leverages these observations to achieve superior OOD detection performance.

### 4.1 Observations in Activation Space

A seminal work ReAct [17] demonstrated that OOD samples often induce abnormally high activations within neural networks. We extend this finding with an important observation: *the positions of such high activations in OOD samples are relatively unstable under minor perturbations compared to ID samples*. This instability provides a valuable signal for distinguishing OOD samples from ID samples. Below, we formalize this observation and our methodology.

#### 4.1.1 Perturbation Mechanism

Let $x \in \mathbb{R}^{C_{\text{in}} \times H \times W}$ be an input image with $C_{\text{in}}$ input channels, $H$ height, and $W$ width. We denote channel value at position $(c, h, w)$ as $x[c, h, w]$. To identify channel values for perturbation, we employ pixel attribution that quantifies each input element's influence on the model's prediction. An attribution function, $AT(x, c, h, w)$, assigns a score to each channel value, with *lower* absolute scores indicating *less* influence. We select $o\%$ of channel value indices with *lowest* absolute attribution scores, forming the set $R$. We use a gradient-based attribution:

$$AT(x, c, h, w) = \frac{\partial (g_W(f_\theta(x)))_{y_{pred}}}{\partial x[c, h, w]} \tag{1}$$

where $y_{pred}$ is the predicted class index. To create a perturbed input, we select a subset $R$ containing $o\%$ of channel values to perturb. The perturbed image $x^\varepsilon$ is obtained as:

$$x^\varepsilon[c, h, w] = \begin{cases} x[c, h, w] + \varepsilon \cdot \text{sign}(AT(x, c, h, w)), & \text{if } (c, h, w) \in R \\ x[c, h, w], & \text{if } (c, h, w) \notin R \end{cases} \tag{2}$$

where $\varepsilon$ is perturbation magnitude. While we employ gradient-based attribution for principled pixel selection for perturbation, as we show later in Appendix C.3, it is important to note that even random selection empirically performs similarly, whereas selecting salient pixels degrades performance.

#### 4.1.2 Activation Shift as OOD Indicator

After obtaining the perturbed input $x^\varepsilon$, we compute its activation $\mathbf{a}^\varepsilon = f_\theta(x^\varepsilon)$. We define the *activation shift* as the absolute element-wise difference between the original activation and the perturbed activation:

$$\mathbf{a}^{\text{shift}} = |\mathbf{a}^\varepsilon - \mathbf{a}| \tag{3}$$

Figure 2 illustrates the key insight of our approach: activation shift at extreme (high-magnitude) activations is consistently more pronounced in OOD samples compared to ID samples. This behavior can be understood intuitively: ID samples activate network features in a stable, predictable manner reflecting learned patterns, while OOD samples trigger less stable, more arbitrary high activations that shift significantly under perturbation. Based on this observation, we propose using activation shift at the top-$k_1$ highest activations as a metric to estimate OOD likelihood of a sample:

$$Q = \sum_{j \in \text{argsort}(\mathbf{a}, \ \text{desc=True})[:k_1]} (|a_j^{\varepsilon} - a_j|) \quad (4)$$

where $\text{argsort}(\mathbf{a}, \ \text{desc} = \text{True})[: k_1]$ returns the indices of the $k_1$ highest values in $\mathbf{a}$. As evidenced by $Q_{\text{OOD}}/Q_{\text{ID}}$ ratio $(> 1)$ shown in Figure 3, the $Q$ statistic generally assigns higher values to OOD samples than ID ones. However,

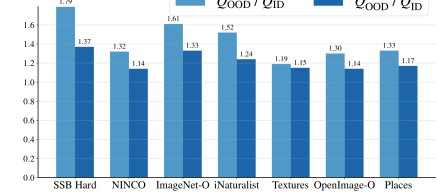

Figure 2: Activation shift comparison (with the mean denoted by a solid line and the standard deviation by a shaded region) between ID and OOD in the ResNet-50 model. The activation shift is significantly more pronounced in OOD samples compared to ID samples at high-magnitude activations (left side of the x-axis), providing a discriminative signal for OOD detection.

the high variance of $Q$ metric (Figure 2) suggests the possibility of overoptimistic estimations. To address this issue, we introduce a correction term $C_o$ that exhibits an opposing behavior: it tends to be higher for ID samples than for OOD samples. Figure 8 in Appendix C shows that the perturbed activations of ID samples tend to be higher than those of OOD ones, especially in high-activation regions. We leverage this complementary signal by defining:

$$C_o = \sum_{j \ \in \ \text{argsort}(\mathbf{a}, \ \text{desc=True})[:k_2]} \text{ReLU}(a_j^{\varepsilon}) \quad (5)$$

where $k_2$ is a hyperparameter denoting the number of considered activations. We refine our OOD quantification by combining both metrics, weighted by a hyperparameter $\lambda$:

$$Q' = \lambda \cdot Q + C_o \quad (6)$$

Indeed, Figure 3 illustrates that $Q'_{\text{OOD}}/Q'_{\text{ID}} > Q_{\text{OOD}}/Q_{\text{ID}}$, suggesting that the correction term $C_o$ helps mitigate overconfident estimations. If $\bar{Q}_s = \{\bar{Q}'_1, \bar{Q}'_2, ..., \bar{Q}'_{n_{\text{val}}}\}$ be the set of $Q'$ values on $n_{\text{val}}$ ID validation samples, we could transform any $Q'$ into a normalized probability scale by constructing empirical cumulative distribution function (eCDF) derived from $\bar{Q}_s$. The eCDF, denoted as $F_{Q'}(Q')$, can be defined as:

$$F_{Q'}(Q') = \frac{1}{n_{\text{val}}} \sum_{i=1}^{n_{\text{val}}} \mathbb{1}(\bar{Q}'_i \leq Q') \quad (7)$$

Figure 3: $Q_{\text{OOD}}/Q_{\text{ID}}$ vs $Q'_{\text{OOD}}/Q'_{\text{ID}}$ in various OOD datasets with ResNet-50 on ImageNet-1k. $Q'_{\text{OOD}}/Q'_{\text{ID}} > Q_{\text{OOD}}/Q_{\text{ID}}$ suggests $C_o$ helps mitigate overconfident estimations.

where $\mathbb{1}(\cdot)$ is the indicator function. A higher value of $F_{Q'}(Q')$ indicates a higher likelihood of the sample being OOD. Importantly, our experiments suggest that as few as 10 ID validation samples are sufficient to construct an effective eCDF for this purpose (See Table 6).

---

*Remark: ODIN vs. AdaSCALE in terms of perturbation*

ODIN [7] perturbs entire image, inducing stronger confidence in ID inputs than OOD ones. In contrast, we apply trivial perturbations, perturbing only small number of trivial/random pixels to primarily compute shifts in top-k activations.

---

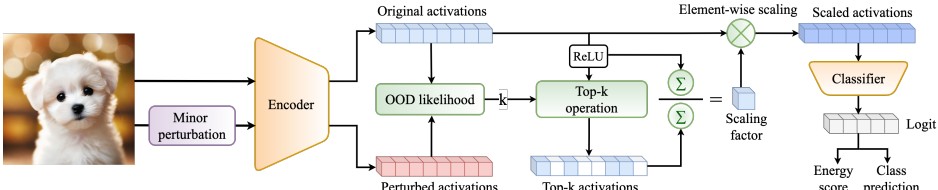

Figure 4: **Schematic diagram of AdaSCALE's working mechanism.** AdaSCALE computes activation shifts between an original image and its slightly perturbed counterpart to estimate OOD likelihood. This likelihood determines an adaptive percentile threshold ($p$ and thereby $k$), which controls the scaling factor $r$. Since $r$ is defined as the ratio of total activation sum to the sum of activations above the percentile threshold, samples with higher OOD likelihood receive lower scaling factors. This adaptive approach ensures stronger scaling for ID samples and weaker scaling for OOD samples, yielding highly separable energy scores that enable effective OOD detection.

## 4.2 Revisiting Static Scaling Mechanism

Scaling baselines [3, 4, 5] operate by scaling activations / logits with scaling factor $r$ computed as:

$$r = \left( \frac{\sum_j \mathbf{a}_j}{\sum_{\mathbf{a}_j > P_p(\mathbf{a})} \mathbf{a}_j} \right) \tag{8}$$

where $P_p(\mathbf{a})$ denotes the $p^{th}$ percentile of all elements in activation $\mathbf{a}$. While this approach yields sample-specific scaling factors, it imposes a critical constraint: the $p^{th}$ percentile threshold is static and identical across all test samples, regardless of the nature of samples. We argue that this static nature limits the effectiveness of the scaling procedure and prevents optimal ID-OOD separability.

## 4.3 Proposed Approach: Adaptive Scaling

Building on our observations, we propose AdaSCALE (Adaptive SCALE), a novel approach that introduces dynamic, sample-specific adjustments to the scaling procedure. The key insight is that $p^{th}$ percentile threshold should be a function of each test sample's estimated OOD likelihood rather than a fixed value. The scaling factor $r$ increases as the $p^{th}$ percentile threshold rises (i.e., when more activations are excluded from the denominator in Equation 8). For optimal ID-OOD separation, we must scale ID samples more strongly than OOD samples, requiring a higher $p^{th}$ percentile for ID samples. We define an adaptive percentile threshold as:

$$p = p_{\min} + (1 - F_{Q'}(Q') \cdot (p_{\max} - p_{\min}) \tag{9}$$

where $p_{\min}$ and $p_{\max}$ are hyperparameters that define the minimum and maximum limits of percentile threshold. It ensures samples with lower OOD likelihood receive higher percentile thresholds, resulting in stronger scaling. (See Algorithm 1). We implement two variants: **AdaSCALE-A** scales activations as $\mathbf{a}_{\text{scaled}} = \mathbf{a} \cdot \exp(r)$ [3, 4]. **AdaSCALE-L** scales logits as $\mathbf{z}_{\text{scaled}} = \mathbf{z} \cdot r^2$ [5]. We use energy score $-\log \sum_{i=1}^{C} e^{(\mathbf{z}_i)}$ on (directly or indirectly) scaled logits, with higher values indicating higher ID likelihood. This approach enables per-sample dynamic scaling, as outlined in Figure 4.

## 5 Experiments

We use pre-trained models provided by PyTorch for ImageNet-1k experiments. For CIFAR experiments, we train three models per network using the standard cross-entropy loss and report the mean results across these three independent trials. The evaluation setup is provided in Table 1.

**Metrics.** We use two commonly used OOD Detection metrics: Area Under Receiver-Operator Characteristics (AUROC) and False Positive

Table 1: Experimental evaluation setup for OOD detection.

| **Conventional OOD detection** | | | |
|---|---|---|---|
| **ID datasets** | **Near-OOD** | **Far-OOD** | **Network** |
| CIFAR-10/100 | CIFAR-100 [66]/10 [67] TIN [70] | MNIST [68], SVHN [69], Textures [71], Places365 [72] | WRN-28-10, DenseNet-101 |
| ImageNet-1k | SSB-Hard [73] NINCO [75] ImageNet-O [76] | iNaturalist [74], OpenImage-O [15] Textures [71] Places [72, 17] | EfficientNetV2-L, ResNet-101 DenseNet-201, ViT-B-16 ResNet-50, ResNeXt-50 RegNet-Y-16, Swin-B |
| **Covariate shifted datasets for full spectrum OOD detection** | | | |
| ImageNet-1k | ImageNet-C [77], ImageNet-R [78], ImageNet-V2 [79], ImageNet-ES [2] | | |

Table 2: OOD detection results (FPR@95 ↓ / AUROC ↑) on ImageNet-1k benchmark.

| | Method | ResNet-50 | ResNet-101 | RegNet-Y-16 | ResNeXt-50 | DenseNet-201 | EfficientNetV2-L | ViT-B-16 | Swin-B | Average |
|---|---|---|---|---|---|---|---|---|---|---|
| near-OOD | MSP | 74.23 / 60.21 | 71.96 / 67.25 | 62.22 / 80.74 | 73.25 / 67.86 | 73.44 / 67.29 | 72.51 / 80.76 | 86.72 / 68.62 | 87.11 / 69.82 | 75.18 / 70.32 |
| | MLS | 74.87 / 64.55 | 72.05 / 71.51 | 62.94 / 84.66 | 74.11 / 71.62 | 75.51 / 68.91 | 81.44 / 79.22 | 93.78 / 63.64 | 94.80 / 64.68 | 78.69 / 71.10 |
| | EBO | 75.32 / 64.52 | 72.32 / 71.54 | 62.80 / 84.76 | 74.21 / 71.61 | 75.85 / 68.68 | 82.86 / 77.15 | 94.37 / 59.19 | 95.34 / 59.79 | 79.13 / 69.66 |
| | ReAct | 72.61 / 68.81 | 68.07 / 75.00 | 70.73 / 75.37 | 70.96 / 74.13 | 69.97 / 73.65 | 72.36 / 71.39 | 86.63 / 68.35 | 82.64 / 73.26 | 74.25 / 72.50 |
| | ASH | 69.47 / 71.33 | 65.24 / 76.61 | 82.51 / 67.81 | 70.98 / 75.25 | 92.83 / 52.30 | 94.85 / 44.78 | 94.45 / 53.20 | 96.37 / 47.58 | 83.34 / 61.11 |
| | SCALE | 67.76 / 74.20 | 63.87 / 78.60 | 67.09 / 82.90 | 70.59 / 76.20 | 71.56 / 73.72 | 89.70 / 60.12 | 94.48 / 56.18 | 88.62 / 61.47 | 76.71 / 70.42 |
| | BFAct | 72.35 / 68.88 | 67.96 / 75.16 | 78.72 / 66.09 | 70.96 / 74.14 | 71.20 / 72.61 | 75.53 / 62.46 | 82.09 / 70.66 | **71.81 / 75.28** | 73.83 / 70.66 |
| | LTS | 68.01 / 73.37 | 63.91 / 78.27 | 69.82 / 80.75 | 70.27 / 76.20 | 71.29 / 74.56 | 87.30 / 73.63 | 88.83 / 67.43 | 86.61 / 67.22 | 75.76 / 73.93 |
| | OptFS | 69.66 / 70.97 | 65.46 / 75.83 | 73.53 / 75.21 | 69.27 / 74.84 | 71.74 / 72.10 | 72.29 / 75.29 | 76.55 / 72.73 | 76.81 / 74.06 | 71.91 / 73.88 |
| | **AdaSCALE-A** | **58.98 / 78.98** | 57.96 / 81.68 | **47.91 / 89.18** | 64.14 / 79.96 | **61.28 / 79.66** | **53.78 / 86.94** | **71.87 / 73.14** | 73.41 / 74.48 | **61.17 / 80.50** |
| | **AdaSCALE-L** | 59.84 / 78.62 | **56.41 / 81.86** | 56.13 / 87.11 | **62.08 / 80.18** | 61.75 / 80.06 | 54.95 / 85.77 | 71.99 / 73.23 | 72.89 / 74.58 | 62.00 / 80.18 |
| far-OOD | MSP | 53.15 / 84.06 | 50.41 / 90.08 | 53.07 / 84.21 | 53.60 / 84.43 | 54.74 / 87.92 | 56.41 / 84.62 | 73.39 / 82.02 | | 54.83 / 85.14 |
| | MLS | 42.57 / 88.19 | 43.89 / 88.30 | 32.92 / 93.70 | 44.91 / 87.97 | 48.43 / 87.44 | 68.64 / 84.80 | 81.89 / 81.42 | 95.16 / 73.37 | 57.30 / 85.65 |
| | EBO | 42.72 / 88.09 | 44.30 / 88.23 | 32.47 / 93.82 | 45.12 / 87.86 | 48.95 / 87.15 | 74.48 / 81.13 | 84.69 / 76.34 | 96.08 / 63.99 | 58.88 / 83.33 |
| | ReAct | 30.14 / 92.98 | 29.89 / 93.10 | 45.20 / 86.17 | 30.06 / 92.69 | 30.72 / 92.65 | 60.05 / 75.33 | 59.31 / 83.65 | 58.86 / 84.77 | 43.03 / 87.67 |
| | ASH | 24.69 / 94.43 | 26.18 / 94.06 | 59.65 / 83.94 | 29.17 / 93.47 | 33.50 / 92.17 | 96.56 / 41.57 | 95.98 / 52.16 | 98.23 / 43.20 | 57.99 / 74.38 |
| | SCALE | 21.44 / 95.39 | 22.54 / 95.05 | 32.16 / 94.16 | 30.62 / 93.54 | 33.17 / 92.70 | 89.63 / 62.58 | 88.36 / 72.32 | 86.59 / 66.77 | 50.56 / 84.06 |
| | BFAct | 29.46 / 93.01 | 29.43 / 93.04 | 58.69 / 77.22 | 29.71 / 92.67 | 32.45 / 92.29 | 66.72 / 65.70 | 51.58 / 85.77 | **38.99 / 88.47** | 42.13 / 86.02 |
| | LTS | 22.20 / 95.24 | 23.07 / 94.94 | 34.99 / 93.57 | 30.37 / 93.49 | 30.92 / 93.29 | 86.85 / 76.30 | 64.37 / 84.43 | 85.84 / 44.80 | 47.33 / 84.51 |
| | OptFS | 25.66 / 93.87 | 26.97 / 93.55 | 47.37 / 86.73 | 27.54 / 93.40 | 34.42 / 91.04 | 53.62 / 83.62 | 46.11 / 87.35 | 44.27 / 87.79 | 38.25 / 89.67 |
| | **AdaSCALE-A** | **17.84 / 96.14** | **18.51 / 95.95** | 21.37 / 95.84 | **22.08 / 95.24** | 28.01 / 93.23 | 37.61 / 91.48 | 47.63 / 86.83 | 47.81 / 87.14 | **30.11 / 92.73** |
| | **AdaSCALE-L** | 17.92 / 96.12 | 19.15 / 95.76 | 20.10 / 96.19 | 22.16 / 95.01 | 28.00 / 93.18 | 38.81 / 90.51 | 47.28 / 86.97 | 46.24 / 87.97 | 29.96 / 92.71 |

Rate at 95% True Positive Rate (FPR@95), where a higher AUROC and lower FPR@95 indicates better OOD detection performance.

**Baselines.** We consider the following post-hoc methods: MSP [6], EBO [8], ReAct [17], MLS [9], ASH [3], SCALE [4], BFAct [18], LTS [5], OptFS [10]. Currently, SCALE is the *best-performing* method (with ResNet-50), while OptFS is the *best-generalizing* method.

**Hyperparameters.** The hyperparameters are determined via automatic parameter search [65, 80]. Although AdaSCALE may appear to require many hyperparameters, our findings indicate that setting $(\lambda, k_1, k_2, o, \epsilon)$ to $(10, 1\%, 5\%, 5\%, 0.5)$ consistently yields near-optimal performance across all setups, only requiring $(p_{min}, p_{max})$ to be tuned for any given architecture. (See Appendix D.) The best results are **bold**, and the second-best results are underlined across all results.

## 5.1 Empirical Results

**ImageNet-1k benchmark:** We compare our proposed method, AdaSCALE, with recent state-of-the-art approaches across eight architectures on the ImageNet-1k benchmark, as presented in Table 2. AdaSCALE demonstrates consistently strong performance across all architectures compared to existing methods. Specifically, it surpasses the *best-generalizing* method, OptFS, by **14.94%**/**8.96%** in the FPR@95/AUROC metric for near-OOD detection across all architectures. Additionally, it outperforms the *best-performing method*, SCALE (on ResNet-50), by **12.96%**/**6.44%** in the same metric. A closer observation reveals that while OptFS excels in architectures such as EfficientNet, ViT-B-16, and Swin-B, scaling baselines perform comparably or even better in architectures like ResNet-50, ResNet-101, RegNet-Y-16, and DenseNet-201. In contrast, AdaSCALE-A achieves the best performance in near-OOD detection across all architectures, except for Swin-B, where BFAct performs optimally. Furthermore, effectiveness of AdaSCALE extends beyond near-OOD detection to far-OOD detection, demonstrating an average gain of **21.67%** over OptFS in the FPR@95 metric.

**CIFAR benchmark:** We also compare AdaSCALE with post-hoc baselines on CIFAR benchmarks using WRN-28-10 and DenseNet-101 networks, reporting the averaged performance in Table 3. AdaSCALE outperforms all methods in average AUROC metric across CIFAR benchmarks in both near- and far-OOD detection. For far-OOD detection on CIFAR-10 benchmark, AdaSCALE-A achieves the best FPR@95 score of **33.11**, outperforming the MSP baseline by approximately 1.4 points. Similarly, AdaSCALE-A attains the best FPR@95 / AUROC of **43.07 / 90.31** in near-OOD detection, though MSP remains competitive. In near-OOD detection on CIFAR-100 benchmark,

Table 3: OOD detection results (FPR@95↓ / AUROC↑) averaged over WRN-28-10 and DenseNet-101 on CIFAR benchmarks across 3 trials. (See Appendix E for complete results.)

| Method | CIFAR-10 | | CIFAR-100 | |
|---|---|---|---|---|
| | Near-OOD | Far-OOD | Near-OOD | Far-OOD |
| MSP | 43.18 / 89.07 | 34.49 / 90.88 | 55.64 / 80.23 | 61.73 / 76.82 |
| MLS | 51.54 / 89.33 | 39.62 / 91.68 | 57.24 / 81.25 | 60.19 / 78.92 |
| EBO | 51.54 / 89.37 | 39.58 / 91.75 | 57.45 / 81.10 | 60.12 / 78.96 |
| ReAct | 49.71 / 88.59 | 37.32 / 92.00 | 63.20 / 79.58 | 54.78 / 80.46 |
| ASH | 78.11 / 77.97 | 63.12 / 83.35 | 80.97 / 70.09 | 69.38 / 79.06 |
| SCALE | 53.00 / 89.20 | 39.27 / 91.93 | 58.38 / 81.00 | 57.19 / 80.56 |
| BFAct | 54.90 / 88.56 | 43.05 / 90.66 | 72.26 / 74.70 | 57.44 / 77.63 |
| LTS | 55.71 / 88.77 | 41.06 / 91.74 | 59.98 / 80.60 | 80.48 / 81.79 |
| OptFS | 64.82 / 85.72 | 47.67 / 89.99 | 76.80 / 73.02 | 60.23 / 77.76 |
| **AdaSCALE-A** | **43.07 / 90.31** | **33.11 / 92.66** | 57.33 / 81.35 | 54.53 / 81.14 |
| **AdaSCALE-L** | 44.71 / 90.14 | 33.43 / 92.69 | 58.70 / 81.07 | 52.49 / 82.21 |

AdaSCALE-A achieves the highest AUROC of **81.35**, while in far-OOD detection, AdaSCALE-L reaches the best performance with FPR@95 / AUROC of **52.49 / 82.21**. While activation-shaping methods perform well in ImageNet-1k, they seem to underperform in CIFAR. In contrast, AdaSCALE achieves consistently superior performance across all setups.

**FSOOD Detection:** FSOOD detection extends conventional OOD detection by incorporating model's ability to generalize on covariate-shifted ID inputs. We present FSOOD detection results in Table 4. We can observe that this is a highly challenging task, as covariate-shifted ID datasets cause a significant performance drop for all methods compared to the conventional case. Despite this, AdaSCALE outperforms OptFS by **4.49** and **4.13** points on average in the FPR@95 metric for FSOOD detection across both near- and far-OOD datasets.

Table 4: FSOOD detection results on ImageNet-1k averaged over 8 architectures.

| Method | Near-OOD | | Far-OOD | |
|---|---|---|---|---|
| | FPR@95 ↓ | AUROC ↑ | FPR@95 ↓ | AUROC ↑ |
| MSP | 86.75 | 50.13 | 74.28 | 65.67 |
| ReAct | 87.22 | 51.38 | 67.23 | 69.53 |
| ASH | 87.01 | 52.02 | 72.36 | 65.70 |
| SCALE | 86.75 | 52.27 | 69.36 | 68.97 |
| BFAct | 87.12 | 51.14 | 66.13 | 69.69 |
| LTS | 86.46 | 53.29 | 66.76 | 71.63 |
| OptFS | 85.83 | 52.17 | 63.32 | 71.44 |
| **AdaSCALE-A** | **81.34** | 55.03 | **58.87** | 72.41 |
| **AdaSCALE-L** | 81.62 | **55.14** | 59.19 | **72.85** |

**Accuracy:** Like SCALE and LTS, AdaSCALE applies linear transformations to scale activations or logits, preserving accuracy, unlike post-hoc rectification methods [3, 17].

## 5.2 Ablation / hyperparameter studies

**Predetermined OOD likelihood** $Q'$**.** Adaptive scaling depends on predetermined OOD likelihood to determine the extent of scaling. We study the effect of various predetermined OOD likelihood functions on OOD detection using ResNet-50 network (ImageNet-1k) in Table 5. It clearly shows Q component of $Q'$ being most critical while $\sum_{k=1}^{k_2} \mathbf{a}_{\text{argsort}(\mathbf{a})_k}^{\varepsilon}$ as correction term being a relatively superior choice. However, predetermined OOD likelihood alone – without adaptive scaling – does not result in strong performance.

Table 5: Ablation studies of $Q'$ in FPR@95 ↓ / AUROC ↑ format.

| $Q'$ | Near-OOD | Far-OOD |
|---|---|---|
| $Q$ without scaling | 79.81 / 72.32 | 84.00 / 68.13 |
| $Q$ | 59.43 / 78.14 | 19.70 / 95.73 |
| $\sum_{k=1}^{k_2} \mathbf{a}_{\text{argsort}(\mathbf{a})_k}^{\varepsilon}$ | 70.39 / 74.00 | 21.40 / 95.31 |
| $\lambda \cdot Q + \sum_{k=1}^{k_2} \mathbf{a}_{\text{argsort}(\mathbf{a})_k}^{\varepsilon}$ | 58.97 / **78.98** | **17.84** / **96.14** |
| $\sum_{k=1}^{k_2} \max_k (\mathbf{a})_k$ | 65.11 / 76.23 | 19.76 / 95.67 |
| $\lambda \cdot Q + \sum_{k=1}^{k_2} \max_k (\mathbf{a})$ | **58.91** / 78.74 | 18.02 / 96.08 |

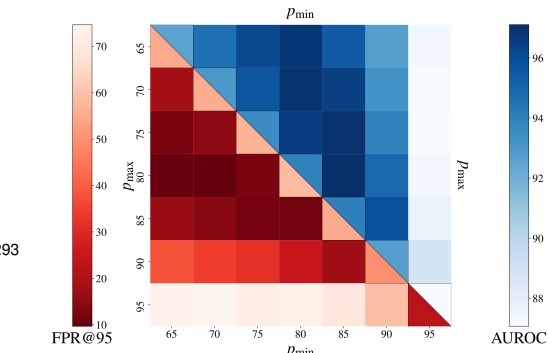

Figure 5: OOD detection performance on ImageNet-1k with varying $p_{\min}$ and $p_{\max}$. Diagonal entries ($p_{\min} = p_{\max}$) represent SCALE, while rest entries represent AdaSCALE.

**Adaptive percentile.** Unlike SCALE which uses constant percentile, AdaSCALE uses dynamic percentile lying in $[p_{\min}, p_{\max}]$ range adaptive to each sample. We show the effect of various percentile limit ranges in Figure 5 in the form of a heatmap on the AUROC validation metric on ImageNet-1k benchmark. The extent of darkness in the heatmap conveys a strong performance (corresponding to highest AUROC / lowest FPR@95). The diagonal entries, representing the results of SCALE, are lighter in comparison to the rest of the cells, denoting the results of AdaSCALE. Hence, it can be observed that using adaptive percentile leads to relatively better OOD detection performance in comparison to static percentile.

**ID statistics.** With the rise of large models, where training data is often undisclosed or inaccessible, relying on full training ID datasets for OOD detection has become increasingly impractical. We rigorously assess AdaSCALE's effectiveness with limited data by conducting experiments on ImageNet-1k with ResNet-50 using $n_{\text{val}}$ ID samples to compute ID statistics, where $n_{\text{val}} \in \{10, 100, 1000, 5000\}$. Table 6 confirms that even with substantially restricted access to ID data, AdaSCALE-A achieves state-of-the-art performance.

Table 6: AdaSCALE-A with restricted access to ID data using ResNet-50 network in FPR@95 ↓ / AUROC ↑ format.

| $n_{\text{val}}$ | **Near-OOD** | **Far-OOD** |
|---|---|---|
| 10 | 59.69 / 78.52 | 18.25 / 96.03 |
| 100 | 59.05 / 78.92 | **17.79** / 96.13 |
| 1000 | 58.99 / 78.95 | 17.86 / 96.13 |
| 5000 | **58.97** / **78.98** | 17.84 / **96.14** |

**Image perturbation study.** A sufficiently small perturbation, as discussed in Section 4.3, is used for deriving scaling factor. We now systematically investigate the impact of the extent and nature of perturbation on pre-determining OOD likelihood which is in-turn responsible for OOD detection. We present the results in Table 7. It is clearly evident that perturbing trivial pixels (with $o = 5\%$) leads to better OOD detection. Another key takeaway is that perturbing even random pixels achieves comparable performance more efficiently, whereas targeting salient pixels results in worse performance.

Table 7: Perturbation study (FPR@95↓ / AUROC↑) with ResNet-50 on ImageNet-1k. (See Table 11 for complete results.)

| Pixel type | $o\%$ | Near-OOD | Far-OOD |
|---|---|---|---|
| Random | 5% | 59.97 / 78.67 | 18.14 / 96.06 |
| | 50% | 62.81 / 76.27 | 19.95 / 95.70 |
| Trivial | 5% | **58.97** / **78.98** | **17.84** / **96.14** |
| | 50% | 66.43 / 74.10 | 21.58 / 95.29 |
| Salient | 5% | 67.31 / 75.78 | 21.24 / 95.44 |
| | 50% | 64.54 / 75.39 | 20.48 / 95.61 |
| All | 100% | 67.37 / 73.08 | 22.93 / 95.01 |

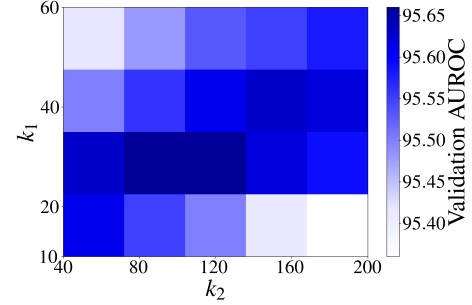

Figure 6: Sensitivity of $k_1$ and $k_2$.

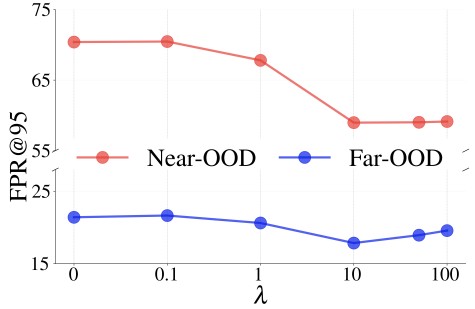

Figure 7: Sensitivity of $\lambda$.

**Sensitivity of $k_1$, $k_2$, and $\lambda$.** As discussed in Section 4.1.2, ID and OOD distinctions are more pronounced in high activations but diminish as more activations are included. We analyze the impact of $k_1$ (used in activation shift) and $k_2$ (used in perturbed activation) using a heatmap of validation AUROC with ResNet-50 (Figure 6). The darker region indicates higher AUROC, suggesting optimal values of $k_1 \approx 1\%$ (20) and $k_2 \approx 5\%$ (100) for ResNet-50 model. Furthermore, the heatmap suggests that $k_1$ is far more critical hyperparameter than $k_2$. The hyperparameter $\lambda$ controls the weighting of $Q$ in computing $Q'$, the predetermined OOD likelihood. The sensitivity analysis is presented in Figure 7 which shows near-OOD and far-OOD detection using FPR@95. It suggests optimal FPR@95 is achieved at $\lambda \approx 10$. Please refer to Appendix C.2 for sensitivity study of $\varepsilon$ and Appendix C.4 for compatibility study with ISH [4] regularization.

**Latency.** AdaSCALE incurs extra forward pass to compute perturbed activation $\mathbf{a}^\epsilon$. Also, top-k operations (time complexity: $\mathcal{O}(D \log D)$) are applied to $Q$ and $C_o$ to estimate OOD likelihood. Comparing variable vs. fixed percentiles for scaling in Table 8 over 10,000 trials, we observe that variable percentiles induce higher latency, though the latency ratio decreases with higher-dimensional activation spaces.

Table 8: Latency with fixed vs. variable percentile.

| | $D = 128$ | $D = 512$ | $D = 1024$ | $D = 2048$ | $D = 3024$ |
|---|---|---|---|---|---|
| Fixed percentile (SCALE) | $33\mu s$ | $40\mu s$ | $45\mu s$ | $48\mu s$ | $54\mu s$ |
| Variable percentile (AdaSCALE) | $152\mu s$ | $149\mu s$ | $155\mu s$ | $152\mu s$ | $164\mu s$ |
| Latency ratio (AdaSCALE / SCALE) | 4.66 | 3.76 | 3.42 | 3.14 | 3.02 |

# 6 Conclusion

We propose **AdaSCALE**, a novel post-hoc OOD detection method that dynamically adjusts the scaling process based on a sample's estimated OOD likelihood. Leveraging the observation that OOD samples exhibit larger activation shifts under minor perturbations, AdaSCALE assigns stronger scaling to likely ID samples and weaker scaling to likely OOD samples, enhancing ID-OOD separability. AdaSCALE achieves state-of-the-art performance as well as generalization across architectures requiring negligibly few ID samples, making it highly practical for real-world deployment.

# 7 Broader Impacts

This work has positive impact in trustworthy deep learning by enabling detection of OOD samples.

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

# Appendix

 ## A  Notations

 Table 9 lists all the notations used in this paper.

Table 9: Table of Notations

| Notation | Meaning |
|---|---|
| $\mathcal{X}$ | Input space. |
| $\mathcal{Y}$ | Label space. |
| $C$ | Number of classes. |
| $C_{\text{in}}$ | Number of input channels. |
| $h$ | Classifier. |
| $\mathcal{P}_{\text{ID}}(x, y)$ | Underlying joint distribution of ID data. |
| $\mathcal{P}_{\text{OOD}}(x)$ | Distribution of OOD data. |
| $\mathcal{D}_{\text{ID}}$ | ID training dataset. |
| $N$ | Number of training samples. |
| $f_{\theta}$ | Feature extractor, parameterized by $\theta$. |
| $\mathcal{A}$ | Activation space. |
| $g_{\mathcal{W}}$ | Classifier (mapping activations to logits), parameterized by $\mathcal{W}$. |
| $\mathcal{Z}$ | Logit space. |
| $\mathbf{a}$ | Activation vector (output of $f_{\theta}(x)$). |
| $a_j$ | The $j$-th element of the activation vector $\mathbf{a}$. |
| $\mathbf{z}$ | Logit vector (output of $g_{\mathcal{W}}(\mathbf{a})$). |
| $\mathcal{L}$ | Loss function (e.g., cross-entropy). |
| $S(x)$ | OOD scoring function. |
| $\tau$ | Threshold for classifying an input as ID or OOD. |
| $x$ | Input image. |
| $x[c, h, w]$ | Channel value of input image $x$ at position $(c, h, w)$. |
| $H$ | Height of the input image. |
| $W$ | Width of the input image. |
| $AT(x, c, h, w)$ | Attribution function, assigning a score to each channel value of input $x$. |
| $o$ | Percent of channel values to perturb. |
| $R$ | Set of channel value indices with lowest absolute attribution scores. |
| $y_{\text{pred}}$ | Predicted class index. |
| $\varepsilon$ | Perturbation magnitude. |
| $x^{\varepsilon}$ | Perturbed input image. |
| $\mathbf{a}^{\varepsilon}$ | Activation vector of the perturbed input $x^{\varepsilon}$. |
| $\mathbf{a}^{\text{shift}}$ | Activation shift vector (absolute element-wise difference between $\mathbf{a}$ and $\mathbf{a}^{\varepsilon}$). |
| $k_1, k_2$ | Number of highest-magnitude activations considered for $Q$ and $C_o$, respectively. |
| $\text{argsort}(\mathbf{v})$ | Same as $\text{argsort}(\mathbf{v}, \text{ desc} = \text{True})$. |
|  | Returns the indices that would sort the vector $\mathbf{v}$ in descending order. |
| $\text{max}_k(\mathbf{v})$ | Returns the $k^{\text{th}}$ maximum element of vector $\mathbf{v}$. |
| $\mathbf{i}_1, \mathbf{i}_2$ | Index sets: $\mathbf{i}_1 = \text{argsort}(\mathbf{a}, \text{ desc} = \text{True})[: k_1]$, $\mathbf{i}_2 = \text{argsort}(\mathbf{a}, \text{ desc} = \text{True})[: k_2]$ |
| $Q$ | Sum of activation shifts for the top-$k_1$ activations. |
| $C_o$ | Correction term: sum of top-$k_2$ perturbed activations. |
| $\lambda$ | Weighting factor for $Q$ in the $Q'$ calculation. |
| $Q'$ | Estimated OOD likelihood. |
| $n_{\text{val}}$ | Number of ID validation samples. |
| $\bar{Q}_s$ | Set of $Q'$ values on the ID validation samples. |
| $F_{Q'}(Q')$ | Empirical cumulative distribution function (eCDF) of $Q'$ values. |
| $p_{\text{min}}, p_{\text{max}}$ | Minimum and maximum percentile thresholds. |
| $p_r$ | Raw ID likelihood from eCDF |
| $p$ | Adjusted percentile threshold |
| $P_p(\mathbf{a})$ | The $p$-th percentile value of all elements in $\mathbf{a}$ |
| $r$ | Scaling factor. |
| $\mathbf{a}_{\text{scaled}}$ | Scaled activation vector (AdaSCALE-A). |
| $\mathbf{z}_{\text{scaled}}$ | Scaled logit vector (AdaSCALE-L). |
| $\text{ReLU}(a_j)$ | Rectified Linear Unit activation function: $\text{ReLU}(a_j) = \max(0, a_j)$. |

## B  Algorithm

The algorithm for computing adaptive scaling factor $r$ is provided in Algorithm 1.

---

**Algorithm 1** Computing the Adaptive Scaling Factor

---

**Input:** Input sample $x$, perturbation magnitude $\varepsilon$, model $f_\theta$, hyperparameters $\lambda$, $k_1$, $k_2$, $p_{\min}$, $p_{\max}$, $\varepsilon$, $o$, precomputed empirical CDF $F_{Q'}$
**Output:** Scaling factor $r$
1:   // Extract features and compute activation shifts
2:   $\mathbf{a} \leftarrow f_\theta(x)$                      {Original activation}
3:   $\nabla_x z_c \leftarrow \frac{\partial g_W(f_\theta(x))_c}{\partial x}$    {Gradient for predicted class $c$}
4:   $R \leftarrow o\%$ of channel values with *lowest* $|\nabla_x z_c|$
5:   $x^\varepsilon \leftarrow x + \varepsilon \cdot \text{sign}(\nabla_x z_c) \cdot \mathbb{1}_R$ {Perturb selected regions}
6:   $\mathbf{a}^\varepsilon \leftarrow f_\theta(x^\varepsilon)$                    {Perturbed activation}
7:   $\mathbf{a}^{\text{shift}} \leftarrow |\mathbf{a}^\varepsilon - \mathbf{a}|$          {Compute activation shift}
8:   // Compute OOD likelihood estimate
9:   $\mathbf{i}_1 \leftarrow \text{argsort}(\mathbf{a}, \ \text{desc} = \text{True})[: k_1]$
10:   $Q \leftarrow \sum_{i \in \mathbf{i}_1} a_i^{\text{shift}}$          {Shift in top activations}
11:   $\mathbf{i}_2 \leftarrow \text{argsort}(\mathbf{a}, \ \text{desc} = \text{True})[: k_2]$
12:   $C_o \leftarrow \sum_{i \in \mathbf{i}_2} \text{ReLU}(a_i^\varepsilon)$        {Correction term}
13:   $Q' \leftarrow \lambda \cdot Q + C_o$       {OOD likelihood estimate}
14:   // Compute adaptive percentile
15:   $p_r \leftarrow (1 - F_{Q'}(Q'))$    {raw ID likelihood from eCDF}
16:   $p \leftarrow p_{\min} + p_r \cdot (p_{\max} - p_{\min})$    {Adjusted percentile}
17:   // Compute scaling factor
18:   $P_p(\mathbf{a}) \leftarrow$ the $p$-th percentile value of all elements in $\mathbf{a}$
19:   $r \leftarrow \sum_j a_j / \sum_{a_j > P_p(\mathbf{a})} a_j$      {Final scaling factor}
20:   **return** $r$

---

## C  Additional studies

### C.1  Additional observation in activation space.

Figure 8 shows perturbed activations $\mathbf{a}^\varepsilon$ are, on average, higher for ID samples than for OOD samples.

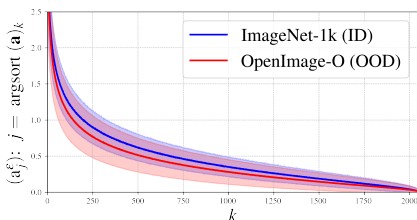

Figure 8: Perturbed activation magnitudes comparison between ID and OOD samples. ID samples consistently maintain higher average activation values in comparison to OOD samples.

### C.2  Sensitivity study of $\varepsilon$

The sensitivity study of $\varepsilon$ presented at Table 10 suggests the optimal value of $\varepsilon$ to be around 0.5.

Table 10: Sensitivity study of $\varepsilon$ with ResNet-50 model on ImageNet-1k benchmark.

| $\varepsilon$ | Near-OOD | | Far-OOD | |
|---|---|---|---|---|
| | FPR@95 ↓ | AUROC ↑ | FPR@95 ↓ | AUROC ↑ |
| 0.1 | 63.76 | 77.50 | 19.26 | 95.85 |
| 0.5 | **58.97** | **78.98** | **17.84** | **96.14** |
| 1.0 | 61.60 | 76.96 | 19.31 | 95.84 |

## C.3 Image Perturbation.

We present the complete results of image perturbation study (FPR@95 ↓ / AUROC ↑) in Table 7.

Table 11: Image perturbation study with ResNet-50 model on ImageNet-1k benchmark.

| Pixel type | $o\%$ | OOD Detection | | FS-OOD Detection | |
|---|---|---|---|---|---|
| | | Near-OOD | Far-OOD | Near-OOD | Far-OOD |
| Random | 1% | 61.73 / 78.15 | 19.44 / 95.74 | 83.19 / 48.59 | 53.45 / 74.10 |
| | 5% | 59.97 / 78.67 | 18.14 / 96.06 | 81.92 / 49.19 | 52.35 / 74.84 |
| | 10% | 60.27 / 78.02 | 18.45 / 96.00 | 82.07 / 48.62 | 52.84 / 74.70 |
| | 50% | 62.81 / 76.27 | 19.95 / 95.70 | 83.40 / 46.55 | 54.94 / 73.42 |
| Trivial | 1% | 61.77 / 78.29 | 19.28 / 95.77 | 82.92 / 48.86 | 53.34 / 74.24 |
| | 5% | **58.97 / 78.98** | **17.84 / 96.14** | **81.52 / 49.35** | **52.33 / 74.89** |
| | 10% | 60.24 / 78.17 | 17.94 / 96.08 | 82.19 / 48.59 | 52.59 / 74.77 |
| | 50% | 66.43 / 74.10 | 21.58 / 95.29 | 85.26 / 44.86 | 56.56 / 72.82 |
| Salient | 1% | 69.43 / 75.13 | 22.62 / 95.17 | 85.59 / 48.32 | 53.76 / 75.23 |
| | 5% | 67.31 / 75.78 | 21.24 / 95.44 | 85.07 / 48.40 | 53.26 / 75.63 |
| | 10% | 65.65 / 76.20 | 20.39 / 95.62 | 84.36 / 48.33 | 53.17 / 75.55 |
| | 50% | 64.54 / 75.39 | 20.48 / 95.61 | 83.78 / 47.07 | 54.11 / 74.54 |
| All | 100% | 67.37 / 73.08 | 22.93 / 95.01 | 85.66 / 44.00 | 57.80 / 72.19 |

## C.4 ISH regularization:

Apart from enhancing the prior postprocessor ASH [3], SCALE [81] introduces a training regularization to emphasize samples with more distinct ID characteristics. We assess the performance (FPR@95 ↓ / AUROC ↑) of each method in ResNet-50 and ResNet-101 model following this regularization in Table 12. The results indicate that AdaSCALE maintains a substantial advantage, surpassing the second-best method, SCALE, by **12.56%/5.82%** and **20.46%/1.21%** in FPR@95 / AUROC for near- and far-OOD detection in ResNet-50, respectively. Moreover, AdaSCALE demonstrates superior performance beyond conventional OOD detection, with corresponding improvements of **4.10%/5.87%** and **9.70%/1.12%** in full-spectrum setting. Furthermore, ISH regularization further amplifies the performance gap between AdaSCALE-A and OptFS, enhancing the near-OOD detection improvement from 12.96% / 6.44% to **15.18%/14.83%**. These findings also generalize to ResNet-101 network.

Table 12: OOD detection results on ImageNet-1k benchmark with ISH [4] regularization.

| Method | OOD Detection | | FS-OOD Detection | |
|---|---|---|---|---|
| | Near-OOD | Far-OOD | Near-OOD | Far-OOD |
| | ResNet-50 | | | |
| MSP | 74.07 / 62.16 | 51.13 / 84.64 | 87.52 / 40.36 | 74.41 / 61.52 |
| MLS | 74.38 / 66.43 | 41.57 / 88.90 | 88.89 / 39.49 | 71.53 / 61.69 |
| EBO | 74.68 / 66.46 | 41.85 / 88.83 | 89.05 / 39.18 | 71.77 / 61.11 |
| ReAct | 71.98 / 70.81 | 28.76 / 93.49 | 87.78 / 43.88 | 61.87 / 71.64 |
| ASH | 67.99 / 73.46 | 23.88 / 94.67 | 85.74 / 45.29 | 57.81 / 72.81 |
| SCALE | 65.68 / 76.41 | 20.77 / 95.62 | 84.31 / 48.40 | 54.48 / 74.79 |
| BFAct | 71.59 / 70.85 | 28.38 / 93.50 | 87.51 / 43.95 | 61.39 / 71.43 |
| LTS | 66.32 / 75.03 | 22.07 / 95.28 | 85.08 / 46.16 | 57.11 / 73.06 |
| OptFS | 67.71 / 73.03 | 24.65 / 94.18 | 85.38 / 45.91 | 57.09 / 72.95 |
| **AdaSCALE-A** | 57.43 / **80.86** | **16.52 / 96.46** | **80.85 / 51.24** | 51.57 / 75.63 |
| **AdaSCALE-L** | **56.83** / 80.81 | 17.62 / 96.22 | 80.97 / 50.59 | 53.43 / 74.62 |
| | ResNet-101 | | | |
| MSP | 71.39 / 68.31 | 51.00 / 84.81 | 85.70 / 45.72 | 73.69 / 62.79 |
| MLS | 72.32 / 71.94 | 41.04 / 88.99 | 87.39 / 44.88 | 69.94 / 63.23 |
| EBO | 72.78 / 71.92 | 41.45 / 88.87 | 87.64 / 44.66 | 70.23 / 62.65 |
| ReAct | 67.74 / 75.74 | 28.53 / 93.47 | 85.40 / 48.83 | 60.50 / 72.18 |
| ASH | 66.03 / 77.79 | 25.21 / 94.43 | 83.95 / 50.57 | 56.91 / 73.37 |
| SCALE | 64.30 / 78.98 | 23.09 / 94.95 | 83.14 / 51.05 | 55.88 / 73.27 |
| BFAct | 67.53 / 75.86 | 28.32 / 93.40 | 85.11 / 48.96 | 60.12 / 71.83 |
| LTS | 66.32 / 75.03 | 22.07 / 95.28 | 85.08 / 46.16 | 57.11 / 73.06 |
| OptFS | 67.71 / 73.03 | 24.65 / 94.18 | 85.38 / 45.91 | 57.09 / 72.95 |
| **AdaSCALE-A** | 54.66 / **83.52** | **16.81 / 96.32** | **78.52 / 55.19** | 49.92 / 76.20 |
| **AdaSCALE-L** | **53.91** / 83.49 | 17.55 / 96.15 | 78.63 / 54.60 | 51.47 / 75.47 |

# D Hyperparameters

All hyperparameters are determined with respect to the AUROC metric using automatic parameter search of OpenOOD [65, 80]. Although AdaSCALE may appear to require many hyperparameters, our findings indicate that setting $(\lambda, k_1, k_2, o, \epsilon)$ to $(10, 1\%, 5\%, 5\%, 0.5)$ consistently yields near-optimal performance across all setups. Consequently, it can be inferred that only the hyperparameters $p_{\min}$ and $p_{\max}$ need to be appropriately tuned for any new architecture for near-optimal performance. We present final hyperparameter values of AdaSCALE-A and AdaSCALE-L in Table 13 and Table 14.

Table 13: Hyperparameters used for each dataset and network for AdaSCALE-A.

| Dataset | Network | Hyperparameters | | | | | | |
|---|---|---|---|---|---|---|---|---|
| | | $p_{\min}$ | $p_{\max}$ | $\lambda$ | $k_1$ | $k_2$ | $o$ | $\epsilon$ |
| CIFAR-10 | WideResNet-28-10 | 60 | 95 | 10 | 1% | 80% | 5% | 0.5 |
| | DenseNet-101 | 65 | 90 | 10 | 1% | 10% | 5% | 0.5 |
| CIFAR-100 | WideResNet-28-10 | 60 | 85 | 10 | 1% | 80% | 5% | 0.5 |
| | DenseNet-101 | 70 | 80 | 10 | 1% | 100% | 5% | 0.5 |
| ImageNet-1k | ResNet-50 | 80 | 85 | 10 | 1% | 5% | 5% | 0.5 |
| | ResNet-101 | 80 | 85 | 10 | 1% | 5% | 5% | 0.5 |
| | RegNet-Y-16 | 60 | 90 | 10 | 1% | 50% | 5% | 0.5 |
| | ResNeXt-50 | 80 | 85 | 10 | 1% | 5% | 5% | 0.5 |
| | DenseNet-201 | 90 | 95 | 10 | 1% | 10% | 5% | 0.5 |
| | EfficientNetV2-L | 60 | 99 | 10 | 1% | 20% | 5% | 0.5 |
| | Vit-B-16 | 60 | 85 | 10 | 1% | 100% | 5% | 0.5 |
| | Swin-B | 90 | 99 | 10 | 1% | 5% | 5% | 0.5 |

Table 14: Hyperparameters used for each dataset and network for AdaSCALE-L.

| Dataset | Network | Hyperparameters | | | | | | |
|---|---|---|---|---|---|---|---|---|
| | | $p_{\min}$ | $p_{\max}$ | $\lambda$ | $k_1$ | $k_2$ | $o$ | $\epsilon$ |
| CIFAR-10 | WideResNet-28-10 | 60 | 85 | 10 | 1% | 80% | 5% | 0.5 |
| | DenseNet-101 | 70 | 85 | 10 | 1% | 10% | 5% | 0.5 |
| CIFAR-100 | WideResNet-28-10 | 60 | 80 | 10 | 1% | 80% | 5% | 0.5 |
| | DenseNet-101 | 65 | 75 | 10 | 1% | 50% | 5% | 0.5 |
| ImageNet-1k | ResNet-50 | 80 | 85 | 10 | 1% | 5% | 5% | 0.5 |
| | ResNet-101 | 70 | 80 | 10 | 1% | 5% | 5% | 0.5 |
| | RegNet-Y-16 | 60 | 85 | 10 | 1% | 5% | 5% | 0.5 |
| | ResNeXt-50 | 70 | 80 | 10 | 1% | 5% | 5% | 0.5 |
| | DenseNet-201 | 90 | 95 | 10 | 1% | 10% | 5% | 0.5 |
| | EfficientNetV2-L | 60 | 99 | 10 | 1% | 5% | 5% | 0.5 |
| | Vit-B-16 | 75 | 85 | 10 | 1% | 100% | 5% | 0.5 |
| | Swin-B | 90 | 99 | 10 | 1% | 100% | 5% | 0.5 |

# E   CIFAR-results

## E.1   WRN-28-10

Table 15: Far-OOD detection results (FPR@95↓ / AUROC↑) on CIFAR-10 and CIFAR-100 benchmarks using the WRN-28-10 network, averaged over 3 trials. The overall average performance is reported. The best results are **bold**, and the second-best results are underlined.

| | *CIFAR-10 benchmark* | | | | |
| --- | --- | --- | --- | --- | --- |
| Method | **MNIST** | **SVHN** | **Textures** | **Places365** | **Average** |
| MSP | 17.02 / 94.61 | 21.71 / 92.96 | 60.50 / 88.06 | 42.27 / 90.04 | 35.38 / 91.42 |
| MLS | 13.01 / 96.76 | 30.35 / 93.06 | 76.12 / 86.65 | 52.56 / 90.46 | 43.01 / 91.73 |
| EBO | **12.93** / **96.93** | 30.35 / 93.12 | 76.15 / 86.68 | 52.57 / 90.56 | 43.00 / 91.82 |
| ReAct | 15.50 / 96.30 | 34.01 / 92.47 | 57.76 / 88.77 | 57.33 / 89.66 | 41.15 / 91.80 |
| ASH | 50.11 / 88.80 | 89.90 / 74.76 | 95.07 / 72.91 | 92.22 / 70.06 | 81.82 / 76.63 |
| SCALE | 13.24 / 96.70 | 32.21 / 92.88 | 75.76 / 86.77 | 55.81 / 90.09 | 44.26 / 91.61 |
| BFAct | 25.79 / 94.64 | 43.08 / 91.10 | **57.16** / **88.80** | 61.00 / 88.32 | 46.75 / 90.71 |
| LTS | 14.04 / 96.60 | 39.85 / 92.21 | 76.85 / 86.43 | 63.13 / 89.19 | 48.47 / 91.11 |
| OptFS | 25.68 / 94.83 | 51.58 / 89.86 | 62.14 / 88.07 | 80.19 / 84.05 | 54.90 / 89.20 |
| **AdaSCALE-A** | 14.93 / 96.02 | **17.84** / **95.14** | 64.96 / 88.31 | **34.57** / **92.31** | **33.08** / **92.95** |
| **AdaSCALE-L** | 15.58 / 95.98 | 18.41 / 95.10 | 62.87 / 88.67 | 37.59 / 91.97 | 33.61 / 92.93 |
| | *CIFAR-100 benchmark* | | | | |
| | **MNIST** | **SVHN** | **Textures** | **Places365** | **Average** |
| MSP | 49.79 / 78.72 | 56.76 / 80.70 | 64.49 / 76.86 | 56.66 / 79.96 | 56.92 / 79.06 |
| MLS | 46.57 / 81.43 | 53.08 / 83.37 | 64.59 / 77.65 | 59.70 / 79.82 | 55.99 / 80.57 |
| EBO | 46.41 / 81.99 | 52.92 / 83.77 | 64.58 / 77.61 | 59.76 / 79.60 | 55.92 / 80.74 |
| ReAct | 49.92 / 81.07 | 40.66 / 86.49 | 52.42 / 80.81 | 60.35 / 79.72 | 50.84 / 82.03 |
| ASH | 44.06 / 85.55 | 41.48 / 87.50 | 61.78 / 81.65 | 80.45 / 71.83 | 56.94 / 81.63 |
| SCALE | 40.65 / 84.68 | 48.56 / 85.56 | 58.45 / 80.81 | 60.51 / 79.90 | 52.04 / 82.74 |
| BFAct | 61.59 / 77.47 | 34.74 / 88.50 | **47.30** / 83.38 | 64.49 / 78.47 | 52.03 / 81.96 |
| LTS | **36.27** / **87.38** | 45.41 / 87.23 | 53.90 / 83.18 | 62.62 / 79.64 | 49.55 / 84.36 |
| OptFS | 57.61 / 79.47 | 37.04 / 86.43 | 53.02 / 80.43 | 70.44 / 76.77 | 54.53 / 80.78 |
| **AdaSCALE-A** | 45.18 / 81.69 | 36.79 / 89.20 | 55.93 / 81.93 | **56.48** / 81.55 | 48.59 / 83.59 |
| **AdaSCALE-L** | 42.13 / 83.58 | **32.44** / **91.02** | 50.87 / **84.14** | 57.83 / 81.51 | **45.82** / **85.06** |

Table 16: Near-OOD detection results (FPR@95↓ / AUROC↑) on CIFAR-10 and CIFAR-100 benchmarks using the WRN-28-10 network, averaged over 3 trials. The overall average performance is reported. The best results are **bold**, and the second-best results are underlined.

| Method | *CIFAR-10 benchmark* | | *CIFAR-100 benchmark* | | **Average** |
| --- | --- | --- | --- | --- | --- |
| | **CIFAR-100** | **TIN** | **CIFAR-10** | **TIN** | |
| MSP | 54.13 / 88.28 | 42.94 / 89.93 | **56.83** / 80.42 | 48.82 / 83.39 | 50.68 / 85.51 |
| MLS | 67.10 / 87.72 | 55.91 / 89.90 | 58.99 / **80.98** | 49.27 / 84.01 | 57.82 / 85.65 |
| EBO | 67.04 / 87.77 | 55.88 / 89.97 | 58.97 / 80.93 | 49.39 / 83.95 | 57.82 / 85.66 |
| ReAct | 65.96 / 87.76 | 51.44 / 90.33 | 69.17 / 79.10 | 51.56 / 83.77 | 59.53 / 85.24 |
| ASH | 91.33 / 70.72 | 90.77 / 73.28 | 85.25 / 69.96 | 78.31 / 74.55 | 86.42 / 72.13 |
| SCALE | 69.52 / 87.35 | 59.48 / 89.53 | 61.30 / 80.35 | 51.25 / 83.65 | 60.39 / 85.22 |
| BFAct | 66.31 / 86.94 | 57.12 / 89.69 | 78.90 / 74.98 | 59.04 / 82.25 | 65.34 / 83.47 |
| LTS | 74.28 / 86.38 | 66.01 / 88.56 | 64.17 / 79.57 | 54.24 / 83.09 | 64.68 / 84.40 |
| OptFS | 76.36 / 84.05 | 66.73 / 86.56 | 85.40 / 75.55 | 64.11 / 80.99 | 73.15 / 79.83 |
| **AdaSCALE-A** | **50.60** / **89.40** | **42.80** / **91.13** | 62.21 / 79.99 | **47.11** / **84.98** | **50.68** / **86.38** |
| **AdaSCALE-L** | 53.98 / 89.01 | 45.95 / 90.77 | 65.41 / 79.27 | 48.74 / 84.75 | 53.52 / 85.95 |

 **E.2 DenseNet-101**

Table 17: Far-OOD detection results (FPR@95↓ / AUROC↑) on CIFAR-10 and CIFAR-100 benchmarks using the DenseNet-101 network, averaged over 3 trials. The overall average performance is reported. The best results are **bold**, and the second-best results are underlined.

| | *CIFAR-10 benchmark* | | | | |
|---|---|---|---|---|---|
| Method | **MNIST** | **SVHN** | **Textures** | **Places365** | **Average** |
| MSP | 17.91 / 94.22 | 32.04 / 90.38 | **46.80** / 87.53 | 37.59 / 89.24 | 33.59 / 90.34 |
| MLS | 10.02 / 97.58 | 31.25 / 92.59 | 64.43 / 85.58 | 39.19 / 90.74 | 36.22 / 91.62 |
| EBO | 9.74 / 97.76 | 31.23 / 92.69 | 64.46 / 85.48 | 39.17 / 90.81 | 36.15 / 91.68 |
| ReAct | 12.60 / 97.24 | 34.79 / 92.02 | 50.41 / 88.21 | **36.12** / **91.35** | 33.48 / 92.20 |
| ASH | 9.40 / **98.12** | 39.42 / 91.25 | 70.95 / 85.39 | 57.90 / 85.48 | 44.42 / 90.06 |
| SCALE | 9.04 / 97.88 | 26.99 / 93.54 | 61.52 / 86.77 | 39.57 / 90.76 | 34.28 / 92.24 |
| BFAct | 23.59 / 94.96 | 42.49 / 89.00 | 53.74 / 87.38 | 37.53 / 91.09 | 39.34 / 90.61 |
| LTS | **8.92** / 97.97 | 27.16 / 93.59 | 59.07 / 87.09 | 39.47 / 90.81 | 33.65 / 92.37 |
| OptFS | 9.74 / 97.88 | 41.20 / 90.71 | 51.35 / **88.48** | 59.47 / 86.03 | 40.44 / 90.77 |
| **AdaSCALE-A** | 12.42 / 96.85 | **25.04** / **94.05** | 58.28 / 87.35 | 36.77 / 91.20 | **33.13** / 92.36 |
| **AdaSCALE-L** | 10.92 / 97.44 | 26.43 / 93.87 | 58.59 / 87.19 | 37.03 / 91.25 | 33.24 / **92.44** |
| | *CIFAR-100 benchmark* | | | | |
| MSP | 65.65 / 72.43 | 63.81 / 76.52 | 75.34 / 72.19 | 61.36 / 77.16 | 66.54 / 74.57 |
| MLS | 58.69 / 78.55 | 57.12 / 79.43 | 79.05 / 72.68 | 62.72 / 78.38 | 64.39 / 77.26 |
| EBO | 58.58 / 78.98 | 56.76 / 79.19 | 79.09 / 72.44 | 62.86 / 78.08 | 64.32 / 77.17 |
| ReAct | 62.71 / 76.37 | 48.48 / 81.64 | **64.65** / 78.62 | **59.00** / **78.89** | 58.71 / 78.88 |
| ASH | **40.69** / **88.57** | 48.03 / **86.24** | 65.24 / **83.29** | 73.29 / 71.88 | **56.81** / **82.49** |
| SCALE | 56.92 / 79.53 | 53.81 / 80.96 | 76.10 / 74.47 | 62.49 / 78.51 | 62.33 / 78.37 |
| BFAct | 73.83 / 67.19 | 60.01 / 75.85 | 69.29 / 76.41 | 68.15 / 73.73 | 67.82 / 73.29 |
| LTS | 55.33 / 80.58 | 51.33 / 82.04 | 73.06 / 75.89 | 62.58 / 78.36 | 60.58 / 79.22 |
| OptFS | 64.24 / 75.24 | 59.81 / 76.46 | 66.15 / 77.50 | 73.47 / 69.76 | 65.92 / 74.74 |
| **AdaSCALE-A** | 62.51 / 74.96 | 46.29 / 84.31 | 71.40 / 76.59 | 61.70 / 78.86 | 60.47 / 78.68 |
| **AdaSCALE-L** | 61.33 / 75.73 | **43.97** / 85.30 | 69.31 / 77.71 | 61.97 / 78.69 | 59.15 / 79.36 |

Table 18: Near-OOD detection results (FPR@95↓ / AUROC↑) on CIFAR-10 and CIFAR-100 benchmarks using the DenseNet-101 network, averaged over 3 trials. The overall average performance is reported. The best results are **bold**, and the second-best results are underlined.

| Method | *CIFAR-10 benchmark* | | *CIFAR-100 benchmark* | | Average |
|---|---|---|---|---|---|
| | CIFAR-100 | TIN | CIFAR-10 | TIN | |
| MSP | **40.13** / 88.45 | **35.50** / 89.61 | **59.94** / 77.53 | 56.96 / 79.57 | **48.13** / 83.79 |
| MLS | 45.14 / 88.85 | 38.01 / 90.85 | 63.61 / **78.26** | 57.09 / 81.75 | 50.96 / 84.93 |
| EBO | 45.19 / 88.85 | 38.05 / 90.90 | 63.90 / 77.94 | 57.53 / 81.58 | 51.17 / 84.82 |
| ReAct | 44.34 / 89.19 | 37.10 / 91.08 | 70.77 / 75.06 | 61.30 / 80.40 | 53.38 / 83.93 |
| ASH | 67.78 / 82.68 | 62.54 / 85.18 | 81.65 / 65.84 | 78.66 / 70.01 | 72.66 / 75.93 |
| SCALE | 45.25 / 88.92 | 37.76 / 91.01 | 64.20 / 78.13 | 56.96 / 81.88 | 51.04 / 84.99 |
| BFAct | 52.06 / 87.70 | 44.09 / 89.89 | 79.31 / 67.39 | 71.79 / 74.18 | 61.81 / 79.79 |
| LTS | 44.89 / 89.03 | 37.67 / 91.10 | 64.66 / 77.87 | 56.83 / 81.87 | 51.01 / 84.97 |
| OptFS | 60.63 / 85.29 | 55.55 / 86.96 | 82.97 / 64.99 | 74.73 / 70.55 | 68.47 / 76.95 |
| **AdaSCALE-A** | 43.29 / 89.37 | 35.57 / 91.32 | 65.51 / 78.00 | **54.49** / 82.44 | 49.72 / **85.28** |
| **AdaSCALE-L** | 43.19 / **89.40** | 35.70 / **91.37** | 66.09 / 77.79 | 54.57 / **82.47** | 49.89 / 85.26 |

# F ImageNet-1k results

## F.1 near-OOD detection

Table 19: Near-OOD detection results (FPR@95↓ / AUROC↑) on ImageNet-1k benchmark using ResNet-50 network. The best results are **bold**, and the second-best results are underlined.

| Method | SSB-Hard | NINCO | ImageNet-O | Average |
|---|---|---|---|---|
| MSP | 74.49 / 72.09 | 56.88 / 79.95 | 91.32 / 28.60 | 74.23 / 60.21 |
| MLS | 76.20 / 72.51 | 59.44 / 80.41 | 88.97 / 40.73 | 74.87 / 64.55 |
| EBO | 76.54 / 72.08 | 60.58 / 79.70 | 88.84 / 41.78 | 75.32 / 64.52 |
| REACT | 77.55 / 73.03 | 55.82 / 81.73 | 84.45 / 51.67 | 72.61 / 68.81 |
| ASH | 73.66 / 72.89 | 53.05 / 83.45 | 81.70 / 57.67 | 69.47 / 71.33 |
| SCALE | 67.72 / 77.35 | 51.80 / 85.37 | 83.77 / 59.89 | 67.76 / 74.20 |
| BFAct | 77.20 / 73.15 | 55.27 / 81.88 | 84.57 / 51.62 | 72.35 / 68.88 |
| LTS | 68.46 / 77.10 | 51.24 / 85.33 | 84.33 / 57.69 | 68.01 / 73.37 |
| OptFS | 78.32 / 71.01 | 52.09 / 82.51 | 78.56 / 59.40 | 69.66 / 70.97 |
| **AdaSCALE-A** | **57.96 / 81.68** | **44.92 / 87.15** | **74.06 / 68.12** | **58.98 / 78.98** |
| **AdaSCALE-L** | 58.68 / 81.42 | 45.01 / 87.11 | 75.83 / 67.33 | 59.84 / 78.62 |

Table 20: Near-OOD detection results (FPR@95↓ / AUROC↑) on ImageNet-1k benchmark using ResNet-101 network. The best results are **bold**, and the second-best results are underlined.

| Method | SSB-Hard | NINCO | ImageNet-O | Average |
|---|---|---|---|---|
| MSP | 73.20 / 72.57 | 55.27 / 80.61 | 87.42 / 48.57 | 71.96 / 67.25 |
| MLS | 74.68 / 74.37 | 55.65 / 82.29 | 85.81 / 57.89 | 72.05 / 71.51 |
| EBO | 74.96 / 74.12 | 56.33 / 81.79 | 85.66 / 58.72 | 72.32 / 71.54 |
| REACT | 75.96 / 74.43 | 52.58 / 83.27 | 75.67 / 67.31 | 68.07 / 75.00 |
| ASH | 72.48 / 74.23 | 49.41 / 84.62 | 73.84 / 70.98 | 65.24 / 76.61 |
| SCALE | 68.47 / 77.10 | 49.03 / 86.20 | 74.09 / 72.50 | 63.87 / 78.60 |
| BFAct | 75.48 / 74.74 | 52.23 / 83.37 | 76.16 / 67.37 | 67.96 / 75.16 |
| OptFS | 76.55 / 72.29 | 50.89 / 83.35 | 68.94 / 71.85 | 65.46 / 75.83 |
| **AdaSCALE-A** | **61.00** / 80.29 | **46.70 / 86.99** | 62.05 / 78.27 | 56.59 / 81.85 |
| **AdaSCALE-L** | 61.05 / **80.41** | 47.77 / 86.84 | **60.40 / 78.35** | **56.41 / 81.86** |

Table 21: Near-OOD detection results (FPR@95↓ / AUROC↑) on ImageNet-1k benchmark using RegNet-Y-16 network. The best results are **bold**, and the second-best results are underlined.

| Method | SSB-Hard | NINCO | ImageNet-O | Average |
|---|---|---|---|---|
| MSP | 65.35 / 78.28 | 48.48 / 86.85 | 72.82 / 77.09 | 62.22 / 80.74 |
| MLS | 62.48 / 84.83 | 42.76 / 91.56 | 83.60 / 77.58 | 62.94 / 84.66 |
| EBO | 62.10 / 85.28 | 42.49 / 91.67 | 83.82 / 77.33 | 62.80 / 84.76 |
| REACT | 73.02 / 73.17 | 59.81 / 80.91 | 79.37 / 72.02 | 70.73 / 75.37 |
| ASH | 80.58 / 67.70 | 77.23 / 71.42 | 89.71 / 64.30 | 82.51 / 67.81 |
| SCALE | 66.98 / 82.35 | 49.84 / 89.93 | 84.44 / 76.43 | 67.09 / 82.90 |
| BFAct | 79.40 / 64.39 | 73.98 / 70.35 | 82.76 / 63.54 | 78.72 / 66.09 |
| LTS | 69.52 / 79.78 | 55.38 / 87.71 | 84.55 / 74.78 | 69.82 / 80.75 |
| OptFS | 79.59 / 69.47 | 63.97 / 80.36 | 77.03 / 75.79 | 73.53 / 75.21 |
| **AdaSCALE-A** | **54.50** / **87.21** | **31.50** / **93.50** | **57.75** / **86.83** | **47.91** / **89.18** |
| **AdaSCALE-L** | 62.61 / 84.60 | 47.84 / 90.13 | 57.94 / 86.61 | 56.13 / 87.11 |

Table 22: Near-OOD detection results (FPR@95↓ / AUROC↑) on ImageNet-1k benchmark using ResNeXt-50 network. The best results are **bold**, and the second-best results are underlined.

| Method | SSB-Hard | NINCO | ImageNet-O | Average |
|---|---|---|---|---|
| MSP | 73.04 / 73.28 | 57.90 / 80.86 | 88.81 / 49.43 | 73.25 / 67.86 |
| MLS | 74.68 / 75.06 | 60.79 / 81.91 | 86.87 / 57.87 | 74.11 / 71.61 |
| EBO | 74.90 / 74.89 | 60.96 / 81.44 | 86.76 / 58.49 | 74.21 / 71.61 |
| REACT | 75.54 / 74.51 | 57.29 / 82.50 | 80.03 / 65.37 | 70.95 / 74.13 |
| ASH | 70.72 / 76.64 | 58.40 / 83.49 | 83.84 / 65.63 | 70.99 / 75.25 |
| SCALE | 67.77 / 79.73 | 56.87 / 85.39 | 87.15 / 63.48 | 70.60 / 76.20 |
| BFAct | 75.36 / 74.65 | 57.65 / 82.46 | 79.86 / 65.30 | 70.96 / 74.14 |
| LTS | 68.26 / 79.36 | 56.35 / 85.39 | 86.22 / 63.85 | 70.28 / 76.20 |
| OptFS | 75.62 / 73.82 | 57.07 / 82.37 | **75.13** / 68.33 | 69.27 / 74.84 |
| **AdaSCALE-A** | **61.03** / **81.86** | 50.80 / **86.54** | 80.57 / 71.48 | 64.13 / 79.96 |
| **AdaSCALE-L** | 61.57 / 81.11 | **48.78** / 86.40 | 75.88 / **73.02** | **62.08** / **80.18** |

Table 23: Near-OOD detection results (FPR@95↓ / AUROC↑) on ImageNet-1k benchmark using DenseNet-201 network. The best results are **bold**, and the second-best results are underlined.

| Method | SSB-Hard | NINCO | ImageNet-O | Average |
|---|---|---|---|---|
| MSP | 74.43 / 72.23 | 56.69 / 80.85 | 89.18 / 48.80 | 73.44 / 67.29 |
| MLS | 76.62 / 72.48 | 60.14 / 80.91 | 89.78 / 53.34 | 75.51 / 68.91 |
| EBO | 76.92 / 72.00 | 60.88 / 80.01 | 89.75 / 54.03 | 75.85 / 68.68 |
| ReAct | 78.62 / 70.93 | 57.51 / 81.19 | 73.78 / 68.83 | 69.97 / 73.65 |
| ASH | 78.80 / 68.71 | 63.84 / 79.45 | 80.07 / 68.19 | 74.24 / 72.12 |
| SCALE | 73.64 / 74.43 | 56.90 / 83.80 | 84.14 / 62.92 | 71.56 / 73.72 |
| BFAct | 81.57 / 67.52 | 65.10 / 77.38 | 66.93 / 72.93 | 71.20 / 72.61 |
| LTS | 73.46 / 74.36 | 57.54 / 83.79 | 82.87 / 65.52 | 71.29 / 74.56 |
| OptFS | 82.76 / 65.38 | 63.26 / 78.12 | 69.21 / 72.79 | 71.74 / 72.10 |
| **AdaSCALE-A** | **68.46** / **77.10** | **56.66** / **84.32** | 58.72 / 77.55 | **61.28** / 79.66 |
| **AdaSCALE-L** | 68.97 / 76.85 | 57.96 / 83.92 | **58.30** / **79.41** | 61.75 / **80.06** |

Table 24: Near-OOD detection results (FPR@95↓ / AUROC↑) on ImageNet-1k benchmark using EfficientNetV2-L network. The best results are **bold**, and the second-best results are underlined.

| Method | SSB-Hard | NINCO | ImageNet-O | Average |
|---|---|---|---|---|
| MSP | 81.28 / 75.03 | 57.97 / 86.70 | 78.26 / 80.53 | 72.51 / 80.76 |
| MLS | 84.74 / 73.50 | 72.88 / 84.83 | 86.71 / 79.32 | 81.44 / 79.22 |
| EBO | 85.27 / 71.58 | 75.81 / 82.07 | 87.49 / 77.81 | 82.86 / 77.15 |
| ReAct | 74.29 / 70.63 | 71.93 / 70.92 | 70.86 / 72.63 | 72.36 / 71.39 |
| ASH | 94.82 / 46.73 | 96.44 / 37.79 | 93.30 / 49.81 | 94.85 / 44.78 |
| SCALE | 90.16 / 57.07 | 89.93 / 59.69 | 89.03 / 63.60 | 89.70 / 60.12 |
| BFAct | 75.36 / 63.66 | 77.03 / 59.56 | 74.19 / 64.18 | 75.53 / 62.46 |
| LTS | 88.43 / 68.29 | 86.68 / 75.87 | 86.78 / 76.73 | 87.30 / 73.63 |
| OptFS | 74.68 / 73.83 | 70.24 / 76.18 | 71.94 / 75.86 | 72.29 / 75.29 |
| **AdaSCALE-A** | 60.84 / 83.48 | **47.45** / **89.47** | 53.04 / **87.87** | **53.78** / **86.94** |
| **AdaSCALE-L** | **53.56** / **85.00** | 58.55 / 84.75 | **52.72** / 87.58 | 54.95 / 85.77 |

Table 25: Near-OOD detection results (FPR@95↓ / AUROC↑) on ImageNet-1k benchmark using ViT-B-16 network. The best results are **bold**, and the second-best results are underlined.

| Method | SSB-Hard | NINCO | ImageNet-O | Average |
|---|---|---|---|---|
| MSP | 86.41 / **68.94** | 77.28 / 78.11 | 96.48 / 58.81 | 86.72 / 68.62 |
| MLS | 91.52 / 64.20 | 92.98 / 72.40 | 96.84 / 54.33 | 93.78 / 63.64 |
| EBO | 92.24 / 58.80 | 94.14 / 66.02 | 96.74 / 52.74 | 94.37 / 59.19 |
| ReAct | 90.46 / 63.10 | 78.50 / 75.43 | 90.94 / 66.53 | 86.63 / 68.35 |
| ASH | 93.50 / 53.90 | 95.37 / 52.51 | 94.47 / 53.19 | 94.45 / 53.20 |
| SCALE | 92.37 / 56.55 | 94.62 / 61.52 | 96.44 / 50.47 | 94.48 / 56.18 |
| BFAct | 89.81 / 64.16 | 71.37 / 78.06 | 85.09 / 69.75 | 82.09 / 70.66 |
| LTS | 91.42 / 64.35 | 82.63 / 75.48 | 92.42 / 62.46 | 88.83 / 67.43 |
| OptFS | 87.98 / 66.30 | 64.24 / 80.46 | 77.43 / 71.43 | 76.55 / 72.73 |
| **AdaSCALE-A** | **85.89** / 66.57 | 61.92 / **80.47** | **67.81** / 72.37 | **71.87** / 73.14 |
| **AdaSCALE-L** | 86.19 / 66.25 | **61.79** / 80.42 | 67.99 / **73.01** | 71.99 / **73.23** |

Table 26: Near-OOD detection results (FPR@95↓ / AUROC↑) on ImageNet-1k benchmark using Swin-B network. The best results are **bold**, and the second-best results are underlined.

| Method | SSB-Hard | NINCO | ImageNet-O | Average |
|---|---|---|---|---|
| MSP | 86.47 / 71.30 | 77.95 / 78.50 | 96.90 / 59.65 | 87.11 / 69.82 |
| MLS | 94.05 / 65.04 | 93.38 / 71.75 | 96.97 / 57.26 | 94.80 / 64.68 |
| EBO | 94.66 / 58.96 | 94.59 / 64.02 | 96.75 / 56.40 | 95.34 / 59.79 |
| ReAct | 89.19 / 68.70 | 68.54 / 80.16 | 90.20 / 70.93 | 82.64 / 73.26 |
| ASH | 97.15 / 45.47 | 96.64 / 47.36 | 95.32 / 49.92 | 96.37 / 47.58 |
| SCALE | 90.84 / 56.53 | 87.86 / 62.49 | 87.16 / 65.38 | 88.62 / 61.47 |
| BFAct | 84.86 / 69.41 | 61.30 / 81.10 | **69.27** / **75.34** | **71.81** / **75.28** |
| LTS | 90.36 / 64.51 | 81.02 / 74.23 | 88.44 / 62.92 | 86.61 / 67.22 |
| OptFS | 88.68 / 68.43 | 66.36 / 80.27 | 75.38 / 73.49 | 76.81 / 74.06 |
| **AdaSCALE-A** | **80.10** / **70.46** | 64.67 / 81.10 | 75.46 / 71.87 | 73.41 / 74.48 |
| **AdaSCALE-L** | 80.12 / 70.06 | **63.68** / **81.35** | 74.87 / 72.34 | 72.89 / 74.58 |

## F.2 far-OOD detection

Table 27: Far-OOD detection results (FPR@95↓ / AUROC↑) on
ImageNet-1k benchmark using ResNet-50 network. The best results are
**bold**, and the second-best results are underlined.

| Method | iNaturalist | Textures | OpenImage-O | Places | Average |
|---|---|---|---|---|---|
| MSP | 43.34 / 88.41 | 60.87 / 82.43 | 50.13 / 84.86 | 58.26 / 80.55 | 53.15 / 84.06 |
| MLS | 30.61 / 91.17 | 46.17 / 88.39 | 37.88 / 89.17 | 55.62 / 84.05 | 42.57 / 88.19 |
| EBO | 31.30 / 90.63 | 45.77 / 88.70 | 38.09 / 89.06 | 55.73 / 83.97 | 42.72 / 88.09 |
| ReAct | 16.72 / 96.34 | 29.64 / 92.79 | 32.58 / 91.87 | 41.62 / 90.93 | 30.14 / 92.98 |
| ASH | 14.09 / 97.06 | 15.30 / 96.90 | 29.19 / 93.26 | 40.16 / 90.48 | 24.69 / 94.43 |
| SCALE | 9.50 / 98.02 | 11.90 / 97.63 | 28.18 / 93.95 | 36.18 / 91.96 | 21.44 / 95.39 |
| BFAct | 15.94 / 96.47 | 28.43 / 92.87 | 32.66 / 91.90 | 40.83 / 90.79 | 29.46 / 93.01 |
| LTS | 10.24 / 97.87 | 13.06 / 97.42 | 27.81 / 94.01 | 37.68 / 91.65 | 22.20 / 95.24 |
| OptFS | 15.88 / 96.65 | 16.60 / 96.10 | 29.94 / 92.53 | 40.24 / 90.20 | 25.66 / 93.87 |
| **AdaSCALE-A** | **7.61 / 98.31** | 10.57 / 97.88 | 20.67 / 95.62 | **32.60 / 92.74** | **17.86 / 96.14** |
| **AdaSCALE-L** | 7.78 / 98.29 | **10.33 / 97.92** | **20.61 / 95.62** | 32.97 / 92.63 | 17.92 / 96.12 |

Table 28: Far-OOD detection results (FPR@95↓ / AUROC↑) on
ImageNet-1k benchmark using ResNet-101 network. The best results are
**bold**, and the second-best results are underlined.

| Method | iNaturalist | Textures | OpenImage-O | Places | Average |
|---|---|---|---|---|---|
| MSP | 48.30 / 86.27 | 59.00 / 83.60 | 49.36 / 84.82 | 58.84 / 80.56 | 53.87 / 83.81 |
| MLS | 41.11 / 88.83 | 43.59 / 89.85 | 38.13 / 89.25 | 52.74 / 85.28 | 43.89 / 88.30 |
| EBO | 41.65 / 88.30 | 43.66 / 90.14 | 38.48 / 89.12 | 53.42 / 85.37 | 44.30 / 88.23 |
| ReAct | 19.86 / 95.66 | 26.94 / 93.78 | 30.18 / 92.54 | 42.58 / 90.41 | 29.89 / 93.10 |
| ASH | 19.90 / 95.68 | 13.94 / 97.32 | 27.76 / 93.63 | 43.11 / 89.59 | 26.18 / 94.06 |
| SCALE | 13.90 / 97.05 | 9.34 / 98.04 | 25.91 / 94.47 | 40.99 / 90.64 | 22.54 / 95.05 |
| BFAct | 19.60 / 95.69 | 25.79 / 93.79 | 30.18 / 92.55 | 42.14 / 90.13 | 29.43 / 93.04 |
| LTS | 15.07 / 96.83 | 10.33 / 97.89 | 25.51 / 94.52 | 41.40 / 90.53 | 23.07 / 94.94 |
| OptFS | 19.11 / 95.70 | 16.53 / 96.35 | 28.76 / 92.94 | 43.47 / 89.22 | 26.97 / 93.55 |
| **AdaSCALE-A** | **10.74 / 97.64** | **8.90 / 98.21** | 18.75 / 96.03 | **35.66 / 91.92** | **18.51 / 95.95** |
| **AdaSCALE-L** | 11.71 / 97.36 | 10.44 / 97.93 | **17.87 / 96.18** | 36.57 / 91.55 | 19.15 / 95.76 |

Table 29: Far-OOD detection results (FPR@95↓ / AUROC↑) on ImageNet-1k benchmark using RegNet-Y-16 network. The best results are **bold**, and the second-best results are underlined.

| Method | iNaturalist | Textures | OpenImage-O | Places | Average |
|---|---|---|---|---|---|
| MSP | 28.13 / 94.67 | 44.73 / 88.48 | 36.27 / 91.96 | 52.51 / 85.21 | 40.41 / 90.08 |
| MLS | 9.10 / 98.05 | 39.74 / 92.82 | 25.71 / 95.70 | 57.14 / 88.22 | 32.92 / 93.70 |
| EBO | 7.72 / 98.29 | 38.18 / 93.02 | 25.94 / 95.83 | 58.04 / 88.13 | 32.47 / 93.82 |
| ReAct | 21.24 / 94.14 | 41.20 / 87.25 | 43.46 / 89.20 | 74.92 / 74.10 | 45.20 / 86.17 |
| ASH | 48.89 / 87.39 | 45.75 / 88.79 | 70.98 / 82.52 | 72.99 / 77.06 | 59.65 / 83.94 |
| SCALE | 11.13 / 97.88 | 28.29 / 95.31 | 33.59 / 94.87 | 55.62 / 88.59 | 32.16 / 94.16 |
| BFAct | 37.88 / 86.24 | 54.87 / 77.64 | 62.53 / 79.59 | 79.46 / 65.39 | 58.69 / 77.22 |
| LTS | 14.29 / 97.52 | 25.21 / 95.72 | 43.38 / 93.53 | 57.08 / 87.51 | 34.99 / 93.57 |
| OptFS | 28.95 / 93.68 | 39.99 / 90.13 | 44.96 / 89.85 | 75.59 / 73.24 | 47.37 / 86.73 |
| **AdaSCALE-A** | **4.34** / **99.09** | 26.06 / 95.21 | **13.09** / **97.57** | **41.98** / **91.48** | 21.37 / 95.84 |
| **AdaSCALE-L** | 4.41 / 99.02 | **13.50** / **97.61** | 18.56 / 96.92 | 43.93 / 91.22 | **20.10** / **96.19** |

Table 30: Far-OOD detection results (FPR@95↓ / AUROC↑) on ImageNet-1k benchmark using ResNeXt-50 network. The best results are **bold**, and the second-best results are underlined.

| Method | iNaturalist | Textures | OpenImage-O | Places | Average |
|---|---|---|---|---|---|
| MSP | 43.56 / 88.04 | 62.23 / 82.13 | 48.06 / 85.65 | 58.42 / 81.02 | 53.07 / 84.21 |
| MLS | 32.96 / 90.93 | 51.58 / 87.39 | 37.33 / 89.80 | 57.76 / 83.77 | 44.91 / 87.97 |
| EBO | 33.42 / 90.54 | 51.73 / 87.56 | 37.79 / 89.72 | 57.56 / 83.62 | 45.12 / 87.86 |
| ReAct | 17.64 / 95.95 | 32.86 / 91.67 | 29.82 / 92.37 | 39.92 / 90.76 | 30.06 / 92.69 |
| ASH | 17.90 / 96.22 | 23.74 / 95.18 | 30.83 / 93.13 | 44.21 / 89.35 | 29.17 / 93.47 |
| SCALE | 15.66 / 96.75 | 27.75 / 94.94 | 31.43 / 93.41 | 47.62 / 89.08 | 30.62 / 93.54 |
| BFAct | 17.40 / 95.91 | 32.00 / 91.83 | 29.53 / 92.38 | 39.89 / 90.57 | 29.71 / 92.67 |
| LTS | 16.29 / 96.63 | 26.64 / 95.07 | 30.50 / 93.50 | 48.04 / 88.78 | 30.37 / 93.49 |
| OptFS | 17.20 / 96.12 | 23.11 / 94.69 | 29.59 / 92.75 | 40.24 / 90.05 | 27.54 / 93.40 |
| **AdaSCALE-A** | **10.02** / **97.80** | **17.99** / **96.38** | 22.93 / 95.17 | **37.38** / **91.62** | **22.08** / **95.24** |
| **AdaSCALE-L** | 11.28 / 97.45 | 18.46 / 96.20 | **21.23** / **95.35** | 37.68 / 91.03 | 22.16 / 95.01 |

Table 31: Far-OOD detection results (FPR@95↓ / AUROC↑) on
ImageNet-1k benchmark using DenseNet-201 network. The best results are
**bold**, and the second-best results are underlined.

| Method | iNaturalist | Textures | OpenImage-O | Places | Average |
|---|---|---|---|---|---|
| MSP | 42.02 / 89.84 | 62.33 / 81.56 | 50.31 / 85.19 | 59.74 / 81.14 | 53.60 / 84.43 |
| MLS | 31.99 / 92.11 | 57.75 / 85.56 | 42.70 / 88.28 | 61.30 / 83.82 | 48.43 / 87.44 |
| EBO | 33.12 / 91.46 | 57.47 / 85.55 | 43.75 / 87.91 | 61.46 / 83.67 | 48.95 / 87.15 |
| ReAct | 19.41 / 95.64 | 23.86 / 94.63 | 32.54 / 91.83 | **47.06** / 88.52 | 30.72 / 92.65 |
| ASH | 21.57 / 95.47 | 21.42 / 95.56 | 41.23 / 90.19 | 49.80 / 87.45 | 33.50 / 92.17 |
| SCALE | 18.13 / **96.29** | 27.22 / 94.52 | 34.52 / 92.15 | 52.82 / 87.83 | 33.17 / 92.70 |
| BFAct | 20.64 / 95.42 | 21.70 / 95.17 | 39.76 / 89.97 | 47.72 / **88.61** | 32.45 / 92.29 |
| LTS | 15.68 / 96.71 | 22.49 / 95.81 | 34.27 / 92.37 | 51.23 / 88.26 | 30.92 / 93.29 |
| OptFS | 25.81 / 93.92 | 21.75 / 95.01 | 38.45 / 89.67 | 51.66 / 85.54 | 34.42 / 91.04 |
| **AdaSCALE-A** | **17.30** / 96.03 | 19.42 / 96.23 | **23.12** / 94.68 | 52.20 / 85.98 | 28.01 / **93.23** |
| **AdaSCALE-L** | 17.97 / 95.87 | **16.87** / **96.69** | 23.64 / **94.69** | 53.50 / 85.46 | **28.00** / 93.18 |

Table 32: Far-OOD detection results (FPR@95↓ / AUROC↑) on
ImageNet-1k benchmark using EfficientNetV2-L network. The best results
are **bold**, and the second-best results are underlined.

| Method | iNaturalist | Textures | OpenImage-O | Places | Average |
|---|---|---|---|---|---|
| MSP | 25.14 / 95.12 | 74.42 / 84.20 | 40.64 / 91.74 | 78.74 / 80.61 | 54.74 / 87.92 |
| MLS | 35.28 / 94.13 | 86.65 / 80.26 | 62.11 / 90.26 | 90.53 / 74.56 | 68.64 / 84.80 |
| EBO | 49.84 / 91.21 | 87.72 / 75.77 | 68.77 / 87.66 | 91.60 / 69.89 | 74.48 / 81.13 |
| ReAct | 46.44 / 80.96 | 54.56 / 77.17 | 60.79 / 78.20 | 78.39 / 64.99 | 60.05 / 75.33 |
| ASH | 96.26 / 37.76 | 95.40 / 50.98 | 97.52 / 43.19 | 97.07 / 34.34 | 96.56 / 41.57 |
| SCALE | 87.08 / 67.69 | 86.22 / 67.44 | 91.05 / 67.21 | 94.18 / 47.99 | 89.63 / 62.58 |
| BFAct | 57.31 / 69.11 | 63.43 / 67.70 | 69.30 / 67.49 | 76.86 / 58.52 | 66.72 / 65.70 |
| LTS | 79.05 / 84.72 | 86.89 / 75.39 | 88.00 / 81.53 | 93.45 / 63.56 | 86.85 / 76.30 |
| OptFS | 38.62 / 89.80 | 45.77 / 86.94 | 53.77 / 85.49 | 76.31 / 72.23 | 53.62 / 83.62 |
| **AdaSCALE-A** | **18.51** / **96.67** | 42.07 / 90.56 | **31.00** / **94.44** | 58.87 / **84.26** | **37.61** / **91.48** |
| **AdaSCALE-L** | 26.58 / 95.02 | **32.81** / **92.38** | 39.19 / 92.31 | **56.66** / 82.33 | 38.81 / 90.51 |

Table 33: Far-OOD detection results (FPR@95↓ / AUROC↑) on ImageNet-1k benchmark using Vit-B-16 network. The best results are **bold**, and the second-best results are underlined.

| Method | iNaturalist | Textures | OpenImage-O | Places | Average |
|---|---|---|---|---|---|
| MSP | 42.40 / 88.19 | 56.46 / 85.06 | 56.19 / 84.86 | 70.59 / 80.38 | 56.41 / 84.62 |
| MLS | 72.98 / 85.29 | 78.93 / 83.74 | 85.78 / 81.60 | 89.88 / 75.05 | 81.89 / 81.42 |
| EBO | 83.56 / 79.30 | 83.66 / 81.17 | 88.82 / 76.48 | 91.77 / 68.42 | 86.95 / 76.34 |
| ReAct | 48.22 / 86.11 | 55.87 / 86.66 | 57.68 / 84.29 | 75.48 / 77.52 | 59.31 / 83.65 |
| ASH | 97.02 / 50.62 | 98.50 / 48.53 | 94.79 / 55.51 | 93.60 / 53.97 | 95.98 / 52.16 |
| SCALE | 86.60 / 73.94 | 84.70 / 79.00 | 89.48 / 72.72 | 92.67 / 63.60 | 88.36 / 72.32 |
| BFAct | 40.56 / 87.96 | 48.65 / 88.31 | 48.24 / 86.59 | 68.86 / 80.21 | 51.58 / 85.77 |
| LTS | 50.42 / 88.92 | 61.70 / 86.53 | 69.26 / 83.45 | 76.07 / 78.82 | 64.37 / 84.43 |
| OptFS | **34.39** / **89.99** | **46.41** / **88.48** | **42.20** / **88.23** | 61.44 / **82.69** | **46.11** / **87.35** |
| **AdaSCALE-A** | 36.38 / 89.60 | 51.13 / 87.16 | 43.02 / 88.07 | **59.97** / 82.48 | 47.63 / 86.83 |
| **AdaSCALE-L** | 35.16 / 89.84 | 50.91 / 87.37 | 43.01 / 88.13 | 60.05 / 82.55 | 47.28 / 86.97 |

Table 34: Far-OOD detection results (FPR@95↓ / AUROC↑) on ImageNet-1k benchmark using Swin-B network. The best results are **bold**, and the second-best results are underlined.

| Method | iNaturalist | Textures | OpenImage-O | Places | Average |
|---|---|---|---|---|---|
| MSP | 55.63 / 86.47 | 79.28 / 80.12 | 81.22 / 81.72 | 77.41 / 79.78 | 73.39 / 82.02 |
| MLS | 93.46 / 78.87 | 94.60 / 74.73 | 97.61 / 70.72 | 94.97 / 69.17 | 95.16 / 73.37 |
| EBO | 95.11 / 67.72 | 95.36 / 69.69 | 97.97 / 60.19 | 95.87 / 58.35 | 96.08 / 63.99 |
| ReAct | 40.77 / 88.60 | 62.26 / 85.54 | 58.19 / 85.76 | 74.21 / 79.16 | 58.86 / 84.77 |
| ASH | 98.59 / 42.18 | 98.55 / 43.37 | 98.23 / 43.28 | 97.57 / 43.98 | 98.23 / 43.20 |
| SCALE | 87.83 / 62.98 | 87.71 / 69.63 | 88.75 / 66.63 | 82.08 / 67.82 | 86.59 / 66.77 |
| BFAct | **25.76** / 91.42 | **45.73** / **87.34** | **32.13** / **91.02** | **52.33** / **84.08** | **38.99** / **88.47** |
| LTS | 57.92 / 86.10 | 77.66 / 78.02 | 73.20 / 80.16 | 82.69 / 72.71 | 72.86 / 79.25 |
| OptFS | 31.94 / 90.56 | 50.27 / 86.91 | 36.50 / 90.18 | 58.38 / 83.51 | 44.27 / 87.79 |
| **AdaSCALE-A** | 32.82 / 90.73 | 61.82 / 85.34 | 38.58 / 89.78 | 58.02 / 82.71 | 47.81 / 87.14 |
| **AdaSCALE-L** | 30.95 / **91.69** | 60.17 / 86.30 | 37.52 / 90.08 | 56.32 / 83.82 | 46.24 / 87.97 |

 **F.3   Full-Spectrum near-OOD detection**

Table 35: Near-FSOOD detection results (FPR@95↓ / AUROC↑) on
ImageNet-1k benchmark using ResNet-50 network. The best results are
**bold**, and the second-best results are underlined.

| Method | SSB-Hard | NINCO | ImageNet-O | Average |
|---|---|---|---|---|
| MSP | 88.17 / 47.34 | 78.15 / 54.73 | 96.29 / 13.81 | 87.54 / 38.63 |
| MLS | 90.04 / 43.32 | 82.06 / 50.23 | 95.59 / 18.94 | 89.23 / 37.50 |
| EBO | 90.19 / 42.62 | 82.64 / 49.01 | 95.54 / 19.57 | 89.46 / 37.07 |
| ReAct | 90.65 / 45.19 | 80.05 / 53.37 | 93.62 / 26.15 | 88.10 / 41.57 |
| ASH | 88.82 / 44.08 | 78.35 / 54.54 | 92.48 / 30.49 | 86.55 / 43.04 |
| SCALE | 85.85 / 48.10 | 77.54 / 57.01 | 93.26 / 32.58 | 85.55 / 45.90 |
| BFAct | 90.43 / 45.29 | 79.62 / 53.50 | 93.62 / 26.20 | 87.89 / 41.66 |
| LTS | 86.37 / 47.43 | 77.54 / 56.40 | 93.61 / 30.57 | 85.84 / 44.80 |
| OptFS | 90.78 / 44.01 | 77.24 / 54.91 | 90.91 / 32.26 | 86.31 / 43.73 |
| **AdaSCALE-A** | **81.30 / 51.88** | **74.13 / 58.55** | **89.15 / 37.62** | **81.52 / 49.35** |
| **AdaSCALE-L** | 81.85 / 51.38 | 74.42 / 58.23 | 90.07 / 36.91 | 82.11 / 48.84 |

Table 36: Near-FSOOD detection results (FPR@95↓ / AUROC↑) on
ImageNet-1k benchmark using ResNet-101 network. The best results are
**bold**, and the second-best results are underlined.

| Method | SSB-Hard | NINCO | ImageNet-O | Average |
|---|---|---|---|---|
| MSP | 87.09 / 49.18 | 76.45 / 56.92 | 94.24 / 28.33 | 85.93 / 44.81 |
| MLS | 88.90 / 46.45 | 79.19 / 53.62 | 93.94 / 31.76 | 87.34 / 43.94 |
| EBO | 89.02 / 45.99 | 79.60 / 52.66 | 93.87 / 32.40 | 87.50 / 43.68 |
| ReAct | 89.50 / 47.79 | 77.22 / 56.02 | 89.37 / 39.62 | 85.36 / 47.81 |
| ASH | 87.84 / 46.39 | 75.36 / 56.72 | 88.48 / 42.80 | 83.90 / 48.64 |
| SCALE | 85.81 / 48.94 | 75.33 / 58.79 | 88.49 / 44.45 | 83.21 / 50.73 |
| BFAct | 89.19 / 48.07 | 76.94 / 56.02 | 89.54 / 39.65 | 85.22 / 47.91 |
| LTS | 86.02 / 48.72 | 74.89 / 58.41 | 89.17 / 43.02 | 83.36 / 50.05 |
| OptFS | 89.63 / 46.23 | 75.52 / 56.71 | 85.82 / 44.21 | 83.65 / 49.05 |
| **AdaSCALE-A** | **82.33 / 51.47** | **74.38 / 59.17** | 82.89 / **48.49** | **79.87 / 53.04** |
| **AdaSCALE-L** | 82.53 / 51.31 | 75.22 / 58.62 | **82.19** / 48.03 | 79.98 / 52.66 |

Table 37: Near-FSOOD detection results (FPR@95↓ / AUROC↑) on ImageNet-1k benchmark using RegNet-Y-16 network. The best results are **bold**, and the second-best results are underlined.

| Method | SSB-Hard | NINCO | ImageNet-O | Average |
|---|---|---|---|---|
| MSP | 83.74 / 57.23 | 72.32 / 67.69 | 87.81 / 56.61 | 81.29 / 60.51 |
| MLS | 82.91 / 60.89 | 71.22 / 70.86 | 93.27 / 55.21 | 82.46 / 62.32 |
| EBO | 82.77 / **61.63** | 71.17 / 71.23 | 93.39 / 55.02 | 82.44 / 62.63 |
| ReAct | 87.74 / 55.64 | 80.22 / 65.24 | 91.11 / 55.13 | 86.36 / 58.67 |
| ASH | 87.26 / 57.45 | 84.81 / 61.59 | 93.78 / 54.76 | 88.61 / 57.93 |
| SCALE | 83.23 / 61.02 | 72.48 / **71.33** | 92.82 / 57.16 | 82.84 / 63.17 |
| BFAct | 90.46 / 54.73 | 87.03 / 61.98 | 92.37 / 54.13 | 89.96 / 56.95 |
| LTS | 83.53 / 60.77 | 74.23 / 70.84 | 92.30 / 57.71 | 83.35 / 63.11 |
| OptFS | 90.33 / 51.78 | 80.82 / 63.86 | 88.86 / 59.45 | 86.67 / 58.36 |
| **AdaSCALE-A** | **81.68** / 60.46 | **68.05** / 70.95 | 83.25 / 60.91 | **77.66** / **64.11** |
| **AdaSCALE-L** | 84.30 / 59.42 | 76.30 / 68.49 | **81.86** / **63.12** | 80.82 / 63.68 |

Table 38: Near-FSOOD detection results (FPR@95↓ / AUROC↑) on ImageNet-1k benchmark using ResNeXt-50 network. The best results are **bold**, and the second-best results are underlined.

| Method | SSB-Hard | NINCO | ImageNet-O | Average |
|---|---|---|---|---|
| MSP | 86.95 / 49.77 | 78.27 / 57.25 | 94.79 / 28.95 | 86.67 / 45.32 |
| MLS | 88.56 / 47.94 | 81.58 / 54.31 | 94.26 / 32.40 | 88.13 / 44.88 |
| EBO | 88.68 / 47.69 | 81.67 / 53.51 | 94.21 / 32.99 | 88.19 / 44.73 |
| ReAct | 89.45 / 47.44 | 80.37 / 55.29 | 91.40 / 37.58 | 87.07 / 46.77 |
| ASH | 86.26 / 49.73 | 79.62 / 57.17 | 92.63 / 39.67 | 86.17 / 48.86 |
| SCALE | 84.64 / **52.60** | 78.75 / 58.90 | 94.20 / 37.99 | 85.86 / 49.83 |
| BFAct | 89.22 / 47.70 | 80.39 / 55.34 | 91.24 / 37.66 | 86.95 / 46.90 |
| LTS | 85.03 / 51.87 | 78.65 / 58.48 | 93.77 / 37.96 | 85.82 / 49.43 |
| OptFS | 89.63 / 46.23 | 75.52 / 56.71 | 85.82 / 44.21 | 83.65 / 49.05 |
| **AdaSCALE-A** | **82.33** / 51.47 | **74.38** / **59.17** | **82.89** / **48.49** | **79.87** / **53.04** |
| **AdaSCALE-L** | 82.70 / 52.14 | 75.85 / 58.05 | 89.55 / 43.81 | 82.70 / 51.33 |

Table 39: Near-FSOOD detection results (FPR@95↓ / AUROC↑) on ImageNet-1k benchmark using DenseNet-201 network. The best results are **bold**, and the second-best results are underlined.

| Method | SSB-Hard | NINCO | ImageNet-O | Average |
|---|---|---|---|---|
| MSP | 87.27 / 49.71 | **76.81** / 58.30 | 95.00 / 29.36 | 86.36 / 45.79 |
| MLS | 89.24 / 47.21 | 80.48 / 55.21 | 95.61 / 30.74 | 88.44 / 44.39 |
| EBO | 89.39 / 46.79 | 80.94 / 54.13 | 95.60 / 31.44 | 88.65 / 44.12 |
| ReAct | 90.76 / 45.54 | 79.54 / 55.57 | 88.40 / 42.63 | 86.23 / 47.91 |
| ASH | 89.75 / 47.08 | 81.35 / 58.13 | 90.47 / 46.81 | 87.19 / 50.67 |
| SCALE | 87.55 / **49.53** | 78.35 / 59.54 | 92.79 / 39.22 | 86.23 / 49.43 |
| BFAct | 92.33 / 44.88 | 83.90 / 54.83 | 84.88 / 49.50 | 87.04 / 49.74 |
| LTS | 87.30 / 50.02 | 78.41 / **60.35** | 92.12 / 42.25 | 85.94 / 50.88 |
| OptFS | 92.32 / 43.61 | 81.45 / 56.09 | 85.02 / **50.48** | 86.26 / 50.06 |
| **AdaSCALE-A** | **86.22** / 48.41 | 80.25 / 56.38 | 81.35 / 47.45 | **82.60** / 50.75 |
| **AdaSCALE-L** | 86.43 / 48.54 | 80.83 / 56.30 | **81.00** / 50.00 | 82.75 / **51.61** |

Table 40: Near-FSOOD detection results (FPR@95↓ / AUROC↑) on ImageNet-1k benchmark using EfficientNetV2-L network. The best results are **bold**, and the second-best results are underlined.

| Method | SSB-Hard | NINCO | ImageNet-O | Average |
|---|---|---|---|---|
| MSP | 83.74 / 57.23 | 72.32 / 67.69 | 87.81 / 56.61 | 81.29 / 60.51 |
| MLS | 82.91 / 60.89 | 71.22 / 70.86 | 93.27 / 55.21 | 82.46 / 62.32 |
| EBO | 82.77 / **61.63** | 71.17 / 71.23 | 93.39 / 55.02 | 82.44 / 62.63 |
| ReAct | 87.74 / 55.64 | 80.22 / 65.24 | 91.11 / 55.13 | 86.36 / 58.67 |
| ASH | 87.26 / 57.45 | 84.81 / 61.59 | 93.78 / 54.76 | 88.61 / 57.93 |
| SCALE | 83.23 / 61.02 | 72.48 / 71.33 | 92.82 / 57.16 | 82.84 / 63.17 |
| BFAct | 90.46 / 54.73 | 87.03 / 61.98 | 92.37 / 54.13 | 89.96 / 56.95 |
| LTS | 83.53 / 60.77 | 74.23 / 70.84 | 92.30 / 57.71 | 83.35 / 63.11 |
| OptFS | 90.33 / 51.78 | 80.82 / 63.86 | 88.86 / 59.45 | 86.67 / 58.36 |
| **AdaSCALE-A** | **81.68** / 60.46 | **68.05** / 70.95 | 83.25 / 60.91 | **77.66** / **64.11** |
| **AdaSCALE-L** | 84.30 / 59.42 | 76.30 / 68.49 | **81.86** / 63.12 | 80.82 / 63.68 |

Table 41: Near-FSOOD detection results (FPR@95↓ / AUROC↑) on
ImageNet-1k benchmark using Vit-B-16 network. The best results are **bold**,
and the second-best results are underlined.

| Method | SSB-Hard | NINCO | ImageNet-O | Average |
|---|---|---|---|---|
| MSP | 92.28 / 47.57 | 87.44 / **56.23** | 98.02 / 39.33 | 92.58 / 47.71 |
| MLS | 94.11 / 44.88 | 95.17 / 52.44 | 98.00 / 37.77 | 95.76 / 45.03 |
| EBO | 94.47 / 42.06 | 95.86 / 48.45 | 97.89 / 38.03 | 96.07 / 42.85 |
| ReAct | 94.95 / 41.84 | 88.65 / 52.48 | 95.21 / 44.64 | 92.94 / 46.32 |
| ASH | **88.95** / **56.47** | 91.09 / 55.11 | 90.00 / **55.78** | 90.01 / 55.79 |
| SCALE | 94.52 / 41.21 | 96.30 / 45.78 | 97.76 / 37.09 | 96.19 / 41.36 |
| BFAct | 94.99 / 41.44 | 85.62 / 53.35 | 92.62 / 45.64 | 91.07 / 46.81 |
| LTS | 95.33 / 43.36 | 90.52 / 53.14 | 95.90 / 41.30 | 93.91 / 45.93 |
| OptFS | 94.19 / 43.01 | 81.66 / 55.60 | 88.90 / 46.04 | 88.25 / **48.22** |
| **AdaSCALE-A** | 93.30 / 42.49 | 81.00 / 54.72 | **84.12** / 45.71 | **86.14** / 47.64 |
| **AdaSCALE-L** | 93.52 / 41.83 | **80.94** / 54.16 | 84.26 / 45.82 | 86.24 / 47.27 |

Table 42: Near-FSOOD detection results (FPR@95↓ / AUROC↑) on
ImageNet-1k benchmark using Swin-B network. The best results are **bold**,
and the second-best results are underlined.

| Method | SSB-Hard | NINCO | ImageNet-O | Average |
|---|---|---|---|---|
| MSP | 91.55 / **53.29** | 86.73 / **60.62** | 97.85 / 42.90 | 92.04 / 52.27 |
| MLS | 94.49 / 50.01 | 93.94 / 56.40 | 97.11 / 43.76 | 95.18 / 50.06 |
| EBO | 94.66 / 47.41 | 94.58 / 52.04 | 96.76 / 45.84 | 95.33 / 48.43 |
| ReAct | 94.04 / 47.83 | 82.85 / 58.52 | 94.60 / 50.41 | 90.50 / 52.25 |
| ASH | 91.77 / 50.35 | 90.91 / 52.09 | 88.80 / 54.50 | 90.49 / 52.31 |
| SCALE | 93.39 / 47.38 | 91.25 / 53.00 | 90.80 / **56.10** | 91.81 / 52.16 |
| BFAct | 92.65 / 48.33 | **79.61** / 59.66 | **84.26** / 53.17 | **85.51** / **53.72** |
| LTS | 94.26 / 48.70 | 88.32 / 57.60 | 93.04 / 46.33 | 91.87 / 50.88 |
| OptFS | 94.06 / 47.77 | 81.91 / 59.14 | 86.95 / 51.34 | 87.64 / 52.75 |
| **AdaSCALE-A** | 90.29 / 46.84 | 81.54 / 57.44 | 87.74 / 47.55 | 86.52 / 50.61 |
| **AdaSCALE-L** | **90.18** / 46.63 | 80.77 / 57.85 | 87.27 / 47.90 | 86.07 / 50.79 |

 **F.4   Full-Spectrum far-OOD detection**

Table 43: Far-FSOOD detection results (FPR@95↓ / AUROC↑) on
ImageNet-1k benchmark using ResNet-50 network. The best results are
**bold**, and the second-best results are underlined.

| Method | iNaturalist | Textures | OpenImage-O | Places | Average |
|---|---|---|---|---|---|
| MSP | 69.31 / 65.65 | 80.57 / 59.22 | 73.94 / 60.74 | 79.02 / 56.04 | 75.71 / 60.41 |
| MLS | 64.71 / 63.30 | 74.69 / 61.67 | 69.73 / 60.60 | 80.04 / 55.08 | 72.29 / 60.16 |
| EBO | 65.30 / 61.43 | 74.48 / 61.87 | 69.92 / 59.93 | 80.12 / 54.48 | 72.45 / 59.42 |
| ReAct | 51.90 / 75.79 | 63.55 / 69.22 | 65.64 / 67.36 | 71.81 / 67.01 | 63.22 / 69.84 |
| ASH | 49.21 / 76.93 | 50.54 / 77.64 | 63.04 / 69.03 | 70.62 / 64.79 | 58.35 / 72.10 |
| SCALE | 43.34 / 79.23 | 46.60 / 79.58 | 62.26 / 70.54 | 67.94 / 67.05 | 55.04 / 74.10 |
| BFAct | 51.01 / 75.85 | 62.45 / 69.03 | 65.52 / 67.21 | 71.17 / 66.45 | 62.54 / 69.63 |
| LTS | 45.12 / 78.72 | 48.77 / 79.13 | 62.43 / 70.17 | 69.31 / 66.20 | 56.41 / 73.55 |
| OptFS | 49.39 / 77.14 | 50.25 / 75.59 | 62.46 / 68.87 | 69.84 / 65.65 | 57.99 / 71.81 |
| **AdaSCALE-A** | **41.24** / 79.53 | 45.55 / 79.64 | **56.43** / **72.85** | **66.13** / **67.54** | **52.33** / **74.89** |
| **AdaSCALE-L** | 41.75 / **79.63** | 45.67 / **79.96** | 56.80 / 72.82 | 66.69 / 67.28 | 52.73 / 74.92 |

Table 44: Far-FSOOD detection results (FPR@95↓ / AUROC↑) on
ImageNet-1k benchmark using ResNet-101 network. The best results are
**bold**, and the second-best results are underlined.

| Method | iNaturalist | Textures | OpenImage-O | Places | Average |
|---|---|---|---|---|---|
| MSP | 71.78 / 64.51 | 78.82 / 62.20 | 72.52 / 62.27 | 78.71 / 57.55 | 75.46 / 61.63 |
| MLS | 70.55 / 62.07 | 72.11 / 65.46 | 68.68 / 62.27 | 77.60 / 58.03 | 72.23 / 61.96 |
| EBO | 70.94 / 60.64 | 72.18 / 65.71 | 68.96 / 61.59 | 77.97 / 57.75 | 72.51 / 61.42 |
| ReAct | 53.59 / 75.05 | 59.92 / 72.13 | 62.45 / 69.16 | 71.14 / 66.87 | 61.78 / 70.80 |
| ASH | 53.84 / 74.07 | 47.67 / 79.09 | 60.66 / 70.12 | 71.45 / 64.40 | 58.40 / 71.92 |
| SCALE | 47.82 / 76.79 | 42.04 / 80.95 | 59.26 / 71.76 | 70.32 / 65.86 | 54.86 / 73.84 |
| BFAct | 53.23 / 74.86 | 58.84 / 71.86 | 62.33 / 68.97 | 70.67 / 66.17 | 61.27 / 70.47 |
| LTS | 49.64 / 76.17 | 43.87 / 80.60 | 59.32 / 71.46 | 70.85 / 65.35 | 55.92 / 73.39 |
| OptFS | 51.52 / 75.52 | 48.88 / 76.96 | 59.99 / 70.28 | 70.86 / 65.15 | 57.81 / 71.98 |
| **AdaSCALE-A** | **44.77** / **77.51** | **42.17** / **80.73** | 53.58 / **73.90** | **67.14** / **66.84** | **51.91** / **74.75** |
| **AdaSCALE-L** | 46.52 / 76.39 | 44.87 / 79.90 | **53.32** / 73.81 | 68.16 / 65.88 | 53.22 / 73.99 |

Table 45: Far-FSOOD detection results (FPR@95↓ / AUROC↑) on
ImageNet-1k benchmark using RegNet-Y-16 network. The best results are
**bold**, and the second-best results are underlined.

| Method | iNaturalist | Textures | OpenImage-O | Places | Average |
|---|---|---|---|---|---|
| MSP | 53.98 / 80.49 | 69.36 / 70.97 | 62.02 / 75.95 | 75.48 / 65.76 | 65.21 / 73.29 |
| MLS | 39.99 / 85.09 | 69.12 / 73.89 | 58.30 / 80.08 | 79.98 / 66.96 | 61.85 / 76.51 |
| EBO | 38.42 / 86.11 | 68.23 / 74.16 | 58.77 / 80.93 | 80.56 / 67.08 | 61.49 / 77.07 |
| ReAct | 43.95 / 85.15 | 66.24 / 73.34 | 68.23 / 77.80 | 88.79 / 57.47 | 66.80 / 73.44 |
| ASH | 61.03 / 79.08 | 58.16 / 80.78 | 80.03 / 74.57 | 81.60 / 67.74 | 70.21 / 75.54 |
| SCALE | 39.13 / 86.34 | 56.30 / 79.93 | 60.66 / 81.10 | 76.25 / 70.04 | 58.09 / 79.35 |
| BFAct | 52.17 / 82.68 | 71.55 / 70.74 | 78.37 / 74.00 | 90.50 / 56.09 | 73.15 / 70.88 |
| LTS | 38.98 / 86.90 | 50.10 / 82.26 | 65.29 / 81.17 | 75.41 / **70.91** | 57.44 / 80.31 |
| OptFS | 52.61 / 82.40 | 62.47 / 76.57 | 66.68 / 76.79 | 88.01 / 56.14 | 67.44 / 72.98 |
| **AdaSCALE-A** | 34.25 / 87.68 | 63.81 / 76.12 | **50.37** / 82.13 | 74.81 / 68.37 | 55.81 / 78.58 |
| **AdaSCALE-L** | **32.72** / **88.82** | **47.93** / **82.84** | 53.84 / **83.09** | **73.90** / 70.37 | **52.10** / **81.28** |

Table 46: Far-FSOOD detection results (FPR@95↓ / AUROC↑) on
ImageNet-1k benchmark using ResNeXt-50 network. The best results are
**bold**, and the second-best results are underlined.

| Method | iNaturalist | Textures | OpenImage-O | Places | Average |
|---|---|---|---|---|---|
| MSP | 68.90 / 66.67 | 80.91 / 60.21 | 71.91 / 63.25 | 78.58 / 57.84 | 75.07 / 62.00 |
| MLS | 64.59 / 65.49 | 76.51 / 62.51 | 67.70 / 63.96 | 79.89 / 56.90 | 72.17 / 62.21 |
| EBO | 64.95 / 64.23 | 76.59 / 62.61 | 68.00 / 63.52 | 79.80 / 56.39 | 72.34 / 61.69 |
| ReAct | 51.97 / 76.08 | 65.34 / 68.27 | 63.05 / 68.94 | 70.13 / **67.58** | 62.62 / 70.22 |
| ASH | 53.84 / 74.07 | 47.67 / 79.09 | 60.66 / 70.12 | 71.45 / 64.40 | 58.40 / 71.92 |
| SCALE | 49.27 / 76.82 | 60.07 / 75.14 | 62.91 / 70.69 | 73.38 / 64.22 | 61.41 / 71.72 |
| BFAct | 53.23 / 74.86 | 58.84 / 71.86 | 62.33 / 68.97 | 70.67 / 66.17 | 61.27 / 70.47 |
| LTS | 49.64 / 76.17 | 48.77 / 79.13 | 62.43 / 70.17 | 70.85 / 65.35 | 56.41 / 73.55 |
| OptFS | 51.52 / 75.77 | 50.25 / 75.59 | 62.46 / 68.87 | 69.84 / 65.65 | 57.99 / 71.81 |
| **AdaSCALE-A** | **43.82** / **78.47** | 45.55 / **79.64** | **56.43** / **72.85** | **66.13** / 67.54 | **52.33** / **74.89** |
| **AdaSCALE-L** | 46.23 / 76.67 | 54.29 / 75.80 | 56.84 / 72.44 | 69.13 / 64.99 | 56.62 / 72.47 |

Table 47: Far-FSOOD detection results (FPR@95↓ / AUROC↑) on ImageNet-1k benchmark using DenseNet-201 network. The best results are **bold**, and the second-best results are underlined.

| Method | iNaturalist | Textures | OpenImage-O | Places | Average |
|---|---|---|---|---|---|
| MSP | 68.90 / 66.67 | 80.91 / 60.21 | 71.91 / 63.25 | 78.58 / 57.84 | 75.07 / 62.00 |
| MLS | 64.59 / 65.49 | 76.51 / 62.51 | 67.70 / 63.96 | 79.89 / 56.90 | 72.17 / 62.21 |
| EBO | 64.95 / 64.23 | 76.59 / 62.61 | 68.00 / 63.52 | 79.80 / 56.39 | 72.34 / 61.69 |
| ReAct | 51.97 / 76.08 | 65.34 / 68.27 | 63.05 / 68.94 | 70.13 / **67.58** | 62.62 / 70.22 |
| ASH | 53.84 / 74.07 | **47.67** / 79.09 | 60.66 / 70.12 | 71.45 / 64.40 | 58.40 / 71.92 |
| SCALE | **49.27** / **76.82** | 60.07 / 75.14 | 62.91 / 70.69 | 73.38 / 64.22 | 61.41 / 71.72 |
| BFAct | 53.23 / 74.86 | 58.84 / 71.86 | 62.33 / 68.97 | 70.67 / 66.17 | 61.27 / 70.47 |
| LTS | 49.64 / 76.17 | 48.77 / **79.13** | 62.43 / 70.17 | 70.85 / 65.35 | **56.41** / **73.55** |
| OptFS | 51.52 / 75.77 | 50.25 / 75.59 | 62.46 / 68.87 | **69.84** / 65.65 | 57.99 / 71.81 |
| **AdaSCALE-A** | 52.55 / 73.62 | 54.70 / 76.19 | **58.11** / 71.31 | 77.77 / 58.83 | 60.78 / 69.99 |
| **AdaSCALE-L** | 53.04 / 73.62 | 51.83 / 78.00 | 58.30 / **71.92** | 78.40 / 58.47 | 60.39 / 70.50 |

Table 48: Far-FSOOD detection results (FPR@95↓ / AUROC↑) on ImageNet-1k benchmark using EfficientNetV2-L network. The best results are **bold**, and the second-best results are underlined.

| Method | iNaturalist | Textures | OpenImage-O | Places | Average |
|---|---|---|---|---|---|
| MSP | 53.98 / 80.49 | 69.36 / 70.97 | 62.02 / 75.95 | 75.48 / 65.76 | 65.21 / 73.29 |
| MLS | 39.99 / 85.09 | 69.12 / 73.89 | 58.30 / 80.08 | 79.98 / 66.96 | 61.85 / 76.51 |
| EBO | **38.42** / 86.11 | 68.23 / 74.16 | 58.77 / 80.93 | 80.56 / 67.08 | 61.49 / 77.07 |
| ReAct | 43.95 / 85.15 | 66.24 / 73.34 | 68.23 / 77.80 | 88.79 / 57.47 | 66.80 / 73.44 |
| ASH | 61.03 / 79.08 | 58.16 / 80.78 | 80.03 / 74.57 | 81.60 / 67.74 | 70.21 / 75.54 |
| SCALE | 39.13 / 86.34 | 56.30 / 79.93 | 60.66 / 81.10 | 76.25 / 70.04 | 58.09 / 79.35 |
| BFAct | 52.17 / 82.68 | 71.55 / 70.74 | 78.37 / 74.00 | 90.50 / 56.09 | 73.15 / 70.88 |
| LTS | 38.98 / 86.90 | 50.10 / 82.26 | 65.29 / 81.17 | 75.41 / **70.91** | 57.44 / 80.31 |
| OptFS | 52.61 / 82.40 | 62.47 / 76.57 | 66.68 / 76.79 | 88.01 / 56.14 | 67.44 / 72.98 |
| **AdaSCALE-A** | 40.76 / **88.62** | 63.95 / 77.69 | **53.96** / **84.71** | 77.17 / 69.22 | 58.96 / 80.06 |
| **AdaSCALE-L** | 43.28 / 87.91 | 50.23 / **83.05** | 57.11 / 84.45 | **73.87** / 70.29 | **56.12** / **81.43** |

Table 49: Far-FSOOD detection results (FPR@95↓ / AUROC↑) on ImageNet-1k benchmark using ViT-B-16 network. The best results are **bold**, and the second-best results are underlined.

| Method | iNaturalist | Textures | OpenImage-O | Places | Average |
|---|---|---|---|---|---|
| MSP | 66.12 / **67.29** | 75.49 / 64.02 | 75.32 / 63.46 | 83.84 / **58.77** | 75.19 / 63.39 |
| MLS | 82.34 / 64.58 | 85.83 / 63.52 | 90.12 / 61.11 | 92.95 / 55.08 | 87.81 / 61.07 |
| EBO | 87.94 / 59.51 | 88.03 / 62.20 | 91.84 / 57.54 | 94.14 / 50.68 | 90.49 / 57.48 |
| ReAct | 70.31 / 62.56 | 75.49 / 64.90 | 76.64 / 61.30 | 87.02 / 54.77 | 77.37 / 60.88 |
| ASH | 93.23 / 53.23 | 95.19 / 51.16 | 90.38 / 57.95 | 89.04 / 56.56 | 91.96 / 54.72 |
| SCALE | 90.13 / 55.74 | 88.71 / 61.09 | 92.30 / 55.21 | 94.75 / 47.62 | 91.47 / 54.92 |
| BFAct | 66.39 / 62.89 | 71.84 / 65.55 | 71.56 / 62.19 | 84.23 / 55.84 | 73.51 / 61.62 |
| LTS | 71.01 / 67.15 | 78.15 / 65.01 | 82.72 / 60.98 | 86.73 / 56.56 | 79.65 / 62.43 |
| OptFS | **62.89** / 66.41 | **70.96** / **65.68** | **68.25** / **64.30** | 80.02 / 58.53 | **70.53** / **63.73** |
| **AdaSCALE-A** | 65.49 / 65.27 | 74.75 / 63.69 | 69.79 / 63.49 | **79.93** / 57.47 | 72.49 / 62.48 |
| **AdaSCALE-L** | 64.72 / 64.84 | 74.72 / 63.49 | 69.92 / 62.86 | 79.99 / 57.04 | 72.34 / 62.06 |

Table 50: Far-FSOOD detection results (FPR@95↓ / AUROC↑) on ImageNet-1k benchmark using Swin-B network. The best results are **bold**, and the second-best results are underlined.

| Method | iNaturalist | Textures | OpenImage-O | Places | Average |
|---|---|---|---|---|---|
| MSP | 73.78 / 70.69 | 87.48 / 63.62 | 88.58 / 65.05 | 86.43 / 62.22 | 84.07 / 65.39 |
| MLS | 94.01 / 64.59 | 94.98 / 61.02 | 97.72 / 57.01 | 95.29 / 54.55 | 95.50 / 59.29 |
| EBO | 95.11 / 55.73 | 95.35 / 58.70 | 97.99 / 49.71 | 95.87 / 47.30 | 96.08 / 52.86 |
| ReAct | 66.17 / 68.53 | 79.27 / 66.85 | 76.92 / 65.82 | 85.99 / 57.89 | 77.09 / 64.77 |
| ASH | 94.55 / 47.18 | 94.43 / 48.22 | 93.77 / 48.28 | 92.55 / 48.99 | 93.82 / 48.17 |
| SCALE | 91.23 / 53.32 | 91.15 / 60.76 | 91.87 / 57.44 | 87.05 / 58.13 | 90.32 / 57.41 |
| BFAct | **54.53** / **73.59** | **69.73** / **68.65** | **59.76** / **74.29** | **74.09** / **64.30** | **64.53** / **70.21** |
| LTS | 72.97 / 70.44 | 86.21 / 62.18 | 83.29 / 63.87 | 89.39 / 56.32 | 82.96 / 63.20 |
| OptFS | 58.91 / 71.84 | 72.14 / 68.00 | 62.45 / 72.16 | 77.16 / 63.35 | 67.66 / 68.83 |
| **AdaSCALE-A** | 61.33 / 68.58 | 79.90 / 64.03 | 65.40 / 67.52 | 77.68 / 59.21 | 71.08 / 64.83 |
| **AdaSCALE-L** | 60.15 / 70.58 | 78.72 / 65.23 | 64.83 / 68.17 | 76.43 / 60.64 | 70.03 / 66.15 |

