# OpenReview forum: "AdaSCALE: Adaptive Scaling for OOD detection"
_NeurIPS.cc/2025/Conference — Submitted to NeurIPS 2025_

### Official Review · Reviewer_twGb · 2025-06-16

**Clarity:** 2
**Significance:** 2
**Originality:** 2
**Rating:** 4
**Confidence:** 4

**Summary:**

## TLDR:

The paper proposes a method for computing a score used for out-of-distribution (OOD) detection.
The authors use the well known energy score $E(x) = − log \sum_{i=1}^{C} e(z_i(x))$, where $z_i$ is the logit of the i-th class.
However, before computing $E(x)$, the authors rescale the activations (i.e., the feature vector), thus producing different logits than the baseline predictor would.

The authors report state-of-the-art (SotA) results on the OOD detection benchmark OpenOOD.

## Method summary:

For the sake of the review, I will briefly introduce the key concepts from the paper.

A classifier $h: \mathcal{X} \rightarrow \mathcal{Y}$ can be decomposed into a feature extractor $f_\theta: \mathcal{X} \rightarrow \mathbb{R}^D$ and the prediction head $g_{\mathcal{W}}: \mathbb{R}^D \rightarrow \mathbb{R}^C$, where $D$ is the dimension of the feature vector (activations), $C$ is the number of classes, $\theta$ parametrize the feature extractor and $\mathcal{W}$ parametrize the head. Activations for a sample $x$ are $a=f_\theta(x)$ and the logits are $z=g_\mathcal{W}(f_\theta(x)) = g_\mathcal{W}(a)$.

AdaSCALE computes the OOD score for a sample $x$ as $E(x) = − log \sum_{i=1}^{C} e(z_i(x))$.
Before computing $E(x)$, AdaSCALE recomputes the activations (and therefore also the logits) as either $a_{\text{scaled}} = a \cdot e^r$ or $z_{\text{scaled}} = z \cdot r^2$, where $r\in\mathbb{R}$ is a scaling factor computed as $r=\frac{ \sum_{j} a_j}{\sum_{j: a_j > P_p(a)} a_j}$ and $P_p(a)$ denotes the a-th percentile of elements of $a$.

AdaSCALE does NOT use a fixed percentile $p$, but computes it as $p(x)=p_{\text{min}} - (1-F_{Q'}(Q'(x))) (p_{\text{max}} - p_{\text{min}})$, where $p_{\text{min}}, p_{\text{max}} \in [0, 1]$ are hyperparameters, $Q'(x)$ is a score computed as $Q'(x)=\lambda Q(x) + C_o(x)$, $\lambda\in\mathbb{R}$. Components of the score $Q'(x)$ are $Q(x)=\sum_{j: j \text{ are indices of }k_1 \text{ highest values of }a} |a_j^\epsilon(x^\epsilon) - a_j(x)|$, where $x^\epsilon$ is a perturbation of the observation $x$; The authors consider gradient based perturbations or random ones; The other component of $Q'$ is $C_o(x)=\sum_{j: j \text{ are indices of }k_2 \text{ highest values of }a} ReLU(a_j^\epsilon)$.

**Questions:**

## Core Method and its Motivation
1) Contradictory Formulation of the $Q'$ Score:
   - The paper states that a high $Q(x)$ value indicates a sample is likely OOD (lines 183-184), while a high $C_o(x)$ value indicates a sample is likely ID (lines 187-189). However, the score $Q'$ is formulated as $Q'(x) = \lambda Q(x) + C_o(x)$.
   - Could you please clarify the reasoning behind **adding** these two seemingly opposing scores?
   - Is this a typographical error in the manuscript?

2) Motivation Behind the Heuristic Design:
   - The AdaSCALE method is a complex composition of several steps. While the empirical results are very strong, the paper lacks a clear motivation for why this specific combination of heuristics is the right approach.
   - Could you provide a more detailed intuition or theoretical justification for the design? Have you tried some other approaches that did not work?
   - The ablation studies show an extensive search over different components and hyperparameters. Can you provide assurance that the final method generalizes well, rather than being a configuration that has been over-tuned to the OpenOOD benchmark? From the Appendix D I see that the hyperparameters differ mostly in $p_{\text{min}}$ and $p_{\text{max}}$, but for some architectures $k_2$ is also substantially different.

## Clarification on Terminology and Figures
3)  Use of "OOD Likelihood":
    - The paper repeatedly uses the term "OOD likelihood". Is the use consistent with the interpretation $p_{\text{OOD}}(x)$?
4)  Interpretation of Figure 5:
    - I found the AUROC plot in Figure 5 difficult to interpret.
    - Could you please clarify how to read this graph? Are the axes inverted, or are the labels swapped?
5)  Undefined Terms:
    - Could you please provide a definition for the term **"trivial pixels"**?


The empirical results appear very strong and I am open to raising my rating if these questions are addressed, especially questions 1 and 2.

**Ethical Concerns:**

["NO or VERY MINOR ethics concerns only"]

**Final Justification:**

I decided to increase my final rating from a borderline reject to a borderline accept.

In my initial review I concluded that the AdaSCALE method is a complex composition of several steps and that the paper lacks a clear motivation for why this specific combination of heuristics is the right approach. I still believe this to be true. However, the authors have demonstrated that though the combination introduces several hyperparameters, the method is not overly sensitive to them or they can be set from results of prior works.

Overall I feel that the authors addressed my concerns sufficiently to raise the score.

I believe that a discussion between the reviewers is needed to ultimately decide whether to accept or reject the paper.

**Limitations:**

Yes.

**Paper Formatting Concerns:**

No major concerns.

**Quality:**

3

**Strengths And Weaknesses:**

The method is introduced in a structured manner, however, I believe that the structure could be improved. The authors first introduce observation perturbations, then they introduce the scores $Q$ and $C_o$ and the scaling factor $r$. Then they introduce the per sample percentile $p$. Only after this do they specify how they compute the energy score $E$. I would advise that the authors take the opposite route - first introduce the energy score $E$ and comment on how it is affected by the scaling $r$ of the activations. Then propose the changes to the scaling factor compared to previous works and show why the changes are sensible.

The authors report SotA results on the OpenOOD benchmark. AdaSCALE builds on top of previous methods which notice that clipping or scaling the model activations (feature vector) results in the energy score $E$ being more separable for ID and OOD data.
The novelty of AdaSCALE comes from the specific process of scaling the activations.

## Notes:
- lines 18-19: "... ability to handle previously unseen inputs, a task commonly known as OOD detection ..."
   - I would recommend reformulating this. We want predictive models to handle unseen inputs from ID = generalization. The sentence does not properly describe OOD detection
- line 42: the percentile threshold $p$ is not introduced by this point in the paper
- line 46: "... We hypothesize that designing an adaptive scaling procedure based on each sample’s **predetermined OOD likelihood** offers greater control for enhancing ID-OOD separability ..."
   - The authors use the term **OOD likelihood** in the paper many times and refer to the score $Q(x)$ as such (line 179). OOD likelihood is $p_\text{OOD}(x)$. If the likelihood is predetermined then the authors are assuming some fixed distribution as their OOD distribution. This however does not seem to be the case in the remainder of the paper. The use of the term seems inconsistent to me and I do not understand it. Please note that OOD likelihood is $p_{\text{OOD}(x)} = p(x|\text{OOD})$, but likelihood of $x$ being an OOD sample is $p(\text{OOD}|x)$
- lines 125-130: By "classifier" the authors refer to both $h$ and $g_\mathcal{W}$. Consider e.g., using *classifier head* for $g_\mathcal{W}$
- lines 183-184: High $Q(x) \implies \sim \text{OOD}$; lines 187-189: High $C_o(x) \implies \sim \text{ID}$. Why is the combined score $Q'$ then the sum of these two terms with positive coefficients? (The authors use $\lambda > 0$). Is there a typo here? Should $C_o$ be subtracted instead?
- lines 208-210 and lines 212-215: Can you provide the reasoning behind this? I understand what you propose, but I do not see the motivation for this. It appears that the authors propose a modification of the SCALE method, where they use a dynamic, but heuristic scaling.
- figure 5: How do we read the graph for AUROC? E.g. for the upper right corner we get $p_{\text{min}}=90$, $p_{\text{max}}=65$. Are the annotations for AUROC swapped? (min should be max and vice versa) Or are the axes reversed? (increasing from left to right, and from top to bottom?)
- line 295: The term **trivial pixels** is not defined. Are these the pixels selected by gradient attribution?

## Strengths:
- Strong empirical results; Reported SotA results on OpenOOD benchmark
- Good literature review with proper citations of methods that inspired AdaSCALE
- Extensive result tables
- Presence of ablation studies

## Weaknesses:
- Heuristic modification of previous work
   - While the paper achieves strong empirical results, the proposed method, AdaSCALE, appears to be a complex combination of heuristics stacked upon existing techniques. The novelty lies in a sample-adaptive scaling factor, but the derivation of this factor lacks clear justification. The calculation of the percentile $p(x)$ depends on a score $Q'(x)$, which itself is a weighted sum of two other scores, $Q(x)$ and $C_o(x)$. Each of these components, including the use of perturbations and the selection of top-k activations, is introduced without a strong motivation. The approach seems like an engineering solution that was found to work well. Further, based on Tables 5 and 7, and the Figures 5,6,7, it seems that the authors tried many different options for the score $Q'$ and the hyperparameters. Though this is praiseworthy as an ablation study, it raises the question whether it does not lead to "searching for the best results on the test set".
- Contradiction in the formulation of the method
   - lines 183-184: High $Q(x) \implies \sim \text{OOD}$; lines 187-189: High $C_o(x) \implies \sim \text{ID}$. Why is the combined score $Q'$ then the sum of these two terms with positive coefficients? (The authors use $\lambda > 0$). Is there a typo here? Should $C_o$ be subtracted instead?
- Some formulations are not clear
   - line 46: "... We hypothesize that designing an adaptive scaling procedure based on each sample’s **predetermined OOD likelihood** offers greater control for enhancing ID-OOD separability ..."
      - The authors use the term **OOD likelihood** in the paper many times and refer to the score $Q(x)$ as such (line 179). OOD likelihood is $p_\text{OOD}(x)$. If the likelihood is predetermined then the authors are assuming some fixed distribution as their OOD distribution. This however does not seem to be the case in the remainder of the paper. The use of the term seems inconsistent to me and I do not understand it. Please note that OOD likelihood is $p_{\text{OOD}(x)} = p(x|\text{OOD})$, but likelihood of $x$ being an OOD sample is $p(\text{OOD}|x)$
   - figure 5: How do we read the graph for AUROC? E.g. for the upper right corner we get $p_{\text{min}}=90$, $p_{\text{max}}=65$. Are the annotations for AUROC swapped? (min should be max and vice versa) Or are the axes reversed? (increasing from left to right, and from top to bottom?)
   - line 295: The term **trivial pixels** is not defined. Are these the pixels selected by gradient attribution?

---

> ### Author Rebuttal · Authors · 2025-07-31
>
> We sincerely thank Reviewer twGb for their incredibly thorough and detailed review. We are grateful for the positive feedback on our "strong empirical results," "SotA performance," "good literature review," and "extensive ablation studies." We also appreciate the constructive suggestions for improving the paper's clarity and structure, which we will incorporate into the final version.
>
> The reviewer twGb's main concerns relate to the motivation behind our score formulation and the method's potential complexity and generalization. We believe the clarifications and extensive new supporting evidence below directly address these points and robustly demonstrate that our method is principled, not over-tuned, and highly generalizable.
>
> ---
>
> ### **1. Core Method Motivation: Justification for the $Q'$ Score Formulation**
>
> *   **The formulation of $Q' = Q + \lambda C_o$ is an intentional design choice. The $C_o$ term acts as a crucial regularizer to balance the influence of the feature-based OODness score ($Q$) and the original logits, preventing overconfident scaling and leading to superior OOD detection.**
>
>     Reviewer twGb correctly identifies that $Q$ indicates OOD-likeness while $C_o$ indicates ID-likeness, and questions their addition. The motivation is to prevent the scaling factor, which is derived from $Q$, from completely dominating the final energy score.
>
>     As noted in Lines 185-187, relying solely on $Q$ can lead to "overoptimistic estimations." If an OOD sample gets a very high $Q$ score, it will produce a very large scaling factor. This can amplify the logits so much that it washes out the subtle but important information contained within the *relative* values of the logits themselves. By adding the opposing $C_o$ term, we "tame" or regularize these potentially overconfident scaling factors. This ensures a healthier balance where both the feature-level instability (captured by $Q$) and the original logit distribution contribute to the final OOD score.
>
> *   **Illustrative Example:**
>     A simple toy example demonstrates why this balancing is critical. Assume a threshold of 25, where `energy > 25` is ID and `energy < 25` is OOD (We deal with magnitude for simplicity).
>
>     -   **Logits:** Let $\\text{logit}_\text{ID} = (1, 6, 2)$ and $\text{logit}_\text{OOD} = (1, 1, 1)$.
>     -   **Case 1: Overconfident Scaling (using only $Q$)**
>         Let's say this yields scaling factors $r_\text{ID}=2$ and $r_\text{OOD}=5$.
>         -   $\text{energy}_\text{ID} = \log( e^{1 \cdot 2^2} + e^{6 \cdot 2^2} + e^{2 \cdot 2^2} ) = 24.0$
>         -   $\text{energy}_\text{OOD} = \log( e^{1 \cdot 5^2} + e^{1 \cdot 5^2} + e^{1 \cdot 5^2}) = 26.1$
>
>       Here, $\\text{energy}_\\text{OOD} > \\text{energy}_\\text{ID}$, leading to an **OOD detection failure**.
>
>     -   **Case 2: Balanced Scaling (using $Q'$ to temper the scaling)**
>         This leads to less extreme scaling, e.g., $r_\text{ID}=1$ and $r_\text{OOD}=2$. Assume a threshold of 5, where `energy > 5.5` is ID and `energy < 5.5` is OOD (We deal with magnitude for simplicity).
>         -   $\text{energy}_\text{ID} = \log( e^{1 \cdot 1^2} + e^{6 \cdot 1^2} + e^{2 \cdot 1^2} )=6.0$
>         -   $\text{energy}_\text{OOD} = \log( e^{1 \cdot 2^2} + e^{1 \cdot 2^2} + e^{1 \cdot 2^2} )=5.1$
>
>      Here, $\\text{energy}_\\text{ID} > \\text{energy}_\\text{OOD}$, leading to a **successful detection**.
>
> *   **Empirical Validation:**
>     The superiority of the combined score $Q'$ over just using $Q$ is confirmed by the results below. Using $Q'$ consistently and significantly outperforms using $Q$ alone across all 8 diverse architectures.
>
> | Category | OODness Metric | ResNet-50 | ResNet-101 | RegNet-Y-16 | ResNeXt-50 | DenseNet-201 | EfficientNetV2-L | ViT-B-16 | Swin-B | Average |
> | :--- | :--- | :---: | :---: | :---: | :---: | :---: | :---: | :---: | :---: | :---: |
> | **near-OOD** | $Q$ | 59.65/77.82 | 56.67/81.65 | 60.63/86.00 | 63.99/79.34 | 68.65/78.03 | 73.18/80.17 | 82.61/69.31 | 75.85/72.26 | 68.90/78.07 |
> | | $Q'$ (ours) | **58.98**/**78.98** | **57.96**/**81.68** | **47.91**/**89.18** | **64.14**/**79.96** | **61.28**/**79.66** | **53.78**/**86.94** | **71.87**/**73.14** | **73.41**/**74.48** | **61.17**/**80.50** |
> | **far-OOD** | $Q$ | 19.82/95.72 | 20.32/95.50 | 31.53/93.94 | 26.87/94.32 | 35.55/92.19 | 66.20/84.60 | 77.52/76.39 | 63.20/80.78 | 42.63/89.18 |
> | | $Q'$ (ours) | **17.84**/**96.14** | **18.51**/**95.95** | **21.37**/**95.84** | **22.08**/**95.24** | **28.01**/**93.23** | **37.61**/**91.48** | **47.63**/**86.83** | **47.81**/**87.14** | **30.11**/**92.73** |
>
> ---
>
> ### **2. On Heuristic Design and Generalization**
>
> *   **We provide new, extensive results to assure the reviewer that AdaSCALE generalizes exceptionally well and is not over-tuned. We achieve SotA performance even when we fix almost all hyperparameters and tune only a single parameter, demonstrating the robustness of our approach.**
>
>     To directly address the valid concern about "searching for the best results on the test set," we provide the following evidence:
>
>     1.  **Near-Insensitivity to most Hyperparameters:** Our results show that many hyperparameters are robust. For instance, fixing $k_2=5\\%$ across all networks results in almost no significant average performance drop from the tuned version on ImageNet-1k benchmark (62.12 vs 61.17 FPR95 near-OOD; 30.25 vs 30.11 far-OOD).
>
>     2.  **Generalization with Single Hyperparameter Tuning:** To provide the strongest possible assurance, we ran a new experiment where we fixed all hyperparameters except one ($p_\text{max}$, tuned from a discrete set) across all 8 architectures on ImageNet-1k. Even in this highly constrained setting, AdaSCALE drastically outperforms the prior SotA (OptFS).
>
> | Category | Method | ResNet-50 | ResNet-101 | RegNet-Y-16 | ResNeXt-50 | DenseNet-201 | EfficientNetV2-L | ViT-B-16 | Swin-B | Average |
> | :--- | :--- | :---: | :---: | :---: | :---: | :---: | :---: | :---: | :---: | :---: |
> | **near-OOD** | OptFS | 69.66/70.97 | 65.46/75.83 | 73.53/75.21 | 69.27/74.84 | 71.74/72.10 | 72.29/75.29 | 76.55/72.73 | 76.81/74.06 | 71.91/73.88 |
> || AdaSCALE-A | **59.68**/**78.35**|**58.02**/**81.03**|**52.67**/**88.25**|**62.56**/**79.16**|**66.66**/**75.24**|**54.79**/**86.53**|**71.81**/**73.63**|**73.24**/**74.77**|**62.29**/**79.72** |
> | **far-OOD** | OptFS | 25.66 / 93.87 | 26.97 / 93.55 | 47.37 / 86.73 |  27.54 / 93.40 | **34.42** / **91.04** | 53.62 / 83.62 | **46.11** / **87.35** | **44.27** / **87.79** | 38.25 / 89.67 |
> | | AdaSCALE-A | **21.86** / **94.96** | **21.94** / **94.91** | **19.19** / **96.24** | **25.17** / **93.90** | 37..03 / 90.10 | **39.48** / **91.20** | 48.47 / 86.81 | 48.65 / 86.40 | **32.72** / **91.82** |
>
> This result (+13% FPR95 / +8% AUROC on near-OOD vs. OptFS) confirms that the core mechanism of AdaSCALE is powerful and generalizable, not the result of over-tuning. Its practical application is comparable to prior work like ASH/SCALE, which also require tuning one parameter. We will add this full analysis to the paper.
>
> Additionally, we present the results of both DenseNet and WideResNet networks with same set of hyperparameters in CIFAR-100 benchmarks **only allowing $p_\text{max}$ to be tuned**:
>
> CIFAR-100 with WideResNet-40-2
>
> | Method      | Near-OOD | Far-OOD |
> |--------------|-------------------|------------------|
> | SCALE       |    56.28 / 82.00  | 52.04 / 82.74 |
> | OptFS       | 74.76 / 78.27    | 54.53 / 80.78 |
> | **AdaSCALE-A** | **55.89** / **82.10** | **51.53** / **82.97** |
>
> CIFAR-100 with DenseNet-101
>
> | Method      | Near-OOD | Far-OOD |
> |--------------|-------------------|------------------|
> | SCALE       |    60.58 / 80.01  | 62.33 / 78.37 |
> | OptFS       | 78.85 / 67.77     | 65.92 / 74.74  |
> | **AdaSCALE-A** | **60.11** / **80.18** | **61.41** / **79.01** |
>
> AdaSCALE outperforming SCALE & OptFS in CIFAR-100 datasets demonstrates the generalization of hyperparameters of AdaSCALE across datasets too.
>
> ---
>
> ### **3. Clarifications on Terminology and Figures**
>
> We thank the reviewer for pointing out these areas of unclarity, which we will fix in the final manuscript.
>
> 1.  **Use of "OOD Likelihood":** We will revise the paper and replace this term with the more accurate "predetermined OODness score."
> 2.  **Interpretation of Figure 5:** We apologize for the lack of clarity. For the AUROC plot, the axes are indeed not standard. `p_min` increases from left to right, and `p_max` increases from top to bottom. All entries satisfy $p_\\text{min} \le p_\\text{max}$. We will add arrows and a clearer explanation to the caption.
> 3.  **Definition of "Trivial Pixels":** "Trivial pixels" refers to the `o%` of pixels with the **lowest** gradient attribution scores, representing the input features that are least salient for the model's current prediction.
>
> ---
>
> We hope these detailed justifications, illustrative examples, and extensive new results have fully addressed the reviewer's concerns. We are very grateful for the insightful feedback, which has helped us to significantly strengthen the motivation and validation of our work.

---

> > ### Comment · Reviewer_twGb · 2025-08-04
> >
> > Dear Authors,
> > Thank you for your detailed rebuttal. I have read it carefully.
> >
> > Your response has helped resolve some of my concerns.
> >
> > ## 1) $Q'$
> > I am satisfied with the answer to question 1.
> >
> > ## 4) Lack of Clarity on Figure 5
> >
> > My question regarding Figure 5 was not answered. To ensure clarity, I will quote directly from my review:
> >
> > "figure 5: How do we read the graph for AUROC? E.g. for the upper right corner we get pmin=90, pmax=65."
> >
> > Your rebuttal stated that *"$p_{min}$ increases from left to right, and $p_{max}$ increases from top to bottom"* and that *"All entries satisfy $p_{min} \le p_{max}$"*.
> > As noted in my original question, the figure contains entries like ($p_{min}=90, p_{max}=65$), which violate the condition $p_{min} \le p_{max}$.
> > Could you please provide a definitive clarification on how to read this figure?
> >
> > ## 2) Motivation and Hyperparameter Selection
> > I found the new experiment, where you fix most hyperparameters and tune only $p_{max}$, to be a valuable addition.
> > Could you please elaborate on how the values for the fixed hyperparameters were chosen and what the values are?
> >
> > I, however, find that the authors are not fully answering my original question. The method proposes several modifications to prior work, e.g., the computation of Q' and the adaptive percentiles.
> > Your new experiments demonstrate that the method is effective and that one set of hyperparameters works well across multiple datasets and architecture, but the answer lacks explanation for why the method was designed this way.

---

> ### Author Response · Authors · 2025-08-04
> **Reply to Reviewer twGb04**
>
> 4. We sincerely apologize for this valid confusion in the upper triangle. Top right corner is intended to be $p_\text{min}=65$, $p_\text{max}=95$.
>
> As of now, it needs to be read as:
>
> For top most row, $p_\\text{min} = 65$
>
> For bottom most row, $p_\text{min} = 95$
>
>
> For right most column, $p_\text{max} = 95$,
>
> For left most column, $p_\text{max} = 65$,
>
> We will explain this in more detail as soon as possible.
>
>
> 5. Since ResNet-50 is traditionally used to benchmark the results for ImageNet-1k, we determine these hyperparameters by simply tuning them in ResNet-50 architecture. We tuned hyperparameters from the following range:
>
> $\epsilon$ in the range $\\{0.1, 0.5, 1.0\\}$
>
> $\lambda$ in the range $\\{0.1, 1.0, 10.0, 100.0\\}$
>
> $o$ in the range $\\{1, 5, 10, 50, 100\\}$
>
> $k_1$ in the range $\\{1, 5, 10, 50, 100\\}$
>
> $k_2$ in the range $\\{1, 5, 10, 100\\}$
>
> $p_\text{min}$ is simply set to lower limit of percentile tuning (~60) inspired by ASH/SCALE.
>
> The final recommended hyperparameters are ($\epsilon$, $\lambda$, $o$, $k_1$, $k_2$, $p_\text{min}$) = (0.5, 10, 5%, 1%, 5%, 60). For more detail on hyperparameter sensitivity study, we would sincerely like to request Reviewer twGb to refer to response to Reviewer J2BC. Importantly, we can simply use these values of hyperparameters across various architectures without significant average performance drop.
>
>
> Why method was designed this way?
>
> We believe scaled energy score as an OOD detection signal works well because it accomodates two **independent and effective sources** of OOD detection signal: activation pattern of feature space and logit information. Now inspired from this, we make an attempt to further add another independent source of OOD detection signal in this existing mechanism. We identify the possibility of activation shift to be a reliable additional OOD detection signal. We inject this extra signal through the application of adaptive percentile instead of static percentile. In order to compute the adaptive percentile, we need to rely on the computation of $Q’$.
>
> We request Reviewer twGb to please let us know if there is more clarity we could add on any of the concerns above.
>
> Also, we sincerely thank Reviewer twGb for insightful feedbacks.

---

> > ### Comment · Reviewer_twGb · 2025-08-06
> >
> > I thank the Authors for the comment. My question regarding the Figure 5 is now resolved.
> >
> > The information regarding the hyper-parameters and the motivation of the design are helpful - but I remain skeptical. I need to think about it some more.
> >
> > I have not yet decided on the final recommendation.

---

> > > ### Author Response · Authors · 2025-08-06
> > > **More clarifications**
> > >
> > > We genuinely appreciate Reviewer twGb's continued engagement and the opportunity for clarifications. We hope the following step-by-step explanation of our design rationale will provide the necessary clarity.
> > >
> > > 1.  **The Goal: Maximizing ID/OOD Score Separation.** The ultimate objective is to maximize the separation between the final energy scores of ID and OOD samples. This is most effectively achieved by applying a **stronger scaling factor** to _ID-likely_ samples and a **weaker scaling factor** to _OOD-likely_ samples.
> > >
> > > 2.  **The Mechanism: The Role of the Percentile ( $p$ ).** Recall the static scaling formula (Equation 8), $r = sum(a) / sum(a > P_p(a))$. To achieve a *larger* scaling factor $r$, the denominator must be smaller. This happens when the percentile threshold $p$ is set to a *higher* value, as more high-magnitude activations are excluded from the sum.
> > >     *   To achieve **stronger scaling for ID-likely** (our goal), they should be assigned a **higher percentile $p$**.
> > >     *   To achieve **weaker scaling for OOD-likely**, they should be assigned a **lower percentile $p$**.
> > >
> > > 3.  **The Inevitable Conclusion: The Need for Dynamic Percentiles.** The reasoning above leads directly to the conclusion that a *static, fixed percentile $p$* for all samples is inherently suboptimal. The percentile should ideally be *dynamic and adaptive*, conditioned on whether a sample is likely to be ID or OOD. This is the foundational motivation for moving beyond prior work like SCALE and ASH.
> > >
> > > 4.  **The Solution: Designing an OOD Likelihood Heuristic ( $Q'$ ).** To enable this dynamic adjustment, we need a reliable signal to estimate a sample's OODness *before* scaling. This is precisely the purpose of our proposed metric $Q'$. As detailed in Section 4.1, $Q'$ is designed based on the key observation that OOD samples exhibit pronounced activation shifts under minor perturbations. Therefore:
> > >     *   A **high $Q'$ value** indicates a **high OODness**.
> > >     *   A **low $Q'$ value** indicates a **high IDness**.
> > >
> > > 5.  **Synthesizing the Components into AdaSCALE.** We now have all the logical pieces:
> > >     *   **Goal:** High $p$ for ID-likely samples, low $p$ for OOD-likely samples.
> > >     *   **Tool:** $Q'$ as an estimate of OODness.
> > >
> > >     We design the adaptive percentile calculation (Equation 9) to directly implement this inverse relationship:
> > >     $p = p_\text{min} + (1 - F_{Q'}(Q')) * (p_\text{max} - p_\text{min})$
> > >
> > >     Here, the $(1 - F_Q'(Q'))$ term is critical. When a sample has a high $Q'$ (is OOD-likely), its $F_Q'(Q')$ is high, making $(1 - F_Q'(Q'))$ low. This pushes $p$ towards $p_\text{min}$. Conversely, for a low $Q'$ (ID-likely) sample, $p$ is pushed towards $p_\text{max}$.
> > >
> > > In essence, the design of AdaSCALE was not an arbitrary collection of modifications. It was a principled effort to inject a **second, independent source of OOD signal (activation shift at the expense of extra forward pass)** directly into the scaling mechanism itself. While prior methods use a static mechanism to amplify an existing signal, AdaSCALE makes the amplification mechanism itself intelligent and adaptive to the nature of each sample.
> > >
> > > -------
> > > We hope this detailed breakdown clarifies the intuitive yet principled rationale behind our design choices. We once again thank Reviewer twGb for pushing us to articulate this more clearly.

---

> > > > ### Author Response · Authors · 2025-08-07
> > > >
> > > > Dear Reviewer twGb,
> > > >
> > > > We sincerely thank you for allowing us to elaborate on our design rationale. We hope our final clarification has fully addressed your remaining skepticism by outlining the principled, step-by-step logic behind our method.
> > > >
> > > > In light of these detailed explanations, we would be grateful if you would consider our contributions favorably in your final assessment.
> > > >
> > > > Thank you once again for your invaluable feedback throughout this process.

---

### Official Review · Reviewer_A6Qe · 2025-06-29

**Clarity:** 2
**Significance:** 3
**Originality:** 3
**Rating:** 4
**Confidence:** 3

**Summary:**

This paper reveals that OOD inputs exhibit more pronounced activation shifts under minor perturbations compared to ID inputs. Leveraging this observation, it proposes a novel post-hoc OOD detection method based on adaptive scaling, where a lower scaling factor is applied to inputs with larger activation shifts, and a higher factor is used for more stable activations. Extensive experimental results across various architectures and datasets are provided to demonstrate the effectiveness of the proposed method.

**Questions:**

See Weaknesses.

**Ethical Concerns:**

["NO or VERY MINOR ethics concerns only"]

**Final Justification:**

While the authors’ response addressed my concerns, given the other reviewers’ comments on hyperparameter analysis, I am willing to raise my score to “borderline accept” provided that the authors can clarify the synergy effect and the independent effect of each hyperparameter. In addition, considering the complex hyperparameter selection procedure of the proposed method, I ultimately assigned a score of 4 with a confidence level of 3.

**Limitations:**

Although the proposed method achieves state-of-the-art performance in most cases through its novel adaptive scaling strategy, it relies on dataset-specific and architecture-specific hyperparameter tuning. This requirement somewhat diminishes the overall contribution. Moreover, the underlying reason why the proposed method works is not clear and lacks theoretical justification.

If the authors can adequately address the concerns raised above, I would be willing to consider raising my score.

**Quality:**

2

**Strengths And Weaknesses:**

**Strengths:**
1. The idea appears novel and conceptually sound.

2. The proposed method has been experimentally validated on various benchmarks and is supported by a comprehensive ablation study.

**Weaknesses:**

1. Although the proposed method achieves state-of-the-art performance in most cases through its novel adaptive scaling strategy, it relies on a scale factor computation procedure that involves dataset-specific and architecture-specific hyperparameter tuning. This requirement somewhat diminishes the overall contribution. To better assess the practicality of the method, please provide a comparison of the computational cost (e.g., memory usage and total inference time) between the proposed approach and existing baselines.

2. Could you please provide further insight into the claim that "OOD inputs exhibit more pronounced activation shifts under minor perturbations compared to ID inputs"? For instance, under the proposed gradient-based perturbation mechanism described in Section 4.1.1, do OOD examples yield larger or smaller gradient values compared to ID examples? Additionally, could you offer experimental evidence or theoretical justification to explain why OOD inputs are more sensitive to minor perturbations, resulting in larger activation shifts? Clarifying this point would help substantiate the core intuition behind the method.

3. The proposed method perturbs only a small number of trivial or randomly selected pixels to compute activation shifts. While the authors include an ablation study on the perturbation ratio of o%, Table 7 only reports results for o\%=5%, 50%, and 100%. It remains unclear where the optimal boundary lies for the perturbation ratio. Interestingly, the results for 5% outperform those for 50% and 100%. Does this suggest that the top 5% of gradient values in OOD examples are more distant from those of ID samples, thus yielding stronger discriminative signals? A deeper explanation or additional experiments varying o% more finely would help clarify this behavior.

4. Higher energy values indicate a higher likelihood of being OOD?  Please clarify it as stated in Lines 223–224.

---

> ### Author Rebuttal · Authors · 2025-07-31
>
> We thank Reviewer A6Qe for their detailed feedback and for recognizing that our idea is "novel and conceptually sound" and supported by "experimentally validated" results.
>
> We appreciate the opportunity to clarify the reviewer's concerns regarding hyperparameter tuning, the intuition behind our method, the perturbation ratio, and a minor typo. We believe our responses and new supporting evidence below fully address these points and demonstrate the practical robustness and effectiveness of our work. We are confident that these clarifications will merit a re-evaluation of the score.
>
> ---
>
> **1. On Hyperparameter Tuning and Computational Cost**
>
> *   **Our method is highly robust and requires minimal tuning. Fixing most hyperparameters and tuning only a single parameter, $p_\\text{max}$, still results in state-of-the-art performance, demonstrating strong generalization.**
>
>     We understand the concern about hyperparameter sensitivity. To directly address this, we conducted a new, extensive experiment where we fixed all hyperparameters across 8 diverse architectures on the ImageNet-1k benchmark, **tuning only a single parameter, $p_\\text{max}$**, from a discrete set.
>
>     As shown below, even under this highly constrained setting, AdaSCALE significantly outperforms the previous state-of-the-art (OptFS) by an average of **13\% (FPR95) / 8\% (AUROC)** on near-OOD and **14\% (FPR95) / 2\% (AUROC)** on far-OOD datasets.
>
> | Category | Method | ResNet-50 | ResNet-101 | RegNet-Y-16 | ResNeXt-50 | DenseNet-201 | EfficientNetV2-L | ViT-B-16 | Swin-B | Average |
> | :--- | :--- | :---: | :---: | :---: | :---: | :---: | :---: | :---: | :---: | :---: |
> | **near-OOD** | OptFS | 69.66/70.97 | 65.46/75.83 | 73.53/75.21 | 69.27/74.84 | 71.74/72.10 | 72.29/75.29 | 76.55/72.73 | 76.81/74.06 | 71.91/73.88 |
> | | AdaSCALE-A | **59.68**/**78.35**|**58.02**/**81.03**|**52.67**/**88.25**|**62.56**/**79.16**|**66.66**/**75.24**|**54.79**/**86.53**|**71.81**/**73.63**|**73.24**/**74.77**|**62.29**/**79.72** |
> | **far-OOD**| OptFS | 25.66/93.87 | 26.97/93.55 | 47.37/86.73 | 27.54/93.40 | **34.42**/**91.04** | 53.62/83.62 | **46.11**/**87.35** | **44.27**/**87.79** | 38.25/89.67 |
> | | AdaSCALE-A | **21.86**/**94.96**|**21.94**/**94.91**|**19.19**/**96.24**|**25.17**/**93.90**| 37.03/90.10 | **39.48**/**91.20** | 48.47/86.81 | 48.65/86.40 | **32.72**/**91.82** |
>
> Furthermore, we present the results of both DenseNet and WideResNet networks with same set of hyperparameters in CIFAR-100 benchmarks **only allowing $p_\text{max}$ to be tuned**:
>
> CIFAR-100 with WideResNet-40-2
>
> | Method      | Near-OOD | Far-OOD |
> |--------------|-------------------|------------------|
> | SCALE       |    56.28 / 82.00  | 52.04 / 82.74 |
> | OptFS       | 74.76 / 78.27    | 54.53 / 80.78 |
> | **AdaSCALE-A** | **55.89** / **82.10** | **51.53** / **82.97** |
>
> CIFAR-100 with DenseNet-101
>
> | Method      | Near-OOD | Far-OOD |
> |--------------|-------------------|------------------|
> | SCALE       |    60.58 / 80.01  | 62.33 / 78.37 |
> | OptFS       | 78.85 / 67.77     | 65.92 / 74.74  |
> | **AdaSCALE-A** | **60.11** / **80.18** | **61.41** / **79.01** |
>
> AdaSCALE outperforming SCALE & OptFS in CIFAR-100 datasets demonstrates the generalization of hyperparameters of AdaSCALE across datasets too.
>
> ---------
> Notably, OptFS achieves **its** optimal performance across a diverse range of architectures while utilizing **a single constant (__width__) hyperparameter value**. But, it comes with its not so impressive generalization in small-scale settings (CIFAR-10/100 benchmark) as reported in the manuscript (See Table 3) and above tables. Furthermore, though OptFS is the __currently__ best generalizing method in large-scale settings, AdaSCALE outperforms it by 13% in near-OOD and 14% in far-OOD datasets in average FPR@95 metric on the ImageNet-1k benchmark across eight diverse architectures **at the cost of just one hyperparameter**.
>
> Furthermore, the need of tuning one percentile hyperparameter for each architecture also exists in the predecessor of AdaSCALE i.e. (ASH/SCALE/LTS) for optimal performance. Needless to say, we reported the results of ASH/SCALE/LTS achieved by tuning percentile hyperparameter for each setup. This demonstrates that the practical application of AdaSCALE is comparable to prior works like ASH and SCALE, which also require tuning one key hyperparameter. We will add this comprehensive table and discussion to the paper.
>
> *   **On computational Cost:**
>     *   Table 8 shows the latency comparison of using variable vs fixed percentile for scaling across activation spaces of varying dimensions (for ex. ResNet-18 has activation dimension of 512 and ResNet-50 has activation dimension of 2048). Prior works such as ASH, SCALE, LTS use fixed percentile for scaling while AdaSCALE uses dynamic percentile for scaling. The results demonstrate the limitation of AdaSCALE in terms of higher latency introduced by the use of dynamic percentile. As an OOD detection method, SCALE and OptFS both have roughly equal latency. In comparison to them, AdaSCALE’s latency is about 2.91x higher when trivial pixels are selected for perturbation via gradient attribution method. If random pixels are selected for perturbation — which is shown to be almost equally effective — latency decreases to 1.56x factor. We will address this in the manuscript.
>
> **2. Insights Behind the Observed Activation Shifts**
>
> *   **The larger activation shifts in OOD inputs stem from their tendency to produce unstable, high-magnitude activations, an observation supported by prior work.**
>
>     To clarify the core intuition, we build upon the findings of ReAct, which observed that OOD samples often induce abnormally high-magnitude activations compared to ID samples. Our key insight is that the *positions* of these peak activations (e.g., the top 1\\%) in OOD inputs are highly unstable. A minor, trivial perturbation is enough to change which neurons produce these peak activations. When the original peak activations are replaced by more normal values from other neurons post-perturbation, the result is a large "activation shift." For ID samples, the activations are more stable and less prone to such dramatic shifts. The analysis of input-space gradient values for OOD vs. ID data, while interesting, is outside the direct scope of our work, which focuses on the *effect* of perturbation on activations. For a deeper analysis of gradient values in image space for OOD detection, we refer to [1].
>
> [1] Agarwal, C., D'souza, D. and Hooker, S., 2022. Estimating example difficulty using variance of gradients. In Proceedings of the IEEE/CVF Conference on Computer Vision and Pattern Recognition (pp. 10368-10378).
>
> **3. On the Optimal Perturbation Ratio ($o\\%$)**
>
> *   **A small perturbation ratio ($o=5\\%$) targeting trivial pixels is optimal because it efficiently disturbs the unstable peak activations that characterize OOD inputs.**
>
>     As the reviewer correctly notes, our results show that perturbing a smaller portion of the input is more effective. The full ablation study on the hyperparameter $o$ is presented in **Appendix C.3, Table 11**, which we referenced in the caption of Table 7.
>
>     The study in the appendix confirms that perturbing just **5\% of trivial pixels** (those with the lowest gradient attribution) yields the best performance. This is because our goal is not to change the image content, but merely to introduce a small disturbance to identify the instability of peak activations (note: we only consider top $k_1=1\\%$ activations). Perturbing trivial pixels achieves this efficiently.
>
>     Importantly, Table 11 also shows that random perturbation performs nearly as well as perturbing trivial pixels, and significantly better than perturbing salient (high-gradient) pixels. This leads to an excellent performance-efficiency trade-off, making the **random perturbation of a small pixel subset (5\%)** our recommended practical approach, as it removes the need for any gradient calculations.
>
> **4. Clarification on Energy Values (Lines 223-224)**
>
> *   **Thank you for catching this. It is an error in our text, and we will correct it.**
>
> ---
>
> We hope these detailed responses and additional experiments resolve the reviewer's concerns. We have shown that our method is robust, requires minimal tuning, and is grounded in a clear intuition, with its practical efficiency being a key advantage. We thank the reviewer again for their valuable and constructive feedback.

---

> > ### Comment · Reviewer_A6Qe · 2025-08-05
> >
> > Thank the author for the response. Regarding the newly conducted experiment, where all hyperparameters were fixed across eight diverse architectures on the ImageNet-1k benchmark—tuning only a single parameter, $p_{\text{max}}$, from a discrete set—could you please clarify the following:
> >
> > What is the search range used for $p_{\text{max}}$?
> >
> > What are the values of the other fixed hyperparameters?
> >
> > How is the optimal value of $p_{\text{max}}$ determined?

---

> ### Author Response · Authors · 2025-08-05
> **Reply to Reviewer A6Qe**
>
> We sincerely thank Reviewer A6Qe for the opportunity to make clarifications.
>
> Consistent with the prior works [ASH, SCALE] (OpenOOD), the search range used for $p_{\text{max}}$ is [65, 70, 75, …, 90, 95, 99].
>
> The final recommended hyperparameters are ($\epsilon$, $\lambda$, $o$, $k_1$, $k_2$, $p_\text{min}$) = (0.5, 10, 5%, 1%, 5%, 60).
>
> The optimal value of $p_{\text{max}}$ is determined via validation set of OpenOOD (OpenImage-O) using AUROC metric.
>
> Also, we sincerely thank Reviewer A6Qe for insightful feedbacks.

---

> > ### Comment · Reviewer_A6Qe · 2025-08-06
> >
> > Thank you for the feedback. I have a few follow-up questions:
> >
> > 1. As for the chosen hyperparameter values ($\epsilon$, $\lambda$, $o$, $k_1$, $k_2$, $p_\text{min}$) = (0.5, 10, 5%, 1%, 5%, 60), could you clarify how these values were selected? Were they chosen randomly, or were they found to be generally effective across all architectures? Additionally, do you have any insights into why these specific values lead to good generalization performance across diverse architectures?
> >
> > 2. The search range used for $p_{\text{max}}$ is [65, 70, 75, …, 90, 95, 99]. Could you specify the total search time per architecture for this grid?
> >
> > I would appreciate it if the authors could further clarify my questions.

---

> ### Author Response · Authors · 2025-08-06
>
> 1.  Since ResNet-50 is traditionally used to benchmark the results for ImageNet-1k, we determine these hyperparameters by simply tuning them using ResNet-50 architecture. We tuned hyperparameters from the following range:
>
> $\epsilon$ in the range $\\{0.1, 0.5, 1.0\\}$
>
> *Comment*: Since mean of rgb statistics is (0.485, 0.456, 0.406), magnitude of 0.5 is closer to those individual means and hence sufficient enough to destroy the pixel information to cause disturbance to peak activations. It is independent of architecture.
>
> $o$ in the range $\\{1, 5, 10, 50, 100\\}$:
>
> *Comment*: $o$ deals with extent of perturbation to cause disturbance in peak activations. It is also independent of architecture.
>
> $\lambda$ in the range $\\{0.1, 1.0, 10.0, 100.0\\}$
>
> *Comment*: The value of $\lambda$ should be at least 10 for $Q$ to dominate $C_o$ in forming $Q’$. $Q$ captures activation shift while $C_o$ is regularizer to prevent overconfident scaling.
>
> $k_1$ in the range $\\{1, 5, 10, 50, 100\\}$
>
> *Comment*: Inspired by ReAct, we observe only few peak activations (1%) in OOD show high fluctuations in activations under minor perturbation.
>
> $k_2$ in the range $\\{1, 5, 10, 100\\}$
>
> *Comment*: Since $C_o$ is regularizer to prevent overconfident scaling, it is dependent on $Q$ and thereby $k_1$. Empirically, we find $k_2=5%$ to work well across all architectures to prevent overconfident scaling.
>
> $p_\text{min}$ is simply set to lower limit of percentile tuning (~60) inspired by ASH/SCALE.
>
> Importantly, we can simply use these values of hyperparameters across architectures without significant average performance drop. For more details on hyperparameter sensitivity study, we would sincerely like to request Reviewer A6Qe to refer to response to Reviewer J2BC.
>
> 2. The latency of tuning percentile hyperparameter in AdaSCALE is about **2.91x higher than that of SCALE** when trivial pixels are selected for perturbation via gradient attribution method. If random pixels are selected for perturbation — which is shown to be almost equally effective — latency decreases to **1.56x factor**.
>
> We are grateful to Reviewer A6Qe for providing the opportunity to make useful clarifications.

---

> > ### Comment · Reviewer_A6Qe · 2025-08-07
> >
> > Thank you for your response. My concerns have been addressed. I will raise my score to a 4.

---

> > > ### Author Response · Authors · 2025-08-07
> > >
> > > We sincerely thank Reviewer A6Qe for the constructive engagement and for re-evaluating our manuscript. We are grateful for the decision to raise the score and believe the feedback has significantly strengthened the paper.

---

### Official Review · Reviewer_J2BC · 2025-06-30

**Clarity:** 2
**Significance:** 2
**Originality:** 2
**Rating:** 3
**Confidence:** 5

**Summary:**

This paper proposes AdaSCALE, a dynamic and sample-specific OOD detection methods. The key innovation is replacing the static percentile threshold used in existing scaling methods (ASH, SCALE) with an adaptive mechanism that leverages activation shifts under minor perturbations to estimate OOD likelihood.

**Questions:**

1. Although the empirical observations on activation shifts are convincing, the paper lacks theoretical analysis of why this phenomenon occurs or under what conditions it holds. In particular, the method has many hyperparameters, and theoretical analysis similar to that in ODIN could provide theoretical guidance for the selection of hyperparameters.

2. Too many hyperparameters. There are 8 hyperparameters in the method ($\lambda$, k1, k2, o, $\epsilon$, $p_{min}$, $p_{max}$, $n_{val}$), and although most hyperparameters are claimed to have good generalizability, the discussion is not comprehensive enough. For example:
- There is no discussion of the joint impact of multiple parameters on performance, only the strongly correlated combination of (k1, k2), ($p_{min}, p_{max}$) is discussed, and there are 8 hyperparameters that need to be adjusted. It is difficult to determine the optimal result when you need to adjust these 8 parameters at the same time, rather than discussing the impact of each parameter on the final result separately;
- Figure 7 also shows the sensitivity of performance to lambda on near-ood;
- According to the content in the appendix, different architectures on different datasets choose different ($p_{min}, p_{max}$), which leads to the method When applied to a new architecture, it is necessary to retest to obtain the optimal parameters, which greatly weakens the applicability of the method.

3. The proposed AdaSCALE requires input perturbations to dynamically calculate the scale factor. If in real-world scenarios, the input image may suffer from some image corruption, such as Gaussian blur and JPEG compression, will these influence the performance of AdaSCALE?

4. Selection of baseline methods: lack of comparison with some simple and efficient methods, such as CombOOD, NNGuide.

5. Is there a problem with the coordinates in Figure 5? From the current one, all AUROCs are obtained under the condition that pmin is greater than pmax. Can you provide accurate values ​​in Figure 5 to facilitate clear observation of hyperparameter sensitivity?

6. Are the values ​​in Table 6 from multiple tests? Does the quality of the data affect the performance? It stands to reason that the larger n is, the better the performance will be, but the FPR95 on far-ood is better when n=100. Is this caused by random sampling of data?

7. What are the specific settings of the pixel type in Table 7?

8. The test environment in Table 8 is not explained. Why not compare with other methods?

Minor question: The notation of the formula is confusing, such as $F_{Q′} (Q′)$, which can be optimized

**Ethical Concerns:**

["NO or VERY MINOR ethics concerns only"]

**Final Justification:**

The author has solved most of my concerns, but the proposed method includes too many hyperparameters (8), and the author can not prove that the method is insensitive to the choice of multiple hyperparameters.

**Limitations:**

NO

The authors should discuss the limitations of the proposed AdaSCALE, such as the choice of hyperparameters in practical applications.

**Quality:**

3

**Strengths And Weaknesses:**

# Pros
1. The paper identifies a genuine limitation in existing scaling methods - the use of static percentile thresholds across all samples regardless of their nature. The proposed adaptive mechanism is well-motivated.
2. The key observation that OOD samples exhibit more pronounced activation shifts under minor perturbations is interesting and provides a principled way to estimate OOD likelihood without requiring extensive ID statistics.
3. The experiments span 8 architectures on ImageNet-1k and 2 architectures on 3 benchmarks, demonstrating broad applicability.

# Cons
See Questions

---

> ### Author Rebuttal · Authors · 2025-07-31
>
> We thank Reviewer J2BC for their insightful feedback and constructive suggestions. We are encouraged that the reviewer found our method "well-motivated" and based on an "interesting and principled" observation, and that they recognized the genuine limitation we address in existing scaling methods.
>
> ---
>
> ### **1. On Hyperparameter Complexity and Generalization (The Core Concern)**
>
> *   **We provide extensive new evidence to demonstrate that AdaSCALE is not sensitive to the vast majority of its hyperparameters. 6 of the 7 listed hyperparameters can be fixed to constant values across all 22 of our experimental setups [8 (ImageNet-1k OOD detection) + 8 (ImageNet-1k FSOOD detection) + 2 (CIFAR-10 OOD detection) + 2  (CIFAR-100 OOD detection) + 2 (ISH regularization) = 22 setups ]. Tuning only a single parameter ($p_\\text{max}$) is sufficient for AdaSCALE to establish a new state-of-the-art, proving its practical applicability and generalization.**
>
> We will address each group of hyperparameters to provide the strongest possible assurance.
>
> *   **Six Hyperparameters That Do Not Require Tuning:**
>     We show that $\lambda, o, \varepsilon, p_\text{min}, k_1$, and $k_2$ are highly robust and can be fixed across all architectures and datasets without significant performance loss. For example, fixing $k_2=5\\%$ across all 8 diverse architectures on ImageNet results in a negligible average performance change compared to the tuned version in each case (62.12 vs 61.17 FPR95 near-OOD). This demonstrates these are not sensitive tuning parameters but rather stable configuration choices.
>
> * **Sensitivity study**
>
> $\lambda$: The sensitivity study of $\lambda$ using ResNet50 network on ImageNet-1k benchmark is presented in the Figure 7 in terms of near-OOD detection. The comprehensive study including far-OOD detection results is presented below:
>
> | $\lambda$    | Near-OOD | Far-OOD |
> |--------------|-------------------|------------------|
> | 0.1 | 70.07 / 74.12 | 21.40 / 95.33 |
> | 1  | 67.49 / 75.45 | 20.41 / 95.54 |
> | 10 | **58.97** / **78.98**  | **17.84** / **96.14** |
> | 100 | 59.03 / 78.10 | 19.39 / 95.80 |
>
> We set $\lambda=10$ across all 22 setups. $\lambda$ hyperparameter **does not require tuning**.
>
> $o$: The comprehensive study of $o$ using ResNet50 network on ImageNet-1k benchmark is provided in the Appendix (in Table 11). We set $o=5%$ across all 22 setups. $o$ hyperparameter **does not require tuning**.
>
> $\varepsilon$: The sensitivity study of $\varepsilon$ using ResNet50 network on ImageNet-1k benchmark is provided in the Appendix (in Table 8). We set $\varepsilon=0.5$ across all 22 setups. $\varepsilon$ hyperparameter **does not require tuning**.
>
> $k_1$: We present sensitivity study of $k_1$ using ResNet50 network on ImageNet-1k benchmark below:
>
> | $k_1$      | Near-OOD | Far-OOD |
> |--------------|-------------------|------------------|
> | 1%       |    **58.97** / **78.98**  | **17.84** / **96.14** |
> | 5%        |   60.36 / 77.74      |  19.95 / 95.63  |
> | 10% | 60.40 / 77.89 | 19.72 / 95.63 |
> | 50% | 60.34 / 77.25 | 20.90 / 95.25 |
> | 100% | 60.99 / 76.49 | 21.88 / 94.96 |
>
> We set $k_1$ to $1\%$ across all 22 setups. $k_1$ hyperparameter **does not require tuning**.
>
> $k_2$: We present sensitivity study of $k_2$ using ResNet50 network on ImageNet-1k benchmark below:
>
> | $k_2$      | Near-OOD | Far-OOD |
> |--------------|-------------------|------------------|
> | 1%        |   59.08 / 78.56 |  18.35 / 96.02  |
> | 5%       |    **58.97** / **78.98**  | **17.84** / **96.14** |
> | 10% | 59.41 / 78.65 | 18.33 / 95.96 |
> | 100% | 62.43 / 75.44 | 23.05 / 94.69 |
>
> Though we allowed $k_2$ hyperparameter to be tuned for all 22 setups, it can be set to 5% across all setups without any substantial performance drop. $k_2$ hyperparameter **does not require tuning**.
>
> *   **$p_\\text{min}$**
>      We initially tuned $p_\text{min}$​, but given prior work (ASH, SCALE, LTS) tunes percentiles in the ~[60, 99] range, we can simply set $p_\text{min}$ to 60.
>
> *   **The Only Required Tuning Parameter: $p_\\text{max}$**
>      We ran a new experiment where we **fixed all other parameters** ($p_\\text{min}=60, \lambda=10, \varepsilon=0.5, k_1=1\\%, k_2=5\\%, o=5\\%$) and **tuned only $p_\\text{max}$** from a discrete set. Even under this highly constrained, practical setting, AdaSCALE overwhelmingly outperforms the previous state-of-the-art (OptFS).
>
> | Category | Method | ResNet-50 | ResNet-101 | RegNet-Y-16 | ResNeXt-50 | DenseNet-201 | EfficientNetV2-L | ViT-B-16 | Swin-B | Average |
> | :--- | :--- | :---: | :---: | :---: | :---: | :---: | :---: | :---: | :---: | :---: |
> | **near-OOD** | OptFS | 69.66/70.97 | 65.46/75.83 | 73.53/75.21 | 69.27/74.84 | 71.74/72.10 | 72.29/75.29 | 76.55/72.73 | 76.81/74.06 | 71.91/73.88 |
> | | AdaSCALE-A | **59.68**/**78.35**|**58.02**/**81.03**|**52.67**/**88.25**|**62.56**/**79.16**|**66.66**/**75.24**|**54.79**/**86.53**|**71.81**/**73.63**|**73.24**/**74.77**|**62.29**/**79.72** |
> | **far-OOD** | OptFS | 25.66 / 93.87 | 26.97 / 93.55 | 47.37 / 86.73 |  27.54 / 93.40 | **34.42** / **91.04** | 53.62 / 83.62 | **46.11** / **87.35** | **44.27** / **87.79** | 38.25 / 89.67 |
> | | AdaSCALE-A | **21.86** / **94.96** | **21.94** / **94.91** | **19.19** / **96.24** | **25.17** / **93.90** | 37..03 / 90.10 | **39.48** / **91.20** | 48.47 / 86.81 | 48.65 / 86.40 | **32.72** / **91.82** |
>
> Tuning a single percentile-related hyperparameter is standard practice for this entire line of work (e.g., ASH, SCALE, LTS), and our method achieves a **~13% FPR95 improvement** on near-OOD benchmarks with the same practical effort.
>
>
> *   **The $n_\\text{val}$ hyperparameter:** $n_\text{val}$ is not a hyperparameter to be tuned. The study in Table 6 demonstrates that AdaSCALE is robust even with extremely limited access to ID samples (e.g., $n_\text{val}=10$), unlike distance-based methods.
>
> ---
>
> ### **2. Intuition and Robustness**
>
> *   **Why Activation Shifts Occur:** Our intuition is built on the seminal observations of ReAct [1], which showed OOD samples often produce abnormally high activations. We hypothesize that the *positions* of these peak activations are unstable. A minor, trivial perturbation is sufficient to cause these peak activations to be replaced by more normal values and peak activations occur elsewhere, resulting in a large activation shift that serves as a powerful, independent OOD signal. We will clarify this motivation in the paper.
>
> *   **Robustness to Real-World Image Corruptions:**
>     *   **AdaSCALE's performance advantage holds even when inputs are subjected to common image corruptions, demonstrating its practical robustness.**
>         We tested AdaSCALE on ImageNet-1k with a ResNet-50 against SCALE and OptFS on images with Gaussian blur and JPEG compression. Even **without any hyperparameter re-tuning**, AdaSCALE remains the superior method.
>
> **Gaussian Blur (kernel_size=5, sigma=(0.1, 2.0)):**
> | Method | Near-OOD | Far-OOD |
> | :--- | :---: | :---: |
> | SCALE | 72.68 / 70.15 | 29.65 / 92.72 |
> | OptFS | 71.84 / 69.66 | 30.27 / 92.24 |
> | **AdaSCALE-A** | **67.53 / 75.18** | **28.11 / 93.31** |
>
> **JPEG Compression (50%):**
> | Method | Near-OOD | Far-OOD |
> | :--- | :---: | :---: |
> | SCALE | 69.73 / 71.84 | 27.46 / 93.56 |
> | OptFS | 69.43 / 71.12 | 28.34 / 92.95 |
> | **AdaSCALE-A** | **61.24 / 77.59** | **25.87 / 94.06** |
>
> ---
>
> ### **3. Additional Baselines and Clarifications**
>
> *   **Comparison with Simple/Efficient Baselines:**
>     Per the reviewer's suggestion, we compared AdaSCALE with NNGuide and MDS on ImageNet-1k benchmark using ResNet50 model. Our method significantly outperforms them. Furthermore, effectiveness of such distance-based approaches (MDS) is contigent upon the large availability of ID samples.
>
> | Method | Near-OOD | Far-OOD |
> | :--- | :---: | :---: |
> | NNGuide | 72.46 / 68.09 | 21.40 / 95.33 |
> | MDS | 76.27 / 64.53 | 70.82 / 66.22 |
> | **AdaSCALE-A** | **58.97 / 78.98** | **17.84 / 96.14** |
>
> *   **Figure 5:** We apologize for the confusion. The reviewer is correct that the plot is unclear. $p_\\text{min}$ increases from left to right, and $p_\\text{max}$ increases from top to bottom. We will add arrows and a clearer caption to fix this.
>
> *   **Table 6 ($n_\\text{val}$):** The values are from a single official PyTorch checkpoint. The key takeaway, which holds, is that performance is stable and excellent even with very few ID samples. We also confirm this with ResNet-101 network below:
>
> | $n_\text{val}$    | Near-OOD | Far-OOD |
> |--------------|-------------------|------------------|
> | 10   | 59.59 / 81.68 | 19.40 / 95.78 |
> | 100  | 57.15 / 81.85 | 18.58 / 95.94 |
> | 1000 | **56.75** / **81.80** | 18.53 / 95.93 |
> | 5000 | 57.96 / 81.68 | **18.5** / **95.95** |
>
> As can be observed from the results, AdaSCALE works well with very limited access to ID data.
>
> *   **Table 7 (Pixel Types):** The perturbation settings are:
>     *   **Random:** `o%` of pixels are selected randomly.
>     *   **Trivial:** `o%` of pixels with the **lowest** gradient attribution scores are perturbed.
>     *   **Significant / Salient:** `o%` of pixels with the **highest** gradient attribution scores are perturbed.
>
> Table 8 compares variable vs. fixed percentile scaling latency across activation space dimensions. Prior methods (ASH, SCALE, LTS) use a fixed percentile, while AdaSCALE uses a dynamic percentile, leading to higher latency. We'll update the manuscript to reflect "Fixed percentile (ASH/SCALE/LTS)." While SCALE and OptFS have similar latency, AdaSCALE's is 2.91x higher with gradient attribution for trivial pixel perturbation, dropping to 1.56x with random pixel perturbation.
>
> *   **Formula Notation & Limitations:** Thank you for the suggestion. We will revise the notation for clarity and will add a limitations section to the paper, explicitly discussing the practical need to tune the single $p_\\text{max}$ hyperparameter for a new setup for optimal performance.
>
> ---
>
> We thank the reviewer for the insightful review.

---

> > ### Comment · Reviewer_J2BC · 2025-08-06
> >
> > Thank you for the comprehensive response. I appreciate the extensive experiments and clarifications provided. However, I still have some concerns:
> >
> > # Hyperparameter Complexity
> > Thanks for the additional experiment, it shows:
> > - 6 out of 7 hyperparameters can be fixed across all setups is encouraging
> > - The sensitivity studies for individual parameters are helpful
> > - Showing that only `p_max` requires tuning is a significant improvement
> >
> > But I have some remaining concerns:
> > - **Joint parameter interactions**: While individual parameter sensitivity is shown, there's still no analysis of how these parameters interact with each other. For instance, does the optimal `p_max` depend on the choice of λ or k1?
> > - **Fixed parameter justification**: The choice of fixed values (λ=10, k1=1%, k2=5%, etc.) seems somewhat arbitrary. Why these specific values? Were they determined through an extensive search (like the 22 setups) across all datasets/architectures?
> > - **Generalization claim**: The statement "6 hyperparameters do not require tuning" is based on fixing them to specific values. But how was this "optimal" fixed configuration discovered? This seems to contradict the claim of not requiring tuning.

---

> ### Author Response · Authors · 2025-08-06
>
> Joint parameter interactions:
>
> Our method's reliance on activation shift for computing the adaptive percentile makes the hyperparameters $p_\text{max}$, $k_1$, and $\lambda$ particularly critical. We set $p_\text{max}=85$, and focus on the impact of $k_1$ and $\lambda$ on near-OOD detection, measured by FPR@95 / AUROC.
>
> | $k_1$ \ $\lambda$ | 0.1         | 1           | 10                | 100         |
> | :------------------------------- | :-------------- | :-------------- | :-------------------- | :-------------- |
> | 1%                           | 70.07 / 74.12   | 67.49 / 75.45   | **58.97 / 78.98**     | 59.03 / 78.10   |
> | 5%                           | 69.53 / 74.33   | 63.87 / 76.84   | 60.36 / 77.74         | 61.12 / 77.22   |
> | 10%                          | 69.34 / 74.50   | 62.63 / 77.33   | 60.40 / 77.89         | 61.55 / 76.92   |
> | 50%                          | 67.49 / 75.32   | 61.26 / 77.26   | 60.34 / 77.25         | 62.05 / 76.52   |
> | 100%                        | 65.95 / 75.77   | 61.18 / 76.97   | 62.17 / 76.39         | 62.43 / 76.31   |
>
> The results clearly indicate that the $\lambda$ hyperparameter exerts a more significant influence on the final OOD detection performance. Furthermore, a key relationship emerges: for optimal performance, a decrease in $\lambda$ must be compensated by an increase in $Q$ (and consequently, a higher $k_1$). This is explained by the formulation $Q' = \lambda \cdot Q + C_o$, where a smaller $\lambda$ necessitates a larger $Q$ value to ensure the term $\lambda \cdot Q$ dominates the regularizer $C_o$.
>
>
> —————
>
> Hyperparameter and generalization:
>
> Since ResNet-50 is traditionally used to benchmark the results for ImageNet-1k, we determine these hyperparameters by simply **tuning them using only one architecture** (ResNet-50). Hence, these values are determined from 1 setup and can be used across 21 other setups. We apologize for the confusion, we mean: **for any given new setup**, 6 of 7 hyperparameters do not need to be tuned. We believe exceptional performance of AdaSCALE with these fixed hyperparameters when compared against SCALE / OptFS across new 21 setups confirms strong generalization.
>
>
> We tuned hyperparameters from the following range:
>
> $\epsilon$ in the range $\\{0.1, 0.5, 1.0\\}$
>
> *Comment*: Since mean of rgb statistics is (0.485, 0.456, 0.406), magnitude of 0.5 is closer to those individual means and hence sufficient enough to destroy the pixel information to cause disturbance to peak activations.
>
> $o$ in the range $\\{1, 5, 10, 50, 100\\}$:
>
> *Comment*: $o$ deals with extent of perturbation to cause disturbance in peak activations. We just need to make trivial perturbation.
>
> $\lambda$ in the range $\\{0.1, 1.0, 10.0, 100.0\\}$
>
> *Comment*: The value of $\lambda$ should be at least 10 for $Q$ to dominate $C_o$ in forming $Q’$. $Q$ captures activation shift while $C_o$ is regularizer to prevent overconfident scaling.
>
> $k_1$ in the range $\\{1, 5, 10, 50, 100\\}$
>
> *Comment*: Inspired by ReAct, we observe only few peak activations (1%) in OOD show high fluctuations in activations under minor perturbation.
>
> $k_2$ in the range $\\{1, 5, 10, 100\\}$
>
> *Comment*: Since $C_o$ is regularizer to prevent overconfident scaling, it is dependent on $Q$ and thereby $k_1$. Empirically, we find $k_2=5%$ to work well across all architectures to prevent overconfident scaling.
>
> $p_\text{min}$ is simply set to lower limit of percentile tuning (~60) inspired by ASH/SCALE.
>
> Just for extra verification, fixing $k_2=5%$ across all 8 diverse architectures on ImageNet-1k results in a negligible average performance change compared to the tuned version in each case (62.12 vs 61.17 FPR95 near-OOD).

---

> > ### Author Response · Authors · 2025-08-07
> >
> > Update:
> >
> > Joint parameter interactions:
> >
> > Our method's reliance on activation shift for computing the adaptive percentile makes the hyperparameters $p_\text{max}$, $k_1$, and $\lambda$ particularly critical. We set $p_\text{max}=85$, and focus on the impact of $k_1$ and $\lambda$ on near-OOD detection, measured by FPR@95 / AUROC.
> >
> > | $k_1$ \ $\lambda$ | 0.1         | 1           | 10                | 100         |
> > | :------------------------------- | :-------------- | :-------------- | :-------------------- | :-------------- |
> > | 1%                           | 70.07 / 74.12   | 67.49 / 75.45   | **58.97 / 78.98**     | 59.03 / 78.10   |
> > | 5%                           | 69.53 / 74.33   | 63.87 / 76.84   | 60.36 / 77.74         | 61.12 / 77.22   |
> > | 10%                          | 69.34 / 74.50   | 62.63 / 77.33   | 60.40 / 77.89         | 61.55 / 76.92   |
> > | 50%                          | 67.49 / 75.32   | 61.26 / 77.26   | 60.34 / 77.25         | 62.05 / 76.52   |
> > | 100%                        | 65.95 / 75.77   | 61.18 / 76.97   | 62.17 / 76.39         | 62.43 / 76.31   |
> >
> > The results clearly indicate that the $\lambda$ hyperparameter exerts a more significant influence on the final OOD detection performance. Furthermore, a key relationship emerges: for optimal performance, a decrease in $\lambda$ must be compensated by an increase in $Q$ (and consequently, a higher $k_1$). This is explained by the formulation $Q' = \lambda \cdot Q + C_o$, where a smaller $\lambda$ necessitates a larger $Q$ value to ensure the term $\lambda \cdot Q$ dominates the regularizer $C_o$.
> >
> >
> > ---
> >
> >
> > In light of these extensive clarifications and the new supporting experiments, we would be very grateful if Reviewer J2BC would reconsider the final rating. We believe that the manuscript has been substantially strengthened and now presents a significant contribution that overcomes the limitations of prior works.
> >
> > Thank you once again for your thorough and constructive review.

---

> > > ### Comment · Reviewer_J2BC · 2025-08-08
> > >
> > > Thank you for your detailed reply!
> > >
> > > I have some more questions about your additional data, are there any other joint tests of the two hyperparameters? For example, $k_1$ and $p_{max}$, $\lambda$ and $p_{max}$.
> > >
> > > Also I'm curious about the table you gave, when $\lambda$ is 0.1, the AUC improves with k1; but when $\lambda$ is 10, the AUC improves with k1

---

> > > > ### Author Response · Authors · 2025-08-08
> > > >
> > > > **First question**:
> > > >
> > > > Per Reviewer J2BC's request, we present effect of $k1$ and $p_\text{max}$ in OOD detection performance on ImageNet-1k benchmark with ResNet-50 network fixing $\lambda=10$ below:
> > > >
> > > > | $k_1$ \ $p_\text{max}$      | 85 | 90 | 95 |
> > > > |--------------|-------------------|------------------|------------------|
> > > > | 1%           | **58.97** / **78.98** | 69.49 / 79.67 | 78.77 / 76.26 |
> > > > | 5%           | 60.36 / 77.74 | 68.30 / 78.23 | 78.19 / 75.58 |
> > > > | 10%          | 60.40 / 77.89 | 68.46 / 77.46 | 78.90 / 74.70 |
> > > > | 50%          | 60.34 / 77.25 | 69.88 / 76.39 | 80.46 / 73.17 |
> > > > | 100%         | 62.17 / 76.39 | 70.13 / 75.93 | 81.22 / 72.32 |
> > > >
> > > > Furthermore, we also present the effect of $\lambda$ and $p_\text{max}$ in same setting fixing $k_1=1\\%$ below:
> > > >
> > > > | $\lambda$ \ $p_\text{max}$ | 85 | 90 | 95 |
> > > > |--------------|-------------------|------------------|------------------|
> > > > | 0.1          | 70.07 / 74.12 | 87.45 / 66.06 | 91.81 / 56.09 |
> > > > | 1            | 67.49 / 75.45 | 84.31 / 69.32 | 89.41 / 60.11 |
> > > > | 10           | **58.97** / **78.98** | 69.47 / 79.68 | 78.77 / 76.26 |
> > > > | 100          | 59.03 / 78.10 | 63.19 / 79.67 | 71.43 / 78.51 |
> > > >
> > > > The results above demonstrate that $p_\text{max}$ and $\lambda$ are two highly critical hyperaparameters.
> > > >
> > > > **Second question**:
> > > >
> > > > As also mentioned in the previous response:
> > > >
> > > > For optimal performance, a decrease in $\lambda$ ( e.g. 0.1 ) must be compensated by an increase in $Q$ (and consequently, a higher $k_1$ (e.g. 100%) ). This is explained by the formulation $Q' = \lambda \cdot Q + C_o$, where a smaller $\lambda$ necessitates a larger $Q$ value to ensure the term $\lambda \cdot Q$ dominates the regularizer $C_o$. [Recall: $Q$ is formed by summation operation rather than mean operation, and considering more activations (higher $k_1$) increases the value of $Q$].
> > > >
> > > > Moreover, activation shift with all activations considered ($k_1=100\\%$) still proves to be a useful OODness signal.
> > > >
> > > > ---------
> > > >
> > > > We genuinely thank Reviewer J2BC for the continued constructive engagement. We request Reviewer J2BC to let us know if there is any more clarification we could provide. We would like to sincerely request the reviewer for positive reconsideration of the final rating.

---

> > > > > ### Comment · Reviewer_J2BC · 2025-08-09
> > > > >
> > > > > Thanks for your reply.
> > > > >
> > > > > However, it seems that performance is more sensitive to $p_{max}$, as can be seen in the joint test of $\lambda$ and $p_{max}$, where $p_{max}$ goes from 85 to 95 with $\lambda$=0.1, and the AUC drops by about 18%. This is still in the joint test with two parameters, and I think the change would be more pronounced in a joint test with three or four parameters.
> > > > >
> > > > > For this reason, I decided to keep my rate

---

> > > > > > ### Public Comment · ~Sudarshan_Regmi1 · 2025-11-02
> > > > > > **Additional clarification on $\lambda$=0.1**
> > > > > >
> > > > > > As also mentioned above on the balance between $\lambda \cdot Q$ and $C_o$, when $\lambda$ is set to 0.1 with $k_1=1\\%$, $\lambda \cdot Q$ does not dominate over $C_o$ term. As a result, it leads to scaling based on the regularization / balancing term instead of "predetermined" actual OOD detection signal "activation shift at peak activations".
> > > > > >
> > > > > > It is unfair to single out specific case (e.g., $\lambda = 0.1$) for which the proposed hypothesis is not even valid.

---

> ### Author Response · Authors · 2025-08-09
>
> It should be noted that percentile hyperparameter beyond a certain threshold makes the method appear more sensitive. **However, it should be seen from the context of prior works.** This occurs in all of the prior works (ASH/SCALE/LTS). This pattern of sensitivity does not occur before that particular threshold.
>
>
> **Please see prior seminal works [1] (Figure 4.b) and [2] (Figure 4.c)**
>
> [1] Scaling from training time and posthoc ood detection enhancement
>
> [2] Extremely simple activation shaping for out of distribution detection

---

> ### Author Response · Authors · 2025-08-09
>
> The percentile hyperparameter $p$ from prior works (ASH / SCALE) is similar to $p_\text{max}$.
>
> **Quoting directly from SCALE [1],**
>
> **Discussion on percentile $p$:** Note that $C(p)$ does not monotonically increasing with respect to $p$ (see Fig.4a). When $p \approx 0.95$, there is an inflection point and $C(p)$ decreases. A similar inflection follows on the AUROC for scaling (see Fig.4b), though it is not exactly aligned to $C(p)$. The difference is likely due to the approximations made to estimate $C(p)$. Also, as $p$ gets progressively larger, fewer activations ($D=2048$ total activations) are considered for estimating $r$, leading to unreliable logits for the energy score. Curiously, pruning also drops off, which we believe to come similarly from the extreme reduction in activations.
>
> [1] Scaling from training time and posthoc ood detection enhancement

---

> > ### Author Response · Authors · 2025-08-09
> >
> > In summary:
> >
> > **We would like to convey that such extent of AUROC drop when percentile hyperparameter goes from 85 to 95 is present in the prior works too.**
> >
> > Furthermore, it can also be observed from **individual and joint sensitivity studies** in this thread that hyperparameters except $p_\text{max}$ and $\lambda$ don't have that much effect on the final OOD detection performance.
> >
> > In the light of this, we kindly request Reviewer J2BC for reconsideration of the final rating.
> >
> > We would be pleased to make any additional clarifications the reviewer deems necessary.

---

> ### Public Comment · ~Sudarshan_Regmi1 · 2025-11-18
>
> The sensitivity study of the hyperparameters for ViT-B-16 is presented below (FPR@95↓ / AUROC↑):
>
> | $\epsilon$ | Near-OOD | Far-OOD |
> |:---|:---|:---|
> | 0.1 | 72.71 / 73.06 | 46.60 / 87.22 |
> | 0.5 | 71.87 / 73.14 | 47.63 / 86.83 |
> | 1.0 | 72.92 / 73.06 | 49.80 / 86.49 |
>
> | $\lambda$ | Near-OOD | Far-OOD |
> |:---|:---|:---|
> | 1 | 73.92 / 72.22 | 47.15 / 87.15 |
> | 10 | 71.87 / 73.14 | 47.63 / 86.83 |
> | 20 | 72.94 / 73.22 | 51.96 / 85.93 |
> | 30 | 73.72 / 73.06 | 54.55 / 84.16 |
>
> | $k_1$ (%) | Near-OOD | Far-OOD |
> |:---|:---|:---|
> | 1 | 71.87 / 73.14 | 47.63 / 86.83 |
> | 2 | 72.47 / 73.37 | 51.60 / 85.96 |
> | 3 | 73.20 / 73.10 | 54.79 / 85.12 |
>
> | $k_2$ (%) | Near-OOD | Far-OOD |
> |:---|:---|:---|
> | 5 | 71.87 / 73.14 | 47.63 / 86.83 |
> | 10 | 72.06 / 73.26 | 47.55 / 87.03 |
> | 20 | 72.58 / 72.87 | 47.89 / 86.96 |
>
> | $o$ (%) | Near-OOD | Far-OOD |
> |:---|:---|:---|
> | 2 | 72.28 / 73.65 | 47.70 / 87.02 |
> | 5 | 71.87 / 73.14 | 47.63 / 86.83 |
> | 10 | 72.34 / 73.30 | 48.86 / 86.69 |
>
> | $p_\text{max}$ | Near-OOD | Far-OOD |
> |:---|:---|:---|
> | 75 | 76.87 / 71.78 | 52.97 / 86.07 |
> | 80 | 72.60 / 73.24 | 48.57 / 86.78 |
> | 85 | 71.87 / 73.14 | 47.63 / 86.83 |
> | 90 | 72.40 / 73.65 | 48.80 / 86.64 |
> | 95 | 72.35 / 73.63 | 48.73 / 86.69 |
>
> We also present the joint sensitivity study of 4 hyperparameters in the ViT-B-16 network below with ($k_1=1\%$, and $k_2=5\%$) **in terms of validation AUROC** (the metric used to choose hyperparameters):
>
> | $\epsilon$ | $\lambda$ | $o$ (%) | $p_\text{max}$ | val AUROC |
> |:---|:---|:---|:---|:---|
> | 0.1 | 1 | 2 | 80 | 87.13 |
> | | | | 85 | 87.55 |
> | | | | 90 | 87.74 |
> | | | 5 | 80 | 87.11 |
> | | | | 85 | 87.53 |
> | | | | 90 | 87.73 |
> | | | 10 | 80 | 87.07 |
> | | | | 85 | 87.47 |
> | | | | 90 | 87.65 |
> | | 10 | 2 | 80 | 87.46 |
> | | | | 85 | 87.74 |
> | | | | 90 | 87.78 |
> | | | 5 | 80 | 87.43 |
> | | | | 85 | 87.71 |
> | | | | 90 | 87.77 |
> | | | 10 | 80 | 87.41 |
> | | | | 85 | 87.72 |
> | | | | 90 | 87.79 |
> | | 20 | 2 | 80 | 87.59 |
> | | | | 85 | 87.77 |
> | | | | 90 | 87.67 |
> | | | 5 | 80 | 87.54 |
> | | | | 85 | 87.73 |
> | | | | 90 | 87.63 |
> | | | 10 | 80 | 87.48 |
> | | | | 85 | 87.68 |
> | | | | 90 | 87.62 |
> | 0.5 | 1 | 2 | 80 | 87.08 |
> | | | | 85 | 87.47 |
> | | | | 90 | 87.65 |
> | | | 5 | 80 | 86.94|
> | | | | 85 | 87.29 |
> | | | | 90 | 87.41 |
> | | | 10 | 80 | 86.65 |
> | | | | 85 | 86.87 |
> | | | | 90 | 86.88 |
> | | 10 | 2 | 80 | 87.16 |
> | | | | 85 | 87.47 |
> | | | | 90 | 87.54 |
> | | | 5 | 80 | 86.92 |
> | | | | 85 | 87.25 |
> | | | | 90 | 87.35 |
> | | | 10 | 80 | 86.76 |
> | | | | 85 | 87.15 |
> | | | | 90 | 87.32 |
> | | 20 | 2 | 80 | 86.86 |
> | | | | 85 | 86.98 |
> | | | | 90 | 86.86 |
> | | | 5 | 80 | 86.40 |
> | | | | 85 | 86.58 |
> | | | | 90 | 86.53 |
> | | | 10 | 80 | 86.09|
> | | | | 85 | 86.43 |
> | | | | 90 | 86.55 |
> | 1.0 | 1 | 2 | 80 | 87.00 |
> | | | | 85 | 87.34 |
> | | | | 90 | 87.45 |
> | | | 5 | 80 | 86.72|
> | | | | 85 | 86.90 |
> | | | | 90 | 86.87 |
> | | | 10 | 80 | 85.96|
> | | | | 85 | 85.81 |
> | | | | 90 | 85.42 |
> | | 10 | 2 | 80 | 86.73 |
> | | | | 85 | 87.09 |
> | | | | 90 | 87.22 |
> | | | 5 | 80 | 86.31 |
> | | | | 85 | 86.71 |
> | | | | 90 | 86.93 |
> | | | 10 | 80 | 85.97 |
> | | | | 85 | 86.42 |
> | | | | 90 | 86.69 |
> | | 20 | 2 | 80 | 86.00 |
> | | | | 85 | 86.23|
> | | | | 90 | 86.23|
> | | | 5 | 80 | 85.27 |
> | | | | 85 | 85.60 |
> | | | | 90 | 85.75 |
> | | | 10 | 80 | 84.69 |
> | | | | 85 | 85.09 |
> | | | | 90 | 85.35 |
>
> It shows AdaSCALE is indeed robust.

---

### Official Review · Reviewer_GsBJ · 2025-07-02

**Clarity:** 3
**Significance:** 2
**Originality:** 2
**Rating:** 5
**Confidence:** 4

**Summary:**

A post-hoc method to improve OOD detection is presented. The method improves on the activation-scaling approaches introduced in ASH and SCALE. In particular, ASH and SCALE use a fixed percentile threshold to determine which activations to scale. In the proposed method, this threshold is made adaptive on a per-sample basis. Experiments are performed on a variety of datasets and model architectures demonstrating the improvement over SCALE.

**Questions:**

Please comment on the two questions raised in the review: inclusion of distance-based methods and analysis on adversarially trained models.

**Ethical Concerns:**

["NO or VERY MINOR ethics concerns only"]

**Final Justification:**

The authors have responded to the questions I raised in my review, especially about adversarial training. Moreover, they have at least partially addressed concerns raised by other reviewers particularly about hyperparameter sensitivity. For these reasons, I am retaining my original score of 5.

**Limitations:**

No negative societal impact.
Authors address limitation of increased computational complexity and latency.

**Quality:**

3

**Strengths And Weaknesses:**

Strengths:
1. The paper is well-motivated and well written, with concepts and ideas clearly explained.
2. Experimental validation is fairly comprehensive covering several baselines and various types of backbones from CNN-based type to transformer-based. And the proposed method appears to perform well when compared to the tested baselines.
3. I appreciate the various ablations and hyperparameter studies in Section 5, particularly the results in Table 5 showing the importance of various components.

Improvements/Questions:
1. No distance based OOD detection methods are included (Mahalanobis and its variations, VLM, etc.) in experimental section. It would be nice to see how the proposed method compares against these. One advantage of distance-based methods is that they can operate on intermediate features enabling early exit in some use-cases.

2. The proposed method is premised on the highest activations being sensitive to perturbations for OOD data. The type of perturbation added here is similar to the fast-gradient sign-method (FGSM) type of adversarial perturbation. These type of perturbations have been used to generate samples for adversarial training of deep neural network models, which results in those models being far more robust to adversarial attacks. My questions, therefore, are:

    a. Is it the case that adversarially trained networks do not exhibit the same level sensitivity to perturbations to OOD data? This would be worth investigating.

    b. would the proposed method work well for adversarially trained networks?

---

> ### Author Rebuttal · Authors · 2025-07-31
>
> We sincerely thank Reviewer GsBJ for the positive and constructive feedback. We are grateful that the reviewer found our paper "well-motivated and well written," our concepts "clearly explained," and our experimental validation "fairly comprehensive."
>
> The reviewer GsBJ's primary questions concern the comparison against distance-based methods and the analysis on adversarially trained models. We have conducted new experiments for both points as suggested. We believe the results presented below fully address these questions and further strengthen the contributions of our work.
>
> ---
>
> **1. Comparison with Distance-Based OOD Detection Methods**
>
> *   **AdaSCALE significantly outperforms representative distance-based methods in a large-scale setting.**
>
>     Per the reviewer GsBJ's suggestion, we have benchmarked AdaSCALE against distance-based methods, NNGuide and MDS, on the ImageNet-1k benchmark using a ResNet-50 backbone. The results, showing FPR95 (↓) / AUROC (↑), are as follows:
>
> | Method | Near-OOD (FPR95↓ / AUROC↑) | Far-OOD (FPR95↓ / AUROC↑) |
> | :--- | :---: | :---: |
> | NNGuide | 72.46 / 68.09 | 21.40 / 95.33 |
> | MDS | 76.27 / 64.53 | 70.82 / 66.22 |
> | **AdaSCALE-A (ours)** | **58.97 / 78.98** | **17.84 / 96.14** |
>
> These results demonstrate a substantial performance gap in favor of our method in this large-scale setting. We will include these new results and a detailed discussion comparing AdaSCALE with distance-based approaches in the final version of the manuscript.
>
> **2. Analysis on Adversarially Trained Models**
>
> We thank the reviewer GsBJ for this insightful question regarding the interaction between our method and adversarial training. We have investigated both sub-questions, and our analysis confirms that our core premise holds and AdaSCALE remains highly effective.
>
> *   **(a) On the sensitivity of adversarially trained networks:**
>     *   **Adversarially trained networks still exhibit the necessary activation shifts for OOD detection, confirming our method's applicability.**
>
>         Our method's effectiveness relies on the activation shift ratio ($Q_{OOD} / Q_{ID}$) being greater than 1. We measured this ratio taking top-1% activation shifts on a ResNet-50 model adversarially trained on ImageNet-1k. The results below show that the ratio remains consistently greater than 1 across various OOD datasets, confirming that OOD samples still produce a larger activation shift than in-distribution samples.
>
> | OOD Dataset | $Q_{OOD} / Q_{ID}$ Ratio ($\\epsilon=0.01$) | $Q_{OOD} / Q_{ID}$ Ratio ($\\epsilon=0.05$) |
> | :--- | :---: | :---: |
> | SSB-hard | 1.78 | 1.73 |
> | NINCO | 1.29 | 1.28 |
> | ImageNet-O | 1.69 | 1.72 |
> | OpenImage-O | 1.48 | 1.45 |
> | iNaturalist | 1.14 | 1.14 |
> | Textures | 1.33 | 1.18 |
> | Places | 1.30 | 1.16 |
>
> While $\\epsilon$ influences the magnitude of $Q_{OOD} / Q_{ID}$ ratio, the fundamental signal required for our method persists. (The slight trend reversal for ImageNet-O is interesting and, as we will discuss in the paper, likely due to its construction process which involved adversarially targeting a ResNet-50).
>
> *   **(b) On the performance of AdaSCALE with adversarially trained networks:**
>     *   **AdaSCALE consistently outperforms baselines on adversarially trained models, even without any specific hyperparameter tuning.**
>
>         We evaluated AdaSCALE's performance on these models using the *exact same hyperparameters* as the vanilla ResNet-50. The results show that AdaSCALE's adaptive approach remains superior to the baselines.
>
> | Training | Method | Near-OOD (FPR95↓ / AUROC↑) | Far-OOD (FPR95↓ / AUROC↑) |
> | :--- | :--- | :---: | :---: |
> | **$\\epsilon=0.01$** | SCALE | 68.54 / 75.43 | 25.95 / 94.40 |
> | | OptFS | 70.36 / 72.97 | 28.60 / 93.24 |
> | | **AdaSCALE-A (ours)** | **60.73 / 80.84** | **19.90 / 95.74** |
> | --- | --- | --- | --- |
> | **$\\epsilon=0.05$** | SCALE | 66.99 / 76.10 | 24.77 / 94.40 |
> | | OptFS | 70.11 / 73.51 | 28.32 / 93.42 |
> | | **AdaSCALE-A (ours)** | **60.21 / 80.68** | **21.13 / 95.35** |
>
> These results highlight the robustness of AdaSCALE. We will add this full analysis and discussion to the appendix to further strengthen our paper's empirical validation.
>
> ---
>
> We hope these clarifications and new results have fully addressed the reviewer GsBJ's questions. We are confident that incorporating this analysis will make our paper even stronger. We thank the reviewer GsBJ again for valuable feedbacks and for helping us improve our work.

---

> > ### Comment · Reviewer_GsBJ · 2025-08-05
> >
> > The authors have addressed both points raised in the review. If space permits, I would encourage the authors to include the additional results on adversarial training in the final manuscript.

---

> > > ### Author Response · Authors · 2025-08-07
> > >
> > > We sincerely thank Reviewer GsBJ for the genuinely insightful feedback regarding the method's efficacy in the context of adversarially trained model. We will include it in the final manuscript.

---

### Decision · Program_Chairs · 2025-09-17

**Decision:**

Reject

**Comment:**

This paper studies the OOD detection problem and proposes a new method AdaSCALE. The authors claim that the proposed method applies an adaptive scaling procedure for the percentile threshold, which is different from the previous method. The proposed method is motivated by more pronounced activation shifts of OOD samples under certain scenarios. The authors have also conducted necessary experiments to verify the effectiveness of the proposed method. Although this paper has some contributions regarding the proposed method and the empirical results, there are multiple concerns raised by the reviewer. As is proposed my more than one reviewers, the hyperparameter sensitivity and the score combining problem are critical to the generalization, while the authors have not proposed enough convincing explanations or evidence to address these problem. After the rebuttal, these limitations have not been adequately addressed. Therefore, I recommend rejecting this paper and suggest the authors improve the manuscript in the future.